# Stein $\Pi$-Importance Sampling

**Congye Wang**[1]**, Wilson Ye Chen**[2]**, Heishiro Kanagawa**[1]**, Chris. J. Oates**[1]
[1] Newcastle University, UK
[2] University of Sydney, Australia

## Abstract

Stein discrepancies have emerged as a powerful tool for retrospective improvement of Markov chain Monte Carlo output. However, the question of how to *design* Markov chains that are well-suited to such post-processing has yet to be addressed. This paper studies Stein importance sampling, in which weights are assigned to the states visited by a $\Pi$-invariant Markov chain to obtain a consistent approximation of $P$, the intended target. Surprisingly, the optimal choice of $\Pi$ is not identical to the target $P$; we therefore propose an explicit construction for $\Pi$ based on a novel variational argument. Explicit conditions for convergence of *Stein $\Pi$-Importance Sampling* are established. For $\approx 70\%$ of tasks in the `PosteriorDB` benchmark, a significant improvement over the analogous post-processing of $P$-invariant Markov chains is reported.

## 1 Introduction

Stein discrepancies are a class of statistical divergences that can be computed without access to a normalisation constant. Originally conceived as a tool to measure the performance of sampling methods (Gorham and Mackey, 2015), these discrepancies have since found wide-ranging statistical applications (see the review of Anastasiou et al., 2023). Our focus here is the use of Stein discrepancies for retrospective improvement of Markov chain Monte Carlo (MCMC), and here two main techniques have been proposed: (i) *Stein importance sampling* (Liu and Lee, 2017; Hodgkinson et al., 2020), and (ii) *Stein thinning* (Riabiz et al., 2022). In Stein importance sampling (also called *black box importance sampling*), the samples are assigned weights such that a Stein discrepancy between the weighted empirical measure and the target $P$ is minimised. Stein thinning constructs a sparse approximation to this optimally weighted measure at a lower computational and storage cost. Together, these techniques provide a powerful set of post-processing tools for MCMC, with subsequent authors proposing a range of generalisations and extensions (Teymur et al., 2021; Chopin and Ducrocq, 2021; Hawkins et al., 2022; Fisher and Oates, 2023; Bénard et al., 2023).

The consistency of these algorithms has been established in the setting of approximate, $\Pi$-invariant MCMC, motivated by challenging inference problems where only approximate sampling can be performed. In these settings, $\Pi$ is implicitly an approximation to $P$ that is as accurate as possible subject to computational budget. However, the critical question of how to *design* Markov chains that are well-suited to such post-processing has yet to be addressed. This paper provides a solution, in the form of a specific construction for $\Pi$ derived from a novel variational argument. Surprisingly, we are able to demonstrate a substantial improvement using the proposed $\Pi$, compared to the case where $\Pi$ and $P$ are equal. The paper proceeds as follows: Section 2 presents an abstract formulation of the task and existing results for optimally-weighted empirical measures are reviewed. Section 3 derives our proposed choice of $\Pi$ and establishes that Stein post-processing of samples from a $\Pi$-invariant Metropolis-adjusted Langevin algorithm (MALA) provides a consistent approximation of $P$. The approach is stress-tested using the recently released `PosteriorDB` suite of benchmark tasks in Section 4, before concluding with a discussion in Section 5.

37th Conference on Neural Information Processing Systems (NeurIPS 2023).

## 2   Background

To properly contextualise our discussion we start with an abstract mathematical description of the task. Let $P$ be a probability measure on a measurable space $\mathcal{X}$. Let $\mathcal{P}(\mathcal{X})$ be the set of all probability measures on $\mathcal{X}$. Let $D_P : \mathcal{P}(\mathcal{X}) \to [0, \infty]$ be a *statistical divergence* for measuring the quality of an approximation $Q$ to $P$, meaning that $D_P(Q) = 0$ if and only if $Q = P$. In this work we consider approximations whose support is contained in a finite set $\{x_1, \dots, x_n\} \subset \mathcal{X}$, and in particular we consider *optimal* approximations of the form

$$P_n^\star = \sum_{i=1}^n w_i^\star \delta(x_i), \qquad w^\star \in \operatorname*{arg\,min}_{w \geq 0, \; 1^\top w = 1} D_P \left( \sum_{i=1}^n w_i \delta(x_i) \right).$$

In what follows we restrict attention to statistical divergences for which such approximations can be shown to exist and be well-defined. The question that we then ask is *which states* $\{x_1, \dots, x_n\}$ *minimise the approximation error* $D_P(P_n^\star)$? Before specialising to Stein discrepancies, it is helpful to review existing results for some standard statistical divergences $D_P$.

### 2.1   Wasserstein Divergence

*Optimal quantisation* focuses on the $r$-Wasserstein ($r \geq 1$) family of statistical divergences $D_P(Q) = \inf_{\gamma \in \Gamma(P,Q)} \int \|x - y\|^r \mathrm{d}\gamma(x, y)$, where $\Gamma(P, Q)$ denotes the set of all *couplings*[1] of $P, Q \in \mathcal{P}(\mathbb{R}^d)$, and the divergence is finite whenever $P$ and $Q$ have finite $r$-th moment. Assuming the states $\{x_1, \dots, x_n\}$ are distinct, the corresponding optimal weights are $w_i^\star = P(A_i)$ where $A_i$ is the *Voronoi neighbourhood*[2] of $x_i$ in $\mathbb{R}^d$. Optimal states achieve the *minimal quantisation error* for $P$;

$$e_{n,r}(P) = \inf_{x_1, \dots, x_n \in \mathbb{R}^d} D_P \left( \sum_{i=1}^n w_i^\star \delta(x_i) \right),$$

the smallest value of the divergence among optimally-weighted distributions supported on at most $n$ states. Though the dependence of optimal states on $n$ and $P$ can be complicated, we can broaden our perspective to consider *asymptotically* optimal states, whose asymptotic properties can be precisely characterised. To this end, for $A \subset \mathbb{R}^d$, let $\mathcal{U}(A)$ denote the uniform distribution on $A$, and define the universal constant $C_r([0, 1]^d) = \inf_{n \geq 1} n^{r/d} e_{n,r}(\mathcal{U}([0, 1]^d))$. Suppose that $P$ admits a density $p$ on $\mathbb{R}^d$. Then the $r$th *quantisation coefficient* of $P$ on $\mathbb{R}^d$, defined as

$$C_r(P) = C_r([0, 1]^d) \left( \int p(x)^{d/(d+r)} \, \mathrm{d}x \right)^{(d+r)/d},$$

plays a central role in the classical theory of quantisation, being the rate constant in the asymptotic convergence of the minimal quantisation error; $\lim_{n \to \infty} n^{r/d} e_{n,r}(P) = C_r(P)$; see Theorem 6.2 of Graf and Luschgy (2007). This suggests a natural definition; a collection $\{x_1, \dots, x_n\}$ is called *asymptotically optimal* if

$$\lim_{n \to \infty} n^{r/d} D_P \left( \sum_{i=1}^n w_i^\star \delta(x_i) \right) = C_r(P),$$

which amounts to $P_n^\star$ asymptotically attaining the minimal quantisation error $e_{n,r}(P)$. The main result here is that, if $\{x_1, \dots, x_n\}$ are asymptotically optimal, then $\frac{1}{n} \sum_{i=1}^n \delta(x_i) \to \Pi_r$, where convergence is in distribution and $\Pi_r$ is the distribution whose density is $\pi_r(x) \propto p(x)^{d/(d+r)}$; see Theorem 7.5 of Graf and Luschgy (2007). This provides us with a key insight; optimal states are *over-dispersed* with respect to the intended distributional target. The extent of the over-dispersion here depends both on $r$, a parameter of the statistical divergence, and the dimension $d$ of the space on which distributions are defined.

The $r$-Wasserstein divergence is, unfortunately, not well-suited for use in the motivating Bayesian context. In particular, computing the optimal weights $w_i = P(A_i)$ requires knowledge of $P$, which is typically not available when $P$ is implicitly defined via an intractable normalisation constant. On

---

[1]A coupling $\gamma \in \Gamma(P, Q)$ is a distribution $\gamma \in \mathcal{P}(\mathbb{R}^d \times \mathbb{R}^d)$ whose marginal distributions are $P$ and $Q$.

[2]The Voronoi neighbourhood of $x_i$ is the set $A_i = \{x \in \mathbb{R}^d : \|x - x_i\| = \min_{j=1,\dots,n} \|x - x_j\|\}$.

the other hand, the optimal sampling distribution $\Pi$ is explicit and can be sampled (for example using MCMC); for discussion of random quantisers in this context see Graf and Luschgy (2007, Chapter 9), Cohort (2004, p126) and Sonnleitner (2022, Section 4.5). The simple form of $\Pi$ is a feature of the classical approach to quantisation that we will attempt to mimic in the sequel.

## 2.2 Kernel Discrepancies

The theory of quantisation using kernels is less well-developed. A *kernel* is a measurable, symmetric, positive-definite function $k : \mathcal{X} \times \mathcal{X} \to \mathbb{R}$. From the Moore–Aronszajn theorem, there is a unique Hilbert space $\mathcal{H}(k)$ for which $k$ is a reproducing kernel, meaning that $k(\cdot, x) \in \mathcal{H}(k)$ for all $x \in \mathcal{X}$ and $\langle f, k(\cdot, x) \rangle_{\mathcal{H}(k)} = f(x)$ for all $f \in \mathcal{H}(k)$ and all $x \in \mathcal{X}$. Assuming that $\mathcal{H}(k) \subset L^1(P)$, we can define the weak (or *Pettis*) integral

$$\mu_P(\cdot) = \int k(\cdot, x) \, \mathrm{d}P(x), \tag{1}$$

called the *kernel mean embedding* of $P$ in $\mathcal{H}(k)$. The *kernel discrepancy* is then defined as the norm of the difference between kernel mean embeddings

$$D_P(Q) = \|\mu_Q - \mu_P\|_{\mathcal{H}(k)} = \sqrt{\iint k(x, y) \, \mathrm{d}(Q - P)(x) \mathrm{d}(Q - P)(y)} \tag{2}$$

where, to be consistent with our earlier notation, we adopt the convention that $D_P(Q)$ is infinite whenever $\mathcal{H}(k) \not\subset L^1(Q)$. The second equality in (2) follows immediately from the stated properties of a reproducing kernel. To satisfy the requirement of a statistical divergence, we assume that the kernel $k$ is *characteristic*, meaning that $\mu_P = \mu_Q$ if and only if $P = Q$. In this setting, the properties of optimal states are necessarily dependent on the choice of kernel $k$, and are in general not well-understood. Indeed, given distinct states $\{x_1, \ldots, x_n\}$, the corresponding optimal weights $w^\star = (w_1^\star, \ldots, w_n^\star)^\top$ are the solution to the linearly-constrained quadratic program

$$\underset{w \in \mathbb{R}^d}{\arg\min} \; w^\top K w - 2z^\top w \qquad \text{s.t.} \qquad w \geq 0, \; 1^\top w = 1 \tag{3}$$

where $K_{i,j} = k(x_i, x_j)$ and $z_i = \mu_P(x_i)$. This program does not admit a closed-form solution, but can be numerically solved. To the best of our knowledge, the only theoretical analysis of approximations based on (3) is due to Hayakawa et al. (2022), who established rates for the convergence of $P_n^\star$ to $P$ in the case where states are independently sampled from $P$. The question of an optimal sampling distribution was not considered in that work.

Although few results are available concerning (3), relaxations of this program have been well-studied. The simplest relaxation of (3) is to remove both the positivity ($w \geq 0$) and normalisation ($1^\top w = 1$) constraints, in which case the optimal weights have the explicit representation $w^* = K^{-1}z$. The analysis of optimal states in this context has developed under the dual strands of *kernel cubature* and *Bayesian cubature*, where it has been theoretically or empirically demonstrated that (i) if states are randomly sampled, the optimal sampling distribution will be $n$-dependent (Bach, 2017) and over-dispersed with respect to the distributional target (Briol et al., 2017), and (ii) *space-filling* designs are asymptotically optimal for typical stationary kernels on bounded domains $\mathcal{X} \subset \mathbb{R}^d$ (Briol et al., 2019). Analysis of optimal states on unbounded domains appears to be more difficult; see e.g. Karvonen et al. (2021). Relaxation of either the positivity or normalisation constraints results in approximations that behave similarly to kernel cubature (see, respectively, Ehler et al., 2019; Karvonen et al., 2018). However, relaxation of either constraint can result in the failure of $P_n^\star$ to be an element of $\mathcal{P}(\mathcal{X})$, limiting the relevance of these results to the posterior approximation task.

Despite relatively little being known about the character of optimal states in this context, kernel discrepancy is widely used. The application of kernel discrepancies to an implicitly defined distributional target, such as a posterior distribution in a Bayesian analysis, is made possible by the use of a *Stein kernel*; a $P$-dependent kernel $k = k_P$ for which $\mu_P(x) = 0$ for all $x \in \mathcal{X}$ (Oates et al., 2017). The associated kernel discrepancy

$$D_P(Q) = \|\mu_Q\|_{\mathcal{H}(k_P)} = \sqrt{\iint k_P(x, y) \, \mathrm{d}Q(x) \mathrm{d}Q(y)} \tag{4}$$

is called a *kernel Stein discrepancy (KSD)* (Chwialkowski et al., 2016; Liu et al., 2016; Gorham and Mackey, 2017), and this will be a key tool in our methodological development. The corresponding optimally weighted approximation $P_n^\star$ is the Stein importance sampling method of Liu and Lee (2017). To retain clarity of presentation in the main text, we defer all details on the construction of Stein kernels to Appendix A.

### 2.3 Sparse Approximation

If the number $n$ of states is large, computation of optimal weights can become impractical. This has motivated a range of sparse approximation techniques, which aim to iteratively construct an approximation of the form $P_{n,m} = \frac{1}{m} \sum_{i=1}^{m} \delta(y_i)$, where each $y_i$ is an element from $\{x_1, \ldots, x_n\}$. The canonical example is the greedy algorithm which, at iteration $j$, selects a state

$$y_j \in \operatorname*{arg\,min}_{y \in \{x_1, \ldots, x_n\}} D_P \left( \frac{1}{j} \delta(y) + \frac{1}{j} \sum_{i=1}^{j-1} \delta(y_i) \right) \tag{5}$$

for which the statistical divergence is minimised. In the context of kernel discrepancy, the greedy algorithm (5) has computational cost $O(m^2 n)$, which compares favourably[3] with the cost of solving (3) when $m \ll n$. Furthermore, under appropriate assumptions, the sparse approximation converges to the optimally weighted approximation; $D_P(P_{n,m}) \to D_P(P_n^\star)$ as $m \to \infty$ with $n$ fixed. See Teymur et al. (2021) for full details, where non-myopic and mini-batch extensions of the greedy algorithm are also considered. The greedy algorithm can be viewed as a regularised version of the *Frank–Wolfe* algorithm (also called *herding*, or the *conditional gradient* method), for which a similar asymptotic result can be shown to hold (Chen et al., 2010; Bach et al., 2012; Chen et al., 2018). Related work includes Dwivedi and Mackey (2021, 2022); Shetty et al. (2022); Hayakawa et al. (2022). Since in what follows we aim to retrospectively improve MCMC output, where it is not unusual to encounter $n \approx 10^4$–$10^6$, sparse approximation will be important.

This completes our overview of background material. In what follows we seek to mimic classical quantisation by deriving a choice for $\Pi$ that is straight-forward to sample using MCMC and is appropriately over-dispersed relative to $P$. This should be achieved while remaining in the framework of kernel discrepancies, so that optimal weights can be explicitly computed, and coupled with a sparse approximation that has low computational and storage cost.

## 3 Methodology

The methods that we consider first sample states $\{x_1, \ldots, x_n\}$ using $\Pi$-invariant MCMC, then post-process these states using kernel discrepancies (Section 2.2) and sparse approximation (Section 2.3), to obtain an approximation to the target $P$. A variational argument, which we present in Section 3.1, provides a suitable $n$-independent choice for $\Pi$ (which agrees with our intuition from Section 2.1 that $\Pi$ should be in some appropriate sense over-dispersed with respect to $P$). Sufficient conditions for strong consistency of the approximation are established in Section 3.3.

### 3.1 Selecting $\Pi$

Here we present a heuristic argument for a particular choice of $\Pi$; rigorous theoretical support for *Stein $\Pi$-Importance Sampling* is then provided in Section 3.3. Our setting is that of Section 2.2, and the following will additionally be assumed:

**Assumption 1.** *It is assumed that*

(A1) $C_1^2 := \inf_{x \in \mathcal{X}} k(x, x) > 0$

(A2) $C_2 := \int \sqrt{k(x, x)} \, \mathrm{d}P(x) < \infty.$

---

[3] It is difficult to quantify the complexity of numerically solving (3), since this will depend on details of the solver and the tolerance that are used. On the other hand, if we ignore the non-negativity and normalisation constraints, then we can see that the computational cost of solving the $n$-dimensional linear system of equations is $O(n^3)$.

Note that (A2) implies that $\mathcal{H}(k) \subset L^1(P)$, and thus (1) is in fact a strong (or *Bochner*) integral.

A direct analysis of the optimal states associated to the optimal weights $w^\star$ appears to be challenging due to the fact that the components of $w^\star$ are strongly inter-dependent. Our solution here is to instead consider optimal states associated with weights that, while not optimal, can be expected to perform much better than alternatives, with the advantage that their components are only weakly dependent. Specifically, we will be assuming that $P$ is absolutely continuous with respect to $\Pi$ (denoted $P \ll \Pi$), and study convergence of self-normalised importance sampling (SNIS), i.e. the approximation $P_n = \sum_{i=1}^{n} w_i \delta(x_i)$, $w_i \propto (\mathrm{d}P/\mathrm{d}\Pi)(x_i)$, where $x_1, \ldots, x_n \sim \Pi$ are independent. Since $w \geq 0$ and $1^\top w = 1$, from the optimality of $w^\star$ under these constraints we have that $D_P(P_n^\star) \leq D_P(P_n)$. It is emphasised that the SNIS weights are a theoretical device only, and will not be used for computation; indeed, we can demonstrate that the SNIS weights $w$ perform substantially worse than $w^\star$ in general.

The analysis of SNIS weights $w$ is tractable when viewed as approximation of the kernel mean embedding $\mu_P$ in the Hilbert space $\mathcal{H}(k)$. Indeed, recall that $D_P(P_n) = \|\xi_n/\sqrt{n}\|_{\mathcal{H}(k)}$ where $\xi_n = \sqrt{n}(\mu_{P_n} - \mu_P)$. Then, following Section 2.3.1 of Agapiou et al. (2017), we observe that

$$\xi_n = \sqrt{n} \left( \sum_{i=1}^{n} w_i k(\cdot, x_i) - \mu_P \right) = \frac{\frac{1}{\sqrt{n}} \sum_{i=1}^{n} \frac{\mathrm{d}P}{\mathrm{d}\Pi}(x_i) \left[ k(\cdot, x_i) - \mu_P \right]}{\frac{1}{n} \sum_{i=1}^{n} \frac{\mathrm{d}P}{\mathrm{d}\Pi}(x_i)}. \tag{6}$$

The idea is to seek $\Pi$ for which the asymptotic variance of $\xi_n$ is small. Supposing that

$$\int \frac{\mathrm{d}P}{\mathrm{d}\Pi}(x)^2 \, \mathrm{d}\Pi(x) < \infty, \tag{S1}$$

from the weak law of large numbers the denominator in (6) converges in probability to 1. Further supposing that

$$\int \left\| \frac{\mathrm{d}P}{\mathrm{d}\Pi}(x)[k(\cdot, x) - \mu_P] \right\|_{\mathcal{H}(k)}^2 \, \mathrm{d}\Pi(x) < \infty, \tag{S2}$$

from the Hilbert space central limit theorem the numerator in (6) converges in distribution to a Gaussian $\frac{1}{\sqrt{n}} \sum_{i=1}^{n} (\mathrm{d}P/\mathrm{d}\Pi)(x_i) [k(\cdot, x_i) - \mu_P] \xrightarrow{\mathrm{d}} \mathcal{N}(0, \mathcal{C})$ where $\mathcal{C} : \mathcal{H}(k) \to \mathcal{H}(k)$ is the covariance operator defined via

$$\langle f, \mathcal{C}g \rangle_{\mathcal{H}(k)} = \int \left\langle f, \frac{\mathrm{d}P}{\mathrm{d}\Pi}(x)[k(\cdot, x) - \mu_P] \right\rangle_{\mathcal{H}(k)} \left\langle g, \frac{\mathrm{d}P}{\mathrm{d}\Pi}(x)[k(\cdot, x) - \mu_P] \right\rangle_{\mathcal{H}(k)} \, \mathrm{d}\Pi(x),$$

see Section 10.1 of Ledoux and Talagrand (1991). Thus, from Slutsky's lemma applied to (6), we conclude that $\xi_n \xrightarrow{\mathrm{d}} \mathcal{N}(0, \mathcal{C})$. Recalling that $n D_P(P_n)^2 = \|\xi_n\|_{\mathcal{H}(k)}^2$, and noting that the mean square of the limiting Gaussian random variable is $\mathrm{tr}(\mathcal{C})$, a natural idea is to select the sampling distribution $\Pi$ such that $\mathrm{tr}(\mathcal{C})$ is minimised.

Fortunately, the trace of $\mathcal{C}$ can be explicitly computed. It simplifies presentation to restrict attention to a Stein kernel $k = k_P$, for which $\mu_P = 0$, giving $\mathrm{tr}(\mathcal{C}) = \int (\mathrm{d}P/\mathrm{d}\Pi)(x)^2 k_P(x) \, \mathrm{d}\Pi(x)$, where for convenience we have let $k_P(x) := k_P(x, x)$. Assuming that $P$ and $\Pi$ admit densities $p$ and $\pi$ on $\mathcal{X} = \mathbb{R}^d$, the variational problem we wish to solve is

$$\arg\min_{\pi \in \mathcal{Q}} \int \frac{p(x)^2}{\pi(x)} k_P(x) \, \mathrm{d}x \quad \text{s.t.} \quad \int \pi(x) \, \mathrm{d}x = 1, \tag{7}$$

where $\mathcal{Q}$ be the set of positive measures on $\mathbb{R}^d$ for which (S1-2) are satisfied. To solve this problem, we first relax the constraints (S1-2) and solve the relaxed problem using the Euler–Lagrange equations, which yield

$$\pi(x) \propto p(x) \sqrt{k_P(x)}. \tag{8}$$

Note that the normalisation constant of $\pi$ is $C_2$ from (1), whose existence we assumed. Then we verify that (S1-2) in fact hold for this choice of $\Pi$. Indeed,

$$(\text{S1}) = \int \frac{\mathrm{d}P}{\mathrm{d}\Pi}(x)^2 \, \mathrm{d}\Pi(x) = C_2 \int \frac{1}{k_P(x)} \, \mathrm{d}\Pi(x) \leq \frac{C_2}{C_1^2} < \infty$$

$$(\text{S2}) = \int \frac{\mathrm{d}P}{\mathrm{d}\Pi}(x)^2 k_P(x) \, \mathrm{d}\Pi(x) = C_2 \int \sqrt{k_P(x)} \, \mathrm{d}P(x) = C_2^2 < \infty,$$

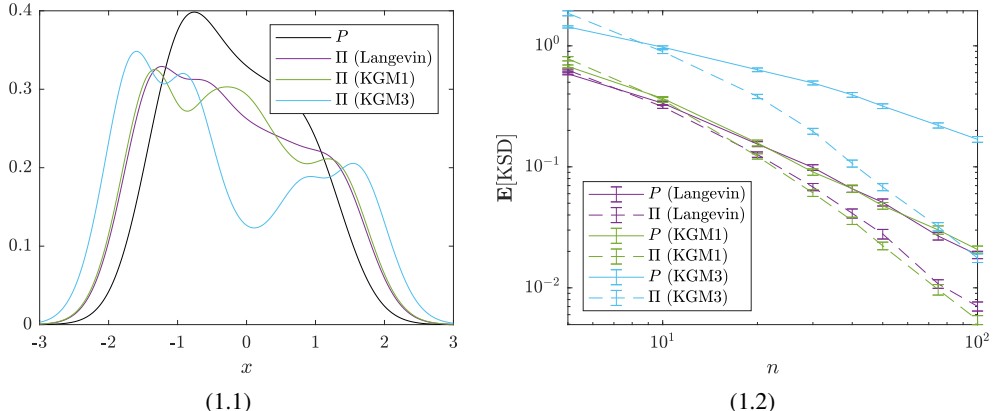

(1.1)        (1.2)

Figure 1: Illustrating our choice of $\Pi$ in 1D. (a) The univariate target $P$ (black), and our choice of $\Pi$ based on the Langevin–Stein kernel (purple), the KGM1–Stein kernel (green), and the KGM3–Stein kernel (blue). (b) The mean kernel Stein discrepancy (KSD) for Stein $\Pi$-Importance Sampling using the Stein kernels from (a); in each case, KSD was computed using the same Stein kernel used to construct $\Pi$. Solid lines indicate the baseline case of sampling from $P$, while dashed lines indicate sampling from $\Pi$. (The experiment was repeated 100 times and standard error bars are plotted.)

which shows that we have indeed solved (7). The sampling distribution $\Pi$ we have obtained is characterised up to a normalisation constant in (8), so just like $P$ we can sample from $\Pi$ using techniques such as MCMC. It is interesting to note that $\Pi$ is also optimal for standard importance sampling (i.e. without self-normalisation); see Lemma 1 of Adachi et al. (2022). The Stein kernel $k_P$ determines the extent to which $\Pi$ differs from $P$, as we illustrate next.

### 3.2 Illustration

For illustration, consider the univariate target $P$ (black curve) in Figure 1.1, a 3-component Gaussian mixture model. Our recommended choice of $\Pi$ in (8) is shown for both the *Langevin*–Stein kernel (purple curve) and the *KGMs*–Stein kernels with $s \in \{1, 3\}$ (green and blue curves). The Stein discrepancy corresponding to the Langevin–Stein kernel provides control over weak convergence (i.e. convergence of integrals of functions that are continuous and bounded), while the KGM$s$–Stein kernel provides additional control over the convergence of polynomial moments up to order $s$; full details about the construction of Stein kernels are contained in Appendix A. The Langevin and KGM1–Stein kernels have $k_P(x) \asymp x^2$, while the KGM3–Stein kernel has $k_P(x) \asymp x^6$, in each case as $|x| \to \infty$, and thus greater over-dispersion results from use of the KGM3–Stein kernel. This over-dispersion is less pronounced[4] in higher dimensions; see Appendix D.1.

To illustrate the performance of Stein $\Pi$-Importance Sampling, we generated a sequence $(x_n)_{n \in \mathbb{N}}$ of independent samples from $\Pi$. For each $n \in \{1, \ldots, 100\}$, the samples $\{x_1, \ldots, x_n\}$ were assigned optimal weights $w^\star$ by solving (3), and the associated KSD was computed. As a baseline, we performed the same calculation using independent samples from $P$. Figure 1.2 indicates that, for both Stein kernels, substantial improvement results from the use of samples from $\Pi$ compared to the use of samples from $P$. Interestingly, the KGM3–Stein kernel demonstrated a larger improvement compared to the Langevin–Stein kernel, suggesting that the choice of $\Pi$ may be more critical in settings where KSD enjoys a stronger form of convergence control.

To illustrate a posterior approximation task, consider a simple regression model $y_i = f_i(x) + \epsilon_i$ with $f_i(x) = x_1(1 + t_i x_2)$, $t_i = i - 5$, $i = 1, \ldots, 10$, with $\epsilon_i$ independent $\mathcal{N}(0, 1)$. The parameter $x = (x_1, x_2)$ was assigned a prior $\mathcal{N}(0, I)$. Data were simulated using $x = (0, 0)$. The posterior distribution $P$ is depicted in the leftmost panel of Figure 2, while our choice of $\Pi$ corresponding to the Langevin (centre left), KGM3 (centre right) and *Riemann*–Stein kernels (right) are also displayed. For the Langevin and KGM3 kernels, the associated $\Pi$ target their mass toward regions where $P$ varies the most. The reason for this behaviour is clearly seen for the Langevin–Stein kernel since

---

[4]The same holds for classical quantisation; c.f. Section 2.1.

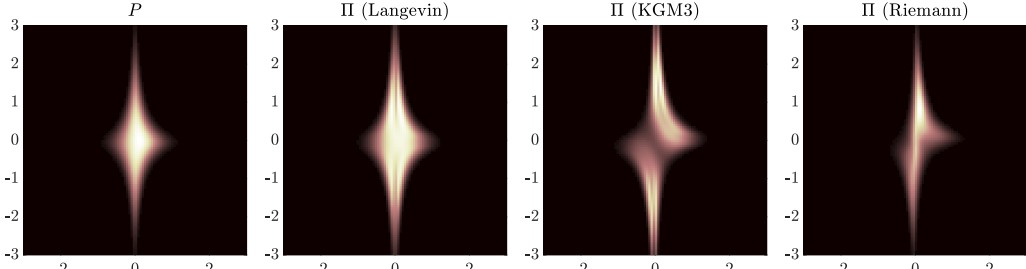

Figure 2: Illustrating our choice of $\Pi$ in 2D. The bivariate target $P$ (left), together with our choice of $\Pi$ based on the Langevin–Stein kernel (centre left), the KGM3–Stein kernel (centre right), and the Riemann–Stein kernel (right).

---

**Algorithm 1** $\Pi$-Invariant Metropolis-Adjusted Langevin Algorithm (MALA)

---

**Require:** $x_0$ (initial state), $\epsilon$ (step size), $M$ (preconditioner matrix), $n$ (chain length), $k_P$ (Stein kernel)
1: **for** $i = 1, \ldots, n$ **do**
2: $\quad x' \leftarrow \underbrace{x_{i-1} + \epsilon M^{-1} \nabla \log p(x_{i-1}) + \frac{\epsilon}{2} M^{-1} \nabla \log k_P(x_{i-1})}_{=:\nu(x_{i-1})} + \sqrt{2\epsilon} M^{-1/2} Z_i \qquad \triangleright Z_i \overset{\text{IID}}{\sim} \mathcal{N}(0, I)$
3: $\quad L \leftarrow \log\left(\frac{p(x')}{p(x_{i-1})}\right) + \frac{1}{2} \log\left(\frac{k_P(x')}{k_P(x_{i-1})}\right) - \frac{1}{4\epsilon} \|x_{i-1} - \nu(x')\|_{M^{-1}}^2 + \frac{1}{4\epsilon} \|x' - \nu(x_{i-1})\|_{M^{-1}}^2$
4: $\quad$ **if** $\log(U_i) < L$ **then** $x_i \leftarrow x'$; **else** $x_i \leftarrow x_{i-1}$; **end if** $\qquad \triangleright U_i \overset{\text{IID}}{\sim} \mathcal{U}([0, 1])$
5: **end for**

---

**Algorithm 2** Stein $\Pi$-Importance Sampling (S$\Pi$IS-MALA)

---

**Require:** $\{x_1, \ldots, x_n\}$ from Algorithm 1, $k_P$ (Stein kernel)
1: $w^\star \in \arg\min_{w \in \mathbb{R}^d} \{\langle w, K_P w\rangle \ : \ w \geq 0, \ 1^\top w = 1\}$ $\qquad \triangleright [K_P]_{i,j} = k_P(x_i, x_j)$

---

$k_P(x) = c_1 + c_2 \|\nabla \log p(x)\|^2$ for some $c_1, c_2 > 0$; see Appendix C for detail. The Riemann–Stein kernel can be viewed as a preconditioned form of the Langevin–Stein kernel which takes into account the geometric structure of $P$; see Appendix A for full detail[5]. Results in Figure S2 demonstrate that Stein $\Pi$-Importance Sampling improves upon the default Stein importance sampling method (i.e. with $\Pi$ and $P$ equal) for all choices of kernel.

An additional illustration involving a GARCH model with $d = 4$ parameters is presented in Appendix D.4, where the effect of varying the order $s$ of the KGM–Stein kernel is explored.

### 3.3 Theoretical Guarantees

The aim of this section is to establish when post-processing of $\Pi$-invariant MCMC produces a strongly consistent approximation of $P$, for our recommended choice of $\Pi$ in (8). Our analysis focuses on MALA (Roberts and Stramer, 2002), leveraging the recent work of Durmus and Moulines (2022) to present explicit and verifiable conditions on $P$ for our results to hold. In fact, we consider the more general *preconditioned* form of MALA, where the symmetric positive definite preconditioner matrix $M$ is to be specified. Our results also allow for (optional) sparse approximation, to circumvent direct solution of (3) (c.f. Section 2.3). The resulting algorithms, which we call *Stein $\Pi$-Importance Sampling* (S$\Pi$IS-MALA) and *Stein $\Pi$-Thinning* (S$\Pi$T-MALA), are quite straight-forward and contained, respectively, in Algorithms 2 and 3. The linearly-constrained quadratic programme in Algorithm 2 was solved using the `Python v3.10.4` packages `qpsolvers v3.4.0` and `proxsuite v0.3.7`. While it is difficult to analyse the computational complexity associated with these methods, we believe they are at worst $O(n^3)$.

---

[5]The use of geometric information may be beneficial, in the sense that the associated diffusion process may mix more rapidly, and rapid mixing leads to sharper bounds from the perspective of convergence control (Gorham et al., 2019). However, the Riemann–Stein kernel is associated with a prohibitive computational cost; it is included here only for academic interest.

---

**Algorithm 3** Stein $\Pi$-Thinning (S$\Pi$T-MALA)

---

**Require:** $\{x_1, \ldots, x_n\}$ from Algorithm 1, $m$ (number of samples to retain), $k_P$ (Stein kernel)

1: **for** $i = 1, \ldots, m$ **do**
2:      $y_i \leftarrow \arg\min_{y \in \{x_1, \ldots, x_n\}} \frac{1}{2} k_P(y) + \sum_{j=1}^{i-1} k_P(y, y_j)$
3: **end for**

---

Let $A \preceq B$ indicate that $A - B$ is a positive semi-definite matrix for $A, B \in \mathbb{R}^{n \times n}$. For a symmetric positive definite matrix $A$ let $\|z\|_A := \sqrt{z^\top A^{-1} z}$ for $z \in \mathbb{R}^d$. Let $C^s(\mathbb{R}^d)$ denote the set of $s$-times continuously differentiable real-valued functions on $\mathbb{R}^d$.

**Theorem 1** (Strong consistency of S$\Pi$IS- and S$\Pi$T-MALA). *Let Assumption 1 hold where $k = k_P$ is a Stein kernel, and let $D_P : \mathcal{P}(\mathcal{X}) \to [0, \infty]$ denote the associated KSD. Assume also that*

(A1) $\nabla \log p \in C^2(\mathbb{R}^d)$ *with* $\sup_{x \in \mathbb{R}^d} \|\nabla^2 \log p(x)\| < \infty$

(A2) $\exists\, b_1 > 0, B_1 \geq 0$ *such that* $-\nabla^2 \log p(x) \succeq b_1 I$ *for all* $\|x\| \geq B_1$

(A3) $k_P \in C^2(\mathbb{R}^d)$.

(A4) $\exists\, 0 < b_2 < 2 b_1 C_1^2,\ B_2 \geq 0$ *such that* $\nabla^2 k_P(x) \preceq b_2 I$ *for all* $\|x\| \geq B_2$

*Let $P_n^\star = \sum_{i=1}^n w_i^\star \delta(x_i)$ be the result of running Algorithm 2 and let $P_{n,m} = \frac{1}{m} \sum_{i=1}^m \delta(y_i)$ be the result of running Algorithm 3. Let $m \leq n$ and $m = \Omega((\log n)^\delta)$ for some $\delta > 2$. Then there exists $\epsilon_0 > 0$ such that, for all step sizes $\epsilon \in (0, \epsilon_0)$ and all initial states $x_0 \in \mathbb{R}^d$, $D_P(P_n^\star) \to 0, D_P(P_{n,m}) \to 0$ almost surely as $m, n \to \infty$.*

The proof is in Appendix B. Compared to earlier authors[6], such as Chen et al. (2019); Riabiz et al. (2022), a major novelty here is that our assumptions are *explicit* and can often be verified (see also Hodgkinson et al., 2020). (A2) is strong log-concavity of $P$ when $B_1 = 0$, while for $B_1 > 0$ this condition is slightly stronger than the related *distant dissipativity* condition assumed in earlier work (Gorham and Mackey, 2017; Riabiz et al., 2022). (A4) holds for the Langevin–Stein kernel (i.e. weak convergence control) and for the KGM1–Stein kernel (i.e. weak convergence control + control over first moments), but not for the higher-order KGM–Stein kernels. Extending our proof strategy to the higher-order KGM–Stein kernels would require further research into the convergence properties of MALA, and this is expected to be difficult.

## 4    Benchmarking on `PosteriorDB`

The area of Bayesian computation has historically lacked a common set of benchmark problems, with classical examples being insufficiently difficult and case-studies being hand-picked (Chopin and Ridgway, 2017). To introduce objectivity into our assessment, we exploited the recently released `PosteriorDB` benchmark (Magnusson et al., 2022). This project is an attempt toward standardised benchmarking, consisting of a collection of posteriors to be numerically approximated. Here, we systematically compared the performance of S$\Pi$IS-MALA against the default Stein importance sampling algorithm (i.e. $\Pi = P$; denoted SIS-MALA), and also against unprocessed $P$-invariant MALA (i.e. uniform weights), reporting results across the breadth of `PosteriorDB`. The test problems in `PosteriorDB` are defined in the `Stan` probabilistic programming language, and so `BridgeStan` (Roualdes et al., 2023) was used to directly access posterior densities and their gradients as required. For all instances of MALA, an adaptive algorithm was used to learn a suitable preconditioner matrix $M$ during the warm-up period; see Appendix D.3. All experiments that we report can be reproduced using code available at `https://github.com/congyewang/Stein-Pi-Importance-Sampling`.

Results are reported in Table 1 for $n = 3 \times 10^3$ samples from MALA. These focus on the Langevin–Stein kernel, for which our theory holds, and the KGM3–Stein kernel, for which it does not. There was a significant improvement of S$\Pi$IS-MALA over SIS-MALA in 73% of test problems for the Langevin–Stein kernel and in 65% of test problems for the KGM3–Stein kernel. Compared to

---

[6]These earlier results required high-level assumptions on the convergence of MCMC, for which explicit sufficient conditions had yet to be derived.

| Task | $d$ | Langevin–Stein Kernel | | | KGM3–Stein Kernel | | |
|---|---|---|---|---|---|---|---|
| | | MALA | SIS - MALA | SΠIS - MALA | MALA | SIS - MALA | SΠIS - MALA |
| earnings-earn_height | 3 | 1.41 | 0.0674 | **0.0332** | 5.33 | 0.656 | **0.181** |
| gp_pois_regr-gp_regr | 3 | 0.298 | 0.0436 | **0.0373** | 1.22 | 0.385 | **0.223** |
| kidiq-kidscore_momhs | 3 | 1.04 | 0.109 | **0.0941** | 4.66 | 0.848 | **0.476** |
| kidiq-kidscore_momiq | 3 | 5.03 | 0.516 | **0.358** | 25.3 | 4.86 | **1.55** |
| mesquite-logmesquite_logvolume | 3 | 1.10 | 0.179 | **0.156** | 4.97 | 1.70 | **0.844** |
| arma-arma11 | 4 | 4.47 | 1.09 | **1.01** | 26.0 | 8.91 | **6.03** |
| earnings-logearn_logheight_male | 4 | 9.46 | 1.96 | **1.59** | 53.9 | 15.4 | **8.65** |
| garch-garch11 | 4 | 0.543 | 0.159 | **0.130** | 4.70 | 1.16 | **1.01** |
| kidiq-kidscore_momhsiq | 4 | 5.21 | 0.982 | **0.897** | 29.3 | 7.25 | **5.05** |
| earnings-logearn_interaction_z | 5 | 3.09 | 1.36 | **1.33** | 19.3 | 10.4 | **8.94** |
| kidiq-kidscore_interaction | 5 | 7.74 | **1.65** | 1.79 | 47.8 | 13.2 | **10.1** |
| kidiq_with_mom_work-kidscore_interaction_c | 5 | 1.35 | **0.659** | 0.711 | 7.92 | **4.05** | 4.17 |
| kidiq_with_mom_work-kidscore_interaction_c2 | 5 | 1.38 | **0.689** | 0.699 | 8.09 | **4.24** | 4.25 |
| kidiq_with_mom_work-kidscore_interaction_z | 5 | 1.11 | 0.500 | **0.499** | 6.62 | **2.63** | 3.25 |
| kidiq_with_mom_work-kidscore_mom_work | 5 | 1.07 | **0.507** | 0.545 | 6.70 | **2.63** | 3.04 |
| low_dim_gauss_mix-low_dim_gauss_mix | 5 | 5.51 | 1.87 | **1.76** | 37.5 | 14.7 | **11.3** |
| mesquite-logmesquite_logva | 5 | 1.83 | 0.821 | **0.818** | 12.6 | 5.73 | **5.59** |
| hmm_example-hmm_example | 6 | 1.99 | 0.578 | **0.523** | 11.6 | 4.13 | **3.40** |
| sblrc-blr | 6 | 479 | 154 | **134** | 3300 | 1100 | **854** |
| sblri-blr | 6 | 201 | 66.7 | **60.3** | 1340 | **514** | 595 |
| arK-arK | 7 | 6.87 | 3.39 | **3.16** | 60.4 | 26.4 | **23.0** |
| mesquite-logmesquite_logvash | 7 | 1.89 | **1.18** | 1.23 | 15.5 | **8.88** | 10.1 |
| bball_drive_event_0-hmm_drive_0 | 8 | 1.15 | **0.679** | 0.698 | 8.55 | 4.72 | **3.99** |
| bball_drive_event_1-hmm_drive_1 | 8 | 42.9 | **11.9** | 12.4 | 285 | 85.6 | **67.8** |
| hudson_lynx_hare-lotka_volterra | 8 | 4.62 | 2.29 | **2.15** | 47.4 | **18.8** | 18.9 |
| mesquite-logmesquite | 8 | 1.46 | **1.00** | 1.06 | 13.3 | **8.28** | 9.14 |
| mesquite-logmesquite_logvas | 8 | 2.02 | **1.31** | 1.35 | 19.2 | **10.8** | 12.2 |
| mesquite-mesquite | 8 | 0.429 | 0.268 | **0.235** | 3.71 | **2.17** | 2.42 |
| eight_schools-eight_schools_centered | 10 | 0.526 | **0.100** | 0.182 | 7.53 | **2.15** | 215 |
| eight_schools-eight_schools_noncentered | 10 | 0.210 | 0.137 | **0.137** | 43.6 | 28.7 | **27.5** |
| nes1972-nes | 10 | 6.16 | 3.89 | **3.45** | 72.9 | 36.2 | **34.4** |
| nes1976-nes | 10 | 6.67 | 3.86 | **3.53** | 77.5 | 35.5 | **34.4** |
| nes1980-nes | 10 | 4.34 | 2.68 | **2.57** | 49.8 | **25.4** | 25.7 |
| nes1984-nes | 10 | 6.18 | 3.75 | **3.43** | 71.3 | 34.9 | **33.6** |
| nes1988-nes | 10 | 7.40 | 3.70 | **3.27** | 81.4 | 34.6 | **32.4** |
| nes1992-nes | 10 | 7.52 | 4.32 | **3.84** | 89.1 | 39.7 | **37.3** |
| nes1996-nes | 10 | 6.44 | 3.87 | **3.53** | 74.1 | 36.4 | **34.3** |
| nes2000-nes | 10 | 3.35 | 2.22 | **2.20** | 38.6 | **21.3** | 22.8 |
| diamonds-diamonds | 26 | 196 | 157 | **143** | 5120 | 2990 | **2620** |
| mcycle_gp-accel_gp | 66 | 11.3 | **8.25** | 9.79 | 960 | **623** | 815 |

Table 1: Benchmarking on `PosteriorDB`. Here we compared raw output from MALA with the post-processed output provided by the default Stein importance sampling method of Liu and Lee (2017) (SIS-MALA) and the proposed Stein Π-Importance Sampling method (SΠIS-MALA). Here $d = \dim(P)$ and the number of MALA samples was $n = 3 \times 10^3$. The Langevin and KGM3–Stein kernels were used for SIS-MALA and SΠIS-MALA and the associated KSDs are reported. Ten replicates were computed and statistically significant improvement is highlighted in **bold**.

unprocessed MALA, a significant improvement occurred in 100% and 97% of cases, respectively for each kernel. However, the extent of improvement decreased when the dimension $d$ of the target increased, supporting the intuition that we set out earlier and in Appendix D.1. An in-depth breakdown of results, including varying the number $n$ of samples that were used, and the performance SΠT-MALA, can be found in Appendices D.5 and D.6.

If $P$ and its gradients are cheap to evaluate, the computational cost of MALA is lower than that of SIS-MALA, and one could run more iterations of MALA for an equivalent computational cost. But for more complex $P$, the computational cost of all algorithms will be gated by the number of times $P$ and its gradients need to be evaluated, making the direct comparison in Table 1 meaningful. Further, if we aim for a compressed representation of $P$, then some form of post-processing of MALA would be required, which would then entail an additional computational cost.

Our focus is on the development of algorithms for minimisation of KSDs; the properties of KSDs themselves are out of scope for this work[7]. Nonetheless, there is much interest in better understanding the properties of KSDs, and we therefore also report performance of SΠIS-MALA in terms of 1-Wasserstein divergence in Appendix D.7. The main contrast between these results and the results in Table 1 is that, being score-based, KSDs suffer from the *blindness to mixing proportions* phenomena which has previously been documented in Wenliang and Kanagawa (2021); Koehler et al. (2022); Liu et al. (2023). Caution should therefore be taken when using algorithms based on Stein discrepancies

---

[7]The interested reader is referred to Gorham and Mackey (2017); Barp et al. (2022b); Kanagawa et al. (2022).

in the context of posterior distributions with multiple high probability regions that are spatially separated. This is also a failure mode for MCMC algorithms such as MALA, and yet there are still many problems for which MALA has been successfully used.

The alternative choice $\Pi_1$, with $\pi_1(x) \propto p(x)^{d/(d+1)}$, which provides a generic form of over-dispersion and is optimal for approximation in 1-Wasserstein divergence (c.f. Section 2.1), was also considered. Results in Appendix D.8 indicate that, while $\Pi_1$ yields an improvement compared to the baseline of using $P$ itself, $\Pi_1$ may be less effective than our proposed $\Pi$ when $P$ is skewed.

## 5   Discussion

This paper presented Stein $\Pi$-Importance Sampling; an algorithm that is simple to implement, admits an end-to-end theoretical treatment, and achieves a significant improvement over existing post-processing methods based on KSD. On the negative side, second order derivatives of the statistical model are required, and we are ultimately bound to the performance of the KSD on which Stein $\Pi$-Importance Sampling is based. Our analysis focused on MALA, but there is in principle no barrier to deriving sufficient conditions for consistent approximation that are applicable to other sampling algorithms, such as the unadjusted Langevin algorithm. Of course, it remains to be seen whether SIS-MALA or any of its variants will stand the test of time compared to continued development in MCMC methodology, but we believe this line of research merits further investigation. For models for which access to second order derivatives is impractical, our methodology and theoretical analysis are directly applicable to gradient-free KSD (Fisher and Oates, 2023), and this would be an interesting direction for future work. Similarly, alternatives to KSD that are better-suited to high-dimensional $P$ could be considered, such as the *sliced* KSD of Gong et al. (2021a,b).

**Acknowledgements**   CW was supported by the China Scholarship Council. HK and CJO were supported by EP/W019590/1. The authors are grateful to François-Xavier Briol for feedback on an earlier draft of the manuscript, and to the anonymous Reviewers for their input.

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

# Appendices

These appendices contains supporting material for the paper *Stein $\Pi$-Importance Sampling*. The mathematical background on Stein kernels is contained in Appendix A. The proof of Theorem 1 is contained in Appendix B. For implementation of Stein $\Pi$-Importance Sampling without the aid of automatic differentiation, various explicit derivatives are required; the relevant calculations can be found in Appendix C. The empirical protocols and additional empirical results are presented in Appendix D.

## A  Mathematical Background

This appendix contains mathematical background on reproducing kernels and Stein kernels, as used in the main text. Appendix A.1 introduces matrix-valued reproducing kernels, while Appendix A.2 specialises to Stein kernels by application of a Stein operator to a matrix-valued kernel. A selection of useful Stein kernels are presented in Appendix A.3.

### A.1  Matrix-Valued Reproducing Kernels

A *matrix-valued kernel* is a function $K : \mathbb{R}^d \times \mathbb{R}^d \to \mathbb{R}^{d \times d}$, that is both

1. symmetric; $K(x, y) = K(y, x)$ for all $x, y \in \mathbb{R}^d$, and

2. positive semi-definite; $\sum_{i=1}^{n} \sum_{j=1}^{n} \langle c_i, K(x_i, x_j) c_j \rangle \geq 0$ for all $x_1, \ldots, x_n \in \mathbb{R}^d$ and $c_1, \ldots, c_n \in \mathbb{R}^d$.

Let $K_x = K(\cdot, x)$. For vector-valued functions $g, g' : \mathbb{R}^d \to \mathbb{R}^d$, defined by $g = \sum_{i=1}^{n} K_{x_i} c_i$ and $g' = \sum_{j=1}^{m} K_{x'_j} c'_i$, define an inner product

$$\langle g, g' \rangle_{\mathcal{H}(K)} = \sum_{i=1}^{n} \sum_{j=1}^{m} \langle c_i, K(x_i, x'_j) c'_j \rangle. \tag{9}$$

There is a unique Hilbert space of such vector-valued functions associated to $K$, denoted $\mathcal{H}(K)$; see Proposition 2.1 of Carmeli et al. (2006). This space is characterised as

$$\mathcal{H}(K) = \overline{\text{span}}\{K_x c : x, c \in \mathbb{R}^d\}$$

where here the closure is taken with respect to the inner product in (9). It can be shown that $\mathcal{H}(K)$ is in fact a reproducing kernel Hilbert space (RKHS) which satisfies the *reproducing property*

$$\langle g, K_x c \rangle_{\mathcal{H}(K)} = \langle g(x), c \rangle$$

for all $g \in \mathcal{H}(K)$ and $x, c \in \mathbb{R}^d$. Matrix-valued kernels are the natural starting point for construction of KSDs, as described next.

### A.2  Stein Kernels

A general construction for Stein kernels is to first identify a matrix-valued RKHS $\mathcal{H}(K)$ and an operator $S_P : \mathcal{H}(K) \to L^1(P)$ for which $\int S_p h \, dP = 0$ for all $h \in \mathcal{H}(K)$. Such an operator will be called a *Stein operator*. The collection $\{S_p h : h \in \mathcal{H}(K)\}$ inherits the structure of an RKHS, whose reproducing kernel

$$k_P(x, y) = \langle S_P K_x, S_P K_y \rangle_{\mathcal{H}(K)} \tag{10}$$

is a Stein kernel, meaning that $\mu_P = 0$ where $\mu_P$ is the kernel mean embedding from (1); see Barp et al. (2022b). Explicit calculations for the Stein kernels considered in this work can be found in Appendix C.

For univariate distributions, Barbour (1988) proposed to obtain Stein operators from infinitessimal generators of $P$-invariant continuous-time Markov processes; see also Barbour (1990); Gotze (1991).

The approach was extended to multivariate distributions in Gorham and Mackey (2015). The starting point is the $P$-invariant Itô diffusion

$$\mathrm{d}X_t = \frac{1}{2}\frac{1}{p(X_t)}\nabla \cdot [p(X_t)M(X_t)]\mathrm{d}t + M(X_t)^{1/2}\mathrm{d}W_t, \tag{11}$$

where $p$ is the density of $P$, assumed to be positive, $M : \mathbb{R}^d \to \mathbb{R}^{d \times d}$ is a symmetric matrix called the *diffusion matrix*, and $W_t$ is a standard Wiener process (Kent, 1978; Roberts and Stramer, 2002). Here the notation $[\nabla \cdot A]_i = \nabla \cdot (A_{i,.}^\top)$ indicates the divergence operator applied to each row of the matrix $A(x) \in \mathbb{R}^{d \times d}$. The infinitessimal generator is

$$(A_P u)(x) = \frac{1}{2}\frac{1}{p(x)}\nabla \cdot [p(x)M(x)\nabla u(x)].$$

Substituting $h(x)$ for $\frac{1}{2}\nabla u(x)$, we obtain a Stein operator

$$(S_P h)(x) = \frac{1}{p(x)}\nabla \cdot [p(x)M(x)h(x)] \tag{12}$$

called the *diffusion Stein operator* (Gorham et al., 2019). This is indeed a Stein operator, since under mild integrability conditions on $K$, the divergence theorem gives that $\int S_p h \, \mathrm{d}P = 0$ for all $h \in \mathcal{H}(K)$; for full details and a proof see Barp et al. (2022b).

## A.3  Selecting a Stein Kernel

There are several choices for a Stein kernel, and which we should use depends on what form of convergence we hope to control (Gorham and Mackey, 2017; Gorham et al., 2019; Hodgkinson et al., 2020; Barp et al., 2022b; Kanagawa et al., 2022). Appendix A.3.1 describes the Langevin–Stein kernel for weak convergence control, Appendix A.3.2 describes the KGM–Stein kernels for additional control over moments, and Appendix A.3.3 presents the Riemann–Stein kernel, whose convergence properties have to-date been less well-studied.

All of the kernels that we consider have length scale parameters that need to be specified, and some also have location parameters to be specified. As a reasonably automatic default we define

$$x_\star \in \arg\max p(x), \qquad \Sigma^{-1} = -\nabla^2 \log p(x_\star)$$

as a location and a matrix of characteristic length scales for $P$ that will be used throughout. These values can typically be obtained using gradient-based optimisation, which is usually cheaper to perform compared to full approximation of $P$. It is assumed that $\nabla^2 \log p(x_\star)$ is positive definite in the sequel.

### A.3.1  Weak Convergence Control with Langevin–Stein Kernels

The first kernel we consider, which we called the Langevin–Stein kernel in the main text, was introduced by Gorham and Mackey (2017). This Stein kernel was developed for the purpose of controlling the weak convergence of a sequence $(Q_n)_{n \in \mathbb{N}} \subset \mathcal{P}(\mathbb{R}^d)$ to $P$. Recall that a sequence $(Q_n)_{n \in \mathbb{N}}$ is said to *converge weakly* (or *in distribution*) to $P$ if $\int f \mathrm{d}Q_n \to \int f \mathrm{d}P$ for all continuous bounded functions $f : \mathbb{R}^d \to \mathbb{R}$. This convergence is denoted $Q_n \xrightarrow{d} P$ in shorthand.

The problem considered in Gorham and Mackey (2017) was how to select a combination of matrix-valued kernel $K$ (and, implicitly, a diffusion matrix $M$) such that the Stein kernel $k_P$ in (10) generates a KSD $D_P(Q)$ in (4) for which $D_P(Q_n) \to 0$ implies $Q_n \xrightarrow{d} P$. Their solution was to combine the inverse multi-quadric kernel with an identity diffusion matrix;

$$K(x, y) = (1 + \|x - y\|_\Sigma^2)^{-\beta} I, \qquad M(x) = I$$

for $\beta \in (0, 1)$. Provided that $P$ has a density $p$ for which $\nabla \log p(x)$ is Lipschitz, and that $P$ is *distantly dissipative* (see Definition 4 of Gorham and Mackey, 2017), the associated KSD enjoys weak convergence control. Technically, the results in Gorham and Mackey (2017) apply only when $\Sigma = I$, but Theorem 4 in Chen et al. (2019) demonstrated that they hold also for any positive definite $\Sigma$. Following the recommendation of several previous authors, including Chen et al. (2018, 2019); Riabiz et al. (2022), we take $\beta = \frac{1}{2}$ throughout.

### A.3.2 Moment Convergence Control with KGM–Stein Kernels

Despite its many elegant properties, weak convergence can be insufficient for applications where we are interested in integrals $\int f \, \mathrm{d}P$ for which the integrand $f : \mathbb{R}^d \to \mathbb{R}$ is unbounded. In particular, this is the case for moments of the form $f(x) = x_1^{\alpha_1} \ldots x_d^{\alpha_d}$, $0 \neq \alpha \in \mathbb{N}_0^d$. In such situations, we may seek also the stronger property of *moment convergence control*. The development of KSDs for moment convergence control was recently considered by Kanagawa et al. (2022), and we refered to their construction as the KGM–Stein kernels in the main text. (For convenience, we have adopted the initials of the authors in naming the KGM–Stein kernel.)

A sequence $(Q_n)_{n \in \mathbb{N}} \subset \mathcal{P}(\mathbb{R}^d)$ is said to converge to $P$ *in the sth order moment* if $\int \|x\|^s \mathrm{d}Q_n(x) \to \int \|x\|^s \mathrm{d}P(x)$. To establish convergence of moments, we need an additional condition on top of weak convergence control: uniform integrability control. A sequence of measures $(Q_n)_{n \in \mathbb{N}}$ is said to have *uniformly integrable sth moments* if for any $\varepsilon > 0$, we can take $r > 0$ such that

$$\sup_{n \in \mathbb{N}} \int_{\|x\| > r} \|x\|^s \, \mathrm{d}Q_n(x) < \varepsilon.$$

This condition essentially states that the tail decay of the measures is well-controlled (so that it has a convergent moment). The KSD convergence $D_P(Q_n) \to 0$ implies uniform integrability if for any $\varepsilon > 0$, we can take $r_\varepsilon > 0$ and $f_\varepsilon \in \mathcal{H}(K)$ such that

$$S_P f_\varepsilon(x) \geq \|x\|^s 1\{\|x\| > r_\varepsilon\} - \varepsilon, \tag{13}$$

i.e., the Stein-modified RKHS can approximate the (norm-weighted) indicator function arbitrarily well. Such a function $f_\varepsilon$ can be explicitly constructed (while not guaranteed to be a member of the RKHS). Specifically, the choice $f_\varepsilon = (1 - \iota_\varepsilon)g$ satisfies (13) under an appropriate dissipativity condition, where $\iota_\varepsilon$ is a differentiable indicator function vanishing outside a ball, and $g(x) = -x/\sqrt{1 + \|x\|^2}$. This motivated Kanagawa et al. (2022) to introduce the *sth order KGM–Stein kernel*, which is based on the matrix-valued kernel and diffusion matrix

$$K(x,y) = [\phi(\|x - y\|_\Sigma) + \kappa_{\mathrm{lin}}(x, y)] I, \qquad M(x) = (1 + \|x - x_\star\|_\Sigma^2)^{\frac{s-1}{2}} I,$$

where $(x, y) \mapsto \phi(\|x - y\|_\Sigma)$ is a $C_0^1$ universal kernel (see Barp et al., 2022b, Theorem 4.8). For comparability of our results, we take $\phi$ to be the inverse multi-quadric $\phi(r) = (1 + r^2)^{-1/2}$, and

$$\kappa_{\mathrm{lin}}(x, y) = \frac{1 + (x - x_\star)^\top \Sigma^{-1}(y - x_\star)}{\sqrt{1 + \|x - x_\star\|_\Sigma^2}\sqrt{1 + \|y - x_\star\|_\Sigma^2}}.$$

Here the normalised linear kernel $\kappa_{\mathrm{lin}}$ ensures $g \in \mathcal{H}(K)$, while the $C_0^1$ universal kernel $\phi$ allows approximation of $S_P \iota_\varepsilon g$; see Kanagawa et al. (2022).

### A.3.3 Exploiting Geometry with Riemann–Langevin–Stein Kernels

For academic interest only, here we describe the *Riemann–Stein kernel* that featured in Figure 2 of the main text. This Stein kernel is motivated by the analysis of Gorham et al. (2019), who argued that the use of rapidly mixing Itô diffusions in Stein operators can lead to sharper convergence control. The Riemann–Stein kernel is based on the class of so-called *Riemannian* diffusions considered in Girolami and Calderhead (2011), who proposed to take the diffusion matrix $M$ in (11) to be $M = (\mathcal{I}_{\mathrm{prior}} + \mathcal{I}_{\mathrm{Fisher}})^{-1}$, the inverse of the Fisher information matrix, $\mathcal{I}_{\mathrm{Fisher}}$, regularised using the Hessian of the negative log-prior, $\mathcal{I}_{\mathrm{prior}}$. For the two-dimensional illustration in Section 3.2, this leads to the diffusion matrix

$$M(x) = \left( I + \sum_{i=1}^{n} [\nabla f_i(x)][\nabla f_i(x)]^\top \right)^{-1},$$

where we recall that $y_i = f_i(x) + \epsilon_i$, where the $\epsilon_i$ are independent with $\epsilon_i \sim \mathcal{N}(0, 1)$, and the prior is $x \sim \mathcal{N}(0, 1)$. For the presented experiment we paired the above diffusion matrix with the inverse multi-quadric kernel $K(x, y) = (1 + \|x - y\|_\Sigma^2)^{-\beta}$ for $\beta = \frac{1}{2}$. The Riemann–Stein kernel extends naturally to distributions $P$ defined on Riemannian manifolds $\mathcal{X}$; see Barp et al. (2022a) and Example 1 of Hodgkinson et al. (2020).

Unfortunately, the Riemann–Stein kernel is prohibitively expensive in most real applications, since each evaluation of $M$ requires a full scan through the size-$n$ dataset. The computational complexity

of Stein $\Pi$-Thinning with the Riemann–Stein kernel is therefore $O(m^2n^2)$, which is unfavourable compared to the $O(m^2n)$ complexity in the case where the Stein kernel is not data-dependent. Furthermore, the convergence control properties of the Riemann–Stein kernel have yet to be established. For these reasons we included the Riemann–Stein kernel for illustration only; further groundwork will be required before the Riemann-Stein kernel can be practically used.

## B    Proof of Theorem 1

This appendix is devoted to the proof of Theorem 1. The proof is based on the recent work of Durmus and Moulines (2022), on the geometric convergence of MALA, and on the analysis of sparse (greedy) approximation of kernel discrepancies performed in Riabiz et al. (2022); these existing results are recalled in Appendix B.1. An additional technical result on preconditioned MALA is contained in Appendix B.2. The proof of Theorem 1 itself is contained in Appendix B.3.

### B.1    Auxiliary Results

To precisely describe the results on which our analysis is based, we first need to introduce some notation and terminology. Let $V : \mathcal{X} \to [1, \infty)$ and, for a function $f : \mathcal{X} \to \mathbb{R}$ and a measure $\mu$ on $\mathcal{X}$, let

$$\|f\|_V := \sup_{x \in \mathcal{X}} \frac{|f(x)|}{V(x)}, \qquad \|\mu\|_V := \sup_{\|f\|_V \leq 1} \left| \int_{\mathcal{X}} f \mathrm{d}\mu \right|.$$

Recall that a $Q$-invariant Markov chain $(x_i)_{i \in \mathbb{N}} \subset \mathcal{X}$ with $n^{\text{th}}$ step transition kernel $Q^n$ is $V$-*uniformly ergodic* (see Theorem 16.0.1 of Meyn and Tweedie, 2012) if and only if $\exists R \in [0, \infty), \rho \in (0, 1)$ such that

$$\|Q^n(x, \cdot) - Q\|_V \leq R\rho^n V(x) \tag{14}$$

for all initial states $x \in \mathcal{X}$ and all $n \in \mathbb{N}$.

Although MALA (Algorithm 1) is classical (Roberts and Stramer, 2002), until recently explicit sufficient conditions for ergodicity of MALA had not been obtained. The first result we will need is due Durmus and Moulines (2022), who presented the first explicit conditions for $V$-uniform convergence of MALA. It applies only to *standard* MALA, meaning that the preconditioning matrix $M$ appearing in Algorithm 1 is the identity matrix. The extension of this result to preconditioned MALA will be handled in Appendix B.2.

**Theorem 2.** *Let $Q \in \mathcal{P}(\mathbb{R}^d)$ admit a density, $q$, such that*

(DM1)  *there exists $x_0$ with $\nabla \log q(x_0) = 0$*

(DM2)  *$q$ is twice continuously differentiable with $\sup_{x \in \mathbb{R}^d} \|\nabla^2 \log q(x - x_0)\| < \infty$*

(DM3)  *there exists $b > 0$ and $B \geq 0$ such that $-\nabla^2 \log q(x - x_0) \succeq bI$ for all $\|x - x_0\| \geq B$.*

*Then there exists $\epsilon_0 > 0$ such that for all step sizes $\epsilon \in (0, \epsilon_0)$, standard $Q$-invariant MALA (i.e. with $M = I$) is $V$-uniformly ergodic for $V(x) = \exp\left(\frac{b}{16}\|x - x_0\|^2\right)$.*

*Proof.* This is Theorem 1 of Durmus and Moulines (2022). $\qquad \square$

The next result that we will need establishes consistency of the greedy algorithm applied to samples from a Markov chain that is $Q$-invariant.

**Theorem 3.** *Let $P, Q \in \mathcal{P}(\mathcal{X})$ with $P \ll Q$. Let $k_P : \mathcal{X} \times \mathcal{X} \to \mathbb{R}$ be a Stein kernel and let $D_P : \mathcal{X} \times \mathcal{X} \to [0, \infty]$ denote the associated KSD. Consider a $Q$-invariant, time-homogeneous Markov chain $(x_i)_{i \in \mathbb{N}} \subset \mathcal{X}$ such that*

(R$^+$1)  *$(x_i)_{i \in \mathbb{N}}$ is $V$-uniformly ergodic, such that $V(x) \geq \frac{\mathrm{d}P}{\mathrm{d}Q}(x)\sqrt{k_P(x)}$*

(R$^+$2)  $\sup_{i \in \mathbb{N}} \mathbb{E}\left[\frac{\mathrm{d}P}{\mathrm{d}Q}(x_i)\sqrt{k_P(x_i)}V(x_i)\right] < \infty$

(R$^+$3) *there exists $\gamma > 0$ such that $\sup_{i \in \mathbb{N}} \mathbb{E}\left[\exp\left\{\gamma \max\left(1, \frac{dP}{dQ}(x_i)^2\right) k_P(x_i)\right\}\right] < \infty$.*

*Let $P_{n,m}$ be the result of running the greedy algorithm in (5). If $m \leq n$ and $\log(n) = O(m^{\gamma/2})$ for some $\gamma < 1$, then $D_P(P_{n,m}) \to 0$ almost surely as $m, n \to \infty$.*

*Proof.* This is Theorem 3 of Riabiz et al. (2022). $\qquad\square$

## B.2 Preconditioned MALA

In addition to the auxiliary results in Appendix B.1, which concern standard MALA (i.e. with $M = I$), we require an elementary fact about MALA, namely that preconditioned MALA is equivalent to standard MALA under a linear transformation of the state variable. Recall that the $M$-preconditioned MALA algorithm is a Metropolis–Hastings algorithm whose proposal is the Euler–Maruyama discretisation of the Itô diffusion (11).

**Proposition 1.** *Let $M(x) \equiv M$ for a symmetric positive definite and position-independent matrix $M \in \mathbb{R}^{d \times d}$. Let $Q \in \mathcal{P}(\mathbb{R}^d)$ admit a probability density function (PDF) $q$ for which the $Q$-invariant diffusion $(X_t)_{t \geq 0}$, given by setting $p = q$ in (11), is well-defined. Then under the change of variables $Y_t := M^{1/2}X_t$,*

$$\mathrm{d}Y_t = \frac{1}{2}(\nabla \log \tilde{q})(Y_t)\mathrm{d}t + \mathrm{d}W_t, \tag{15}$$

*where $\tilde{q}(x) \propto q(M^{-1/2}x)$ for all $x \in \mathbb{R}^d$.*

*Proof.* From the chain rule,

$$(\nabla \log \tilde{q})(y) = \nabla_y \log q(M^{-1/2}y) = M^{-1/2}(\nabla \log q)(M^{-1/2}y),$$

and thus, substituting $Y_t = M^{1/2}X_t$, (15) is equal to

$$\mathrm{d}X_t = M^{-1/2}\left[\frac{1}{2}M^{-1/2}(\nabla \log q)(M^{-1/2}M^{1/2}X_t) + \mathrm{d}W_t\right]$$
$$= \frac{1}{2}M^{-1}(\nabla \log q)(X_t) + M^{-1/2}\mathrm{d}W_t,$$

which is identical to (11) in the case where $M(x) = M$ is constant. $\qquad\square$

Let $Q$ and $\tilde{Q}$ be the distributions referred to in Proposition 1, whose PDFs are respectively $q(x)$ and $\tilde{q}(x) \propto q(M^{-1/2}x)$. Proposition 1 then implies that the $M$-preconditioned MALA algorithm applied to $Q$ (i.e. Algorithm 1 for $\Pi = Q$) is equivalent to the standard MALA algorithm (i.e. $M = I$) applied to $\tilde{Q}$. This fact allows us to generalise the result of Theorem 2 as follows:

**Corollary 1.** *Consider a symmetric positive definite matrix $M \in \mathbb{R}^{d \times d}$. Assume that conditions (DM1-3) in Theorem 2 are satisfied. Then there exists $\epsilon_0' > 0$ and $b' > 0$ such that for all step sizes $\epsilon \in (0, \epsilon_0')$, the $M$-preconditioned $Q$-invariant MALA is $V$-uniformly ergodic for $V(x) = \exp\left(\frac{b'}{16}\|x - x_0\|^2\right)$.*

*Proof.* From Theorem 2 and Proposition 1, the result follows if we can establish (DM1-3) for $\tilde{Q}$, since $M$-preconditioned MALA is equivalent to standard MALA applied to $\tilde{Q}$. For a matrix $A \in \mathbb{R}^{d \times d}$, let $\lambda_{\min}(A)$ and $\lambda_{\max}(A)$ respectively denote the minimum and maximum eigenvalues of $A$. For (DM1) we set $y_0 = M^{1/2}x_0$ and observe that

$$(\nabla \log \tilde{q})(y_0) = M^{-1/2}(\nabla \log q)(x_0) = 0.$$

For (DM2) we have that

$$\sup_{y \in \mathbb{R}^d} \|\nabla^2(\log \tilde{q})(y - y_0)\| = \sup_{y \in \mathbb{R}^d} \|M^{-1/2}(\nabla^2 \log q)(M^{-1/2}(y - y_0))M^{-1/2}\|$$
$$\leq \lambda_{\min}(M)^{-1} \sup_{x \in \mathbb{R}^d} \|(\nabla^2 \log q)(x - x_0)\| < \infty.$$

For (DM3) we have that

$$-(\nabla^2 \log \tilde{q})(y - y_0) = -M^{-1/2}(\nabla^2 \log q)(M^{-1/2}(y - y_0))M^{-1/2}$$

$$= -M^{-1/2}(\nabla^2 \log q)(x - x_0)M^{-1/2} \succeq M^{1/2}(bI)M^{1/2} = bM^{-1} \succeq b'I$$

where $b' = b\lambda_{\max}(M)^{-1}$, which holds for all $\|x - x_0\| \geq B$, and in particular for all $\|y - y_0\| \geq B'$ where $B' = B\lambda_{\max}(M)^{1/2}$. Thus (DM1-3) are established for $\tilde{Q}$. $\qquad\square$

**Remark 1.** *The choice $M = \Sigma^{-1}$, which sets the preconditioner matrix $M$ equal to the inverse of the length scale matrix $\Sigma$ used in the specification of the kernel $K$ (c.f. Appendix A.3), leads to the elegant interpretation that Stein $\Pi$-Importance Sampling applied to $M$-preconditioned MALA is equivalent to the Stein $\Pi$-Importance Sampling applied to standard MALA (i.e. with $M = I$) for the whitened target $\tilde{P}$ with PDF $\tilde{p}(x) \propto p(M^{-1/2}x)$. For our experiments, however, the preconditioner matrix $M$ was learned during a warm-up phase of MALA, since in general the curvature of $P$ (captured by $\Sigma$) and the curvature of $\Pi$ (captured by $M^{-1}$) may be different.*

### B.3   Proof of Theorem 1

The route to establishing Theorem 1 has three parts. First, we establish (DM1-3) of Theorem 2 with $Q = \Pi$, to deduce from Corollary 1 that $\Pi$-invariant $M$-preconditioned MALA is $V$-uniformly ergodic. This in turn enables us to establish conditions (R$^+$1-3) of Theorem 3, again for $Q = \Pi$, from which the strong consistency $D_P(P_{n,m}) \overset{\text{a.s.}}{\to} 0$ of SΠT-MALA is established. Finally, we note that $0 \leq D_P(P_n^\star) \leq D_P(P_{n,m})$, since the support of $P_{n,m}$ is contained in the support of $P_n^\star$, and the latter is optimally weighted, whence also the strong consistency of SΠIS-MALA.

**Establish (DM1-3)**   First we establish (DM1-3) for $Q = \Pi$. Fix $x_0 \in \mathbb{R}^d$. For (DM2), first recall that the range of $k_P$ is $[C_1^2, \infty)$ where $C_1 > 0$, from Assumption 1. Since $\log(\cdot)$ has bounded second derivatives on $[C_1^2, \infty)$, there is a constant $C > 0$ such that

$$\forall x \in \mathbb{R}^d, \qquad \|\nabla^2 \log k_P(x)\| \leq C\|\nabla^2 k_P(x)\|.$$

Thus, using compactness of the set $\{x : \|x - x_0\| \leq B_2\}$,

$$\sup_{x \in \mathbb{R}^d} \|\nabla^2 \log k_P(x)\| \leq C \max \left( \underbrace{\sup_{\|x - x_0\| \leq B_2} \|\nabla^2 k_P(x)\|}_{<\infty \text{ by (A3)}}, \underbrace{\sup_{\|x - x_0\| \geq B_2} \|\nabla^2 k_P(x)\|}_{<b_2\|I\| \text{ by (A4)}} \right) < \infty. \quad (16)$$

Now, $\pi$ is twice differentiable as it is the product of twice differentiable functions $p$ and $k_P^{1/2}$ from (A1) and (A3), and moreover

$$\sup_{x \in \mathbb{R}^d} \|\nabla^2 \log \pi(x - x_0)\| \leq \underbrace{\sup_{x \in \mathbb{R}^d} \|\nabla^2 \log p(x)\|}_{<\infty \text{ by (A1)}} + \frac{1}{2} \underbrace{\sup_{x \in \mathbb{R}^d} \|\nabla^2 \log k_P(x)\|}_{<\infty \text{ by (16)}} < \infty,$$

so (DM2) is satisfied. For (DM3), first note from the chain and product rules that for all $\|x\| \geq B_2$

$$\nabla^2 \log k_P(x - x_0) = \underbrace{\frac{\nabla^2 k_P(x - x_0)}{k_P(x - x_0)}}_{\preceq (b_2/C_1^2)I \text{ by (A4)}} - \underbrace{\frac{[\nabla k_P(x - x_0)][\nabla k_P(x - x_0)]^\top}{k_P(x - x_0)^2}}_{\succeq 0} \preceq \frac{b_2}{C_1^2}I. \quad (17)$$

Thus, for all $\|x - x_0\| \geq B := \|x_0\| + \max(B_1, B_2)$,

$$-\nabla^2 \log \pi(x - x_0) = \underbrace{-\nabla^2 \log p(x - x_0)}_{\succeq b_1 I \text{ by (A2)}} - \frac{1}{2} \underbrace{\nabla^2 \log k_P(x - x_0)}_{\preceq (b_2/C_1^2)I \text{ by (17)}} \succeq \underbrace{\left( b_1 - \frac{b_2}{2C_1^2} \right)}_{=:b>0} I \quad (18)$$

as required. The same argument establishes (DM1); from (18) we have $\lim_{\|x\| \to \infty} \pi(x) = 0$, and since $\pi$ is a continuously differentiable density there must exist an $x_0$ at which $\pi$ is locally minimised. Thus we have established (DM1-3) for $Q = \Pi$ and we may conclude from Corollary 1 that there is an $\epsilon_0' > 0$ and $b' > 0$ such that, for all $\epsilon \in (0, \epsilon_0')$, the $\Pi$-invariant $M$-preconditioned MALA chain $(x_i)_{i \in \mathbb{N}}$ is $V$-uniformly ergodic for $V(x) = C_2 \exp\left( \frac{b'}{16}\|x - x_0\|^2 \right)$ (since if a Markov chain is $V$-uniformly ergodic, then it is also $CV$-uniformly ergodic).

**Establish (R$^+$1-3)** The aim is now to establish conditions (R$^+$1-3) of Theorem 3 for $Q = \Pi$. By construction $\mathrm{d}P/\mathrm{d}\Pi = C_2/\sqrt{k_P(x)} < C_2/C_1 < \infty$, where $C_1$ and $C_2$ were defined in Assumption 1, so that $P \ll \Pi$. It has already been established that $(x_i)_{i \in \mathbb{N}}$ is $V$-uniformly ergodic, and further

$$V(x) = C_2 \exp\left(\frac{b'}{16}\|x - x_0\|^2\right) \geq C_2 = \frac{\mathrm{d}P}{\mathrm{d}\Pi}(x)\sqrt{k_P(x)}$$

for all $x$, which establishes (R$^+$1). Let $R$ and $\rho$ denote constants for which the $V$-uniform ergodicity property (14) is satisfied. From $V$-uniform ergodicity, the integral $\int V \, \mathrm{d}\Pi$ exists and

$$\left|\mathbb{E}\left[\frac{\mathrm{d}P}{\mathrm{d}\Pi}(x_i)\sqrt{k_P(x_i)}V(x_i)\right] - C_2\int V \, \mathrm{d}\Pi\right| = C_2\left|\mathbb{E}[V(x_i)] - \int V \, \mathrm{d}\Pi\right|$$
$$\leq C_2 R\rho^n V(x_0) \to 0$$

which establishes (R$^+$2). Fix $\gamma > 0$. By construction $\mathrm{d}P/\mathrm{d}\Pi \leq C_2/C_1$, and thus

$$\exp\left\{\gamma \max\left(1, \frac{\mathrm{d}P}{\mathrm{d}\Pi}(x)^2\right)k_P(x)\right\} < \exp\{\tilde{\gamma}k_P(x)\}$$

where $\tilde{\gamma} = \max(1, C_2/C_1)\gamma$. Since we have assumed that $k_P$ is continuous with, from (A4),

$$b_3 := \limsup_{\|x\| \to \infty} \frac{k_P(x)}{\|x\|^2} < \infty,$$

we may take $\gamma$ such that $\tilde{\gamma}b_3 < b'/16$, so that $\|x \mapsto \exp\{\tilde{\gamma}k_P(x)\}\|_V < \infty$ and in particular

$$\left|\mathbb{E}\left[\exp\{\tilde{\gamma}k_P(x_i)\}\right] - \int \exp\{\tilde{\gamma}k_P(x)\} \, \mathrm{d}\Pi(x)\right| \leq \|x \mapsto \exp\{\tilde{\gamma}k_P(x)\}\|_V \times R\rho^n V(x_0) \to 0$$

which establishes (R$^+$3). Thus we have established (R$^+$1-3) for $Q = \Pi$, so from Theorem 3 we have strong consistency of SΠT-MALA (i.e. $D_P(P_{n,m}) \overset{\text{a.s.}}{\to} 0$) provided that $m \leq n$ with $\log(n) = O(m^{\gamma/2})$ for some $\gamma < 1$. The latter condition is equivalent to $m = \Omega((\log n)^\delta)$ for some $\delta > 2$, which we used for the statement. Since $0 \leq D_P(P_n^\star) \leq D_P(P_{n,m})$, the strong consistency of SΠIS-MALA is also established.

## C  Explicit Calculation of Stein Kernels

This appendix contains explicit calculations for the Langevin–Stein and KGM–Stein kernels $k_P$, which are sufficient to implement Stein Π-Importance Sampling and Stein Π-Thinning. These calculations can also be performed using automatic differentiation, but comparison to the analytic expressions is an important step in validation of computer code.

To proceed, we observe that the diffusion Stein operator $S_P$ in (12) applied to a matrix-valued kernel $K$ is equivalent to the Langevin–Stein operator applied to the kernel $C(x,y) = M(x)K(x,y)M(y)^\top$. In the case of the Langevin–Stein and KGM–Stein kernels we have $K(x,y) = \kappa(x,y)I$ for some $\kappa(x,y)$ and $M(x) = (1 + \|x - x_\star\|_\Sigma^2)^{(s-1)/2}I$ for some $s \in \{0, 1, 2, \dots\}$. Thus $C(x,y) = c(x,y)I$ where

$$c(x,y) := (1 + \|x - x_\star\|_\Sigma^2)^{(s-1)/2}(1 + \|y - x_\star\|_\Sigma^2)^{(s-1)/2}\kappa(x,y)$$

and

$$k_P(x,y) = \nabla_x \cdot \nabla_y c(x,y) + [\nabla_x c(x,y)] \cdot [\nabla_y \log p(y)] + [\nabla_y c(x,y)] \cdot [\nabla_x \log p(x)]$$
$$+ c(x,y)[\nabla_x \log p(x)] \cdot [\nabla_y \log p(y)],$$

following the calculations in Oates et al. (2017). To evaluate the terms in this formula we start by differentiating $c(x, y)$, to obtain

$$\nabla_x c(x, y) = (1 + \|x - x_\star\|_\Sigma^2)^{(s-1)/2}(1 + \|y - x_\star\|_\Sigma^2)^{(s-1)/2}$$
$$\times \left[ \frac{(s-1)\kappa(x,y)\Sigma^{-1}(x - x_\star)}{1 + \|x - x_\star\|_\Sigma^2} + \nabla_x \kappa(x, y) \right]$$

$$\nabla_y c(x, y) = (1 + \|x - x_\star\|_\Sigma^2)^{(s-1)/2}(1 + \|y - x_\star\|_\Sigma^2)^{(s-1)/2}$$
$$\times \left[ \frac{(s-1)\kappa(x,y)\Sigma^{-1}(y - x_\star)}{1 + \|y - x_\star\|_\Sigma^2} + \nabla_y \kappa(x, y) \right]$$

$$\nabla_x \cdot \nabla_y c(x, y) = (1 + \|x - x_\star\|_\Sigma^2)^{(s-1)/2}(1 + \|y - x_\star\|_\Sigma^2)^{(s-1)/2}$$
$$\times \left[ \frac{(s-1)^2 \kappa(x,y)(x - x_\star)^\top \Sigma^{-2}(y - x_\star)}{(1 + \|x - x_\star\|_\Sigma^2)(1 + \|y - x_\star\|_\Sigma^2)} + \frac{(s-1)(y - x_\star)^\top \Sigma^{-1} \nabla_x \kappa(x, y)}{(1 + \|y - x_\star\|_\Sigma^2)} \right.$$
$$\left. + \frac{(s-1)(x - x_\star)^\top \Sigma^{-1} \nabla_y \kappa(x, y)}{(1 + \|x - x_\star\|_\Sigma^2)} + \nabla_x \cdot \nabla_y \kappa(x, y) \right].$$

These expressions involve gradients of $\kappa(x, y)$, and explicit formulae for these are presented for the choice of $\kappa(x, y)$ corresponding to the Langevin–Stein kernel in Appendix C.1, and to the KGM–Stein kernel in Appendix C.2.

To implement Stein $\Pi$-Thinning we require access to both $k_P(x)$ and $\nabla k_P(x)$, the latter for use in the proposal distribution and acceptance probability in MALA. These quantities will now be calculated. In what follows we assume that $\kappa(x, y)$ is continuously differentiable, so that partial derivatives with respect to $x$ and $y$ can be interchanged. Then

$$c_0(x) := c(x, x)$$
$$= (1 + \|x - x_\star\|_\Sigma^2)^{s-1} \kappa(x, x)$$
$$c_1(x) := \nabla_x c(x, y)|_{y \to x}$$
$$= (1 + \|x - x_\star\|_\Sigma^2)^{s-1} \left[ \frac{(s-1)\kappa(x,x)\Sigma^{-1}(x - x_\star)}{(1 + \|x - x_\star\|_\Sigma^2)} + \nabla_x \kappa(x, y)|_{y \to x} \right]$$
$$c_2(x) := \nabla_x \cdot \nabla_y c(x, y)|_{y \to x}$$
$$= (1 + \|x - x_\star\|_\Sigma^2)^{s-1} \left[ \frac{(s-1)^2 \kappa(x,x)(x - x_\star)^\top \Sigma^{-2}(x - x_\star)}{(1 + \|x - x_\star\|_\Sigma^2)^2} \right.$$
$$\left. + \frac{2(s-1)(x - x_\star)^\top \Sigma^{-1} \nabla_x \kappa(x, y)|_{y \to x}}{(1 + \|x - x_\star\|_\Sigma^2)} + \nabla_x \cdot \nabla_y \kappa(x, y)|_{y \to x} \right]$$

so that

$$k_P(x) := k_P(x, x) = c_2(x) + 2c_1(x) \cdot \nabla_x \log p(x) + c_0(x)\|\nabla_x \log p(x)\|^2. \qquad (19)$$

Let $[\nabla_x c_1(x)]_{i,j} = \partial_{x_i}[c_1(x)]_j$ and $[\nabla_x^2 \log p(x)]_{i,j} = \partial_{x_i}\partial_{x_j} \log p(x)$. Now we can differentiate (19) to get

$$\nabla_x k_P(x) = \nabla_x c_2(x) + 2[\nabla_x c_1(x)][\nabla_x \log p(x)] + 2[\nabla_x^2 \log p(x)]c_1(x)$$
$$+ [\nabla_x c_0(x)]\|\nabla_x \log p(x)\|^2 + 2c_0(x)[\nabla_x^2 \log p(x)][\nabla_x \log p(x)]. \qquad (20)$$

In what follows we also derive explicit formulae for $c_0(x)$, $c_1(x)$ and $c_2(x)$, and hence for $\nabla_x c_0(x)$, $\nabla_x c_1(x)$ and $\nabla_x c_2(x)$, for the case of the Langevin–Stein kernel in Appendix C.1, and the KGM–Stein kernel in Appendix C.2.

## C.1 Explicit Formulae for the Langevin–Stein Kernel

The Langevin–Stein kernel from Appendix A.3.1 corresponds to the choice $s = 1$ and $\kappa(x, y)$ the inverse multi-quadric kernel, so that

$$\kappa(x, y) = (1 + \|x - y\|_\Sigma^2)^{-\beta}$$
$$\nabla_x \kappa(x, y) = -2\beta(1 + \|x - y\|_\Sigma^2)^{-\beta-1}\Sigma^{-1}(x - y)$$
$$\nabla_y \kappa(x, y) = 2\beta(1 + \|x - y\|_\Sigma^2)^{-\beta-1}\Sigma^{-1}(x - y)$$
$$\nabla_x \cdot \nabla_y \kappa(x, y) = -4\beta(\beta + 1)(1 + \|x - y\|_\Sigma^2)^{-\beta-2}(x - y)^\top \Sigma^{-2}(x - y)$$
$$+ 2\beta\,\mathrm{tr}(\Sigma^{-1})(1 + \|x - y\|_\Sigma^2)^{-\beta-1}.$$

Evaluating on the diagonal:

$$\kappa(x, x) = 1$$
$$\nabla_x \kappa(x, y)|_{y \to x} = \nabla_y \kappa(x, y)|_{y \to x} = 0$$
$$\nabla_x \cdot \nabla_y \kappa(x, y)|_{y \to x} = 2\beta\,\mathrm{tr}(\Sigma^{-1}),$$

so that $c_0(x) = 1$, $c_1(x) = 0$, $c_2(x) = 2\beta\,\mathrm{tr}(\Sigma^{-1})$. Differentiating these formulae, $\nabla_x c_0(x) = 0$, $\nabla_x c_1(x) = 0$, $\nabla_x c_2(x) = 0$.

## C.2 Explicit Formulae for the KGM–Stein Kernel

The KGM kernel of order $s$ from Appendix A.3.2 corresponds to the choice

$$\kappa(x, y) = (1 + \|x - y\|_\Sigma^2)^{-\beta} + \frac{1 + (x - x_\star)^\top \Sigma^{-1}(y - x_\star)}{(1 + \|x - x_\star\|_\Sigma^2)^{s/2}(1 + \|y - x_\star\|_\Sigma^2)^{s/2}},$$

for which we have

$$\nabla_x \kappa(x, y) = -2\beta(1 + \|x - y\|_\Sigma^2)^{-\beta-1}\Sigma^{-1}(x - y)$$
$$+ \frac{\Sigma^{-1}(y - x_\star) - s[1 + (x - x_\star)^\top \Sigma^{-1}(y - x_\star)]\Sigma^{-1}(x - x_\star)(1 + \|x - x_\star\|_\Sigma^2)^{-1}}{(1 + \|x - x_\star\|_\Sigma^2)^{s/2}(1 + \|y - x_\star\|_\Sigma^2)^{s/2}}$$

$$\nabla_y \kappa(x, y) = 2\beta(1 + \|x - y\|_\Sigma^2)^{-\beta-1}\Sigma^{-1}(x - y)$$
$$+ \frac{\Sigma^{-1}(x - x_\star) - s[1 + (x - x_\star)^\top \Sigma^{-1}(y - x_\star)]\Sigma^{-1}(y - x_\star)(1 + \|y - x_\star\|_\Sigma^2)^{-1}}{(1 + \|x - x_\star\|_\Sigma^2)^{s/2}(1 + \|y - x_\star\|_\Sigma^2)^{s/2}}$$

$$\nabla_x \cdot \nabla_y \kappa(x, y) = -4\beta(\beta + 1)(1 + \|x - y\|_\Sigma^2)^{-\beta-2}(x - y)^\top \Sigma^{-2}(x - y) + 2\beta\,\mathrm{tr}(\Sigma^{-1})(1 + \|x - y\|_\Sigma^2)^{-\beta-1}$$
$$+ \frac{\begin{bmatrix} \mathrm{tr}(\Sigma^{-1}) - s(1 + \|x - x_\star\|_\Sigma^2)^{-1}(x - x_\star)^\top \Sigma^{-2}(x - x_\star) \\ -s(1 + \|y - x_\star\|_\Sigma^2)^{-1}(y - x_\star)^\top \Sigma^{-2}(y - x_\star) \\ +s^2[1 + (x - x_\star)^\top \Sigma^{-1}(y - x_\star)](1 + \|x - x_\star\|_\Sigma^2)^{-1}(1 + \|y - x_\star\|_\Sigma^2)^{-1} \\ \times (x - x_\star)^\top \Sigma^{-2}(y - x_\star) \end{bmatrix}}{(1 + \|x - x_\star\|_\Sigma^2)^{s/2}(1 + \|y - x_\star\|_\Sigma^2)^{s/2}}.$$

Evaluating on the diagonal:

$$\kappa(x, x) = 1 + (1 + \|x - x_\star\|_\Sigma^2)^{-s+1}$$
$$\nabla_x \kappa(x, y)|_{y \to x} = \nabla_y \kappa(x, y)|_{y \to x} = -(s - 1)\Sigma^{-1}(x - x_\star)(1 + \|x - x_\star\|_\Sigma^2)^{-s}$$
$$\nabla_x \cdot \nabla_y \kappa(x, y)|_{y \to x} = 2\beta\,\mathrm{tr}(\Sigma^{-1}) + \mathrm{tr}(\Sigma^{-1})(1 + \|x - x_\star\|_\Sigma^2)^{-s}$$
$$+ s(s - 2)(1 + \|x - x_\star\|_\Sigma^2)^{-s-1}(x - x_\star)^\top \Sigma^{-2}(x - x_\star)$$

so that

$$c_0(x) = 1 + (1 + \|x - x_\star\|_\Sigma^2)^{s-1}$$
$$c_1(x) = (s - 1)(1 + \|x - x_\star\|_\Sigma^2)^{s-2}\Sigma^{-1}(x - x_\star)$$
$$c_2(x) = \frac{[(s - 1)^2(1 + \|x - x_\star\|_\Sigma^2)^{s-1} - 1](x - x_\star)^\top \Sigma^{-2}(x - x_\star)}{(1 + \|x - x_\star\|_\Sigma^2)^2} + \frac{\mathrm{tr}(\Sigma^{-1})[1 + 2\beta(1 + \|x - x_\star\|_\Sigma^2)^s]}{(1 + \|x - x_\star\|_\Sigma^2)}.$$

Differentiating these formulae:

$$\nabla_x c_0(x) = 2(s-1)(1 + \|x - x_\star\|_\Sigma^2)^{s-2}\Sigma^{-1}(x - x_\star)$$

$$\nabla_x c_1(x) = 2(s-1)(s-2)(1 + \|x - x_\star\|_\Sigma^2)^{s-3}[\Sigma^{-1}(x - x_\star)][\Sigma^{-1}(x - x_\star)]^\top$$
$$\qquad + (s-1)(1 + \|x - x_\star\|_\Sigma^2)^{s-2}\Sigma^{-1}$$

$$\nabla_x c_2(x) = 2(s-1)^2(s-3)(1 + \|x - x_\star\|_\Sigma^2)^{s-4}[(x - x_\star)^\top\Sigma^{-2}(x - x_\star)]\Sigma^{-1}(x - x_\star)$$
$$\qquad + 2(s-1)^2(1 + \|x - x_\star\|_\Sigma^2)^{s-3}\Sigma^{-2}(x - x_\star)$$
$$\qquad + 4\beta\mathrm{tr}(\Sigma^{-1})(s-1)(1 + \|x - x_\star\|_\Sigma^2)^{s-2}\Sigma^{-1}(x - x_\star)$$
$$\qquad - 2(1 + \|x - x_\star\|_\Sigma^2)^{-2}[\Sigma^{-2}(x - x_\star) + \mathrm{tr}(\Sigma^{-1})\Sigma^{-1}(x - x_\star)]$$
$$\qquad + 4(1 + \|x - x_\star\|_\Sigma^2)^{-3}[(x - x_\star)^\top\Sigma^{-2}(x - x_\star)]\Sigma^{-1}(x - x_\star).$$

These complete the analytic calculations necessary to compute the Stein kernel $k_P$ and its gradient.

## D  Empirical Assessment

This appendix contains full details of the empirical protocols that were employed and the additional empirical results described in the main text. Appendix D.1 discusses the effect of dimension on our proposed $\Pi$. Additional illuatrative results from Section 3.2 are contained in Appendix D.2. The full details for how MALA was implemented are contained in Appendix D.3. An additional illustration using a generalised auto-regressive moving average (GARCH) model is presented in Appendix D.4. The full results for S$\Pi$IS-MALA are contained in Appendix D.5, and in Appendix D.6 the convergence of the sparse approximation provided by S$\Pi$T-MALA to the optimal weighted approximation is investigated. Finally, the performance of KSDs is quantified using the 1-Wasserstein divergence in Appendix D.7.

### D.1  The Effect of Dimension on $\Pi$

The improvement of Stein $\Pi$-Importance Sampling over the default Stein importance sampling algorithm (i.e. $\Pi = P$) can be expected to reduce as the dimension $d$ of the target $P$ is increased. To see this, consider the Langevin–Stein kernel

$$k_P(x) = c_1 + c_2\|\nabla \log p(x)\|_\Sigma^2 \tag{21}$$

for some $c_1, c_2 > 0$; see Appendix C. Taking $P = \mathcal{N}(0, I_{d \times d})$, for which the length scale matrix $\Sigma$ appearing in Appendix A.3 is $\Sigma = I_{d \times d}$, we obtain

$$k_P(x) = c_1 + c_2\|x\|^2.$$

However, the sampling distribution $\Pi$ defined in (8) depends on $k_P$ only up to an unspecified normalisation constant; we may therefore equally consider the asymptotic behaviour of $\tilde{k}_P(x) := k_P(x)/d$. Let $X \sim P$. Then $\mathbb{E}[\tilde{k}_P(X)] = c_2$ is a $d$-independent constant, and

$$\left\|\tilde{k}_P - \mathbb{E}[\tilde{k}_P(X)]\right\|_{L^2(P)}^2 = \int \left[\frac{k_P(x) - (c_1 + c_2 d)}{d}\right]^2 \mathrm{d}P(x) = \frac{2c_2^2}{d} \to 0$$

as $d \to \infty$. This shows that $\tilde{k}_P$ converges to a constant function in $L^2(P)$, and thus for "typical" values of $x$ in the effective support of $P$,

$$\pi(x) \propto p(x)\sqrt{\tilde{k}_P(x)} \overset{\approx}{\propto} p(x),$$

so that $\Pi \approx P$ in the $d \to \infty$ limit. This intuition is borne out in simulations involving both the Langevin–Stein kernel (as just discussed) and also the KGM3–Stein kernel. Indeed, Figure S1 shows that as the dimension $d$ is increased, the marginal distributions of $\Pi$ become increasingly similar to those of $P$.

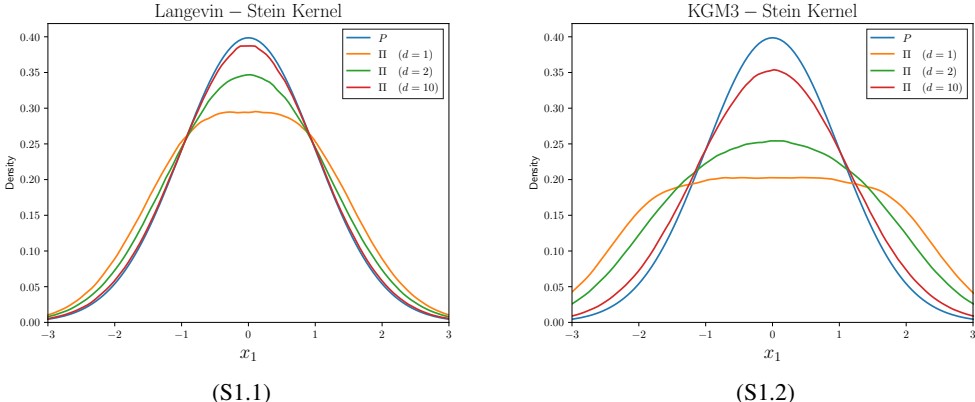

(S1.1)                (S1.2)

Figure S1: The effect of dimension on $\Pi$: Here $P$ was taken to be the standard Gaussian distribution $\mathcal{N}(0, I_{d \times d})$ in $\mathbb{R}^d$ and the proposed distribution $\Pi$ was computed. The marginal distribution of the first component of $\Pi$ is plotted for $d \in \{1, 2, 10\}$, for both (a) the Langevin–Stein kernel and (b) the KGM3–Stein kernel.

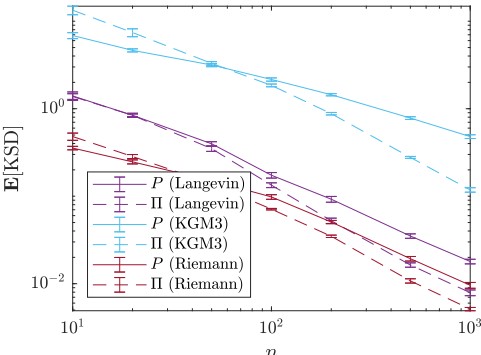

Figure S2: Assessing the performance of the sampling distributions $\Pi$ shown in Figure 2. The mean kernel Stein discrepancy (KSD) for computation performed using the Langevin–Stein kernel (purple), the KGM3–Stein kernel (blue), and the Riemann–Stein kernel (red); in each case, KSD was computed using the same Stein kernel used to construct $\Pi$. Solid lines indicate the baseline case of sampling from $P$, while dashed lines indicate the proposed approach of sampling from $\Pi$. (The experiment was repeated 10 times and standard error bars are plotted.)

## D.2 2D Illustration from the Main Text

Section 3.2 of the main text contained a 2-dimensional illustration of Stein $\Pi$-Importance Sampling and presented the distributions $\Pi$ corresponding to different choices of Stein kernel. Here, in Figure S2, we present the mean KSDs for Stein $\Pi$-Importance Sampling performed using the Langevin–Stein kernel (purple), the KGM3–Stein kernel (blue), and the Riemann–Stein kernel (red), corresponding to the sampling distributions $\Pi$ displayed in Figure 2 of the main text.

For this experiment, exact sampling from both $P$ and $\Pi$ was performed using a fine grid on which all probabilities were calculated and appropriately normalised. Results are in broad agreement with the 1-dimensional illustration contained in the main text, in the sense that in all cases Stein $\Pi$-Importance Sampling provides a significant improvement over the default Stein importance sampling method with $\Pi$ equal to $P$.

---

**Algorithm 4** Adaptive MALA

---

**Require:** $x_{0,0}$ (initial state), $\epsilon_0$ (initial step size), $M_0$ (initial preconditioner matrix), $\{n_i\}_{i=0}^{h-1}$ (epoch lengths), $\{\alpha_i\}_{i=1}^{h-1}$ (learning schedule), $h$ (number of epochs), $k_P$ (Stein kernel)

1: $\{x_{0,1}\ldots,x_{0,n_0}\} \leftarrow \texttt{MALA}(x_{0,0},\epsilon_0,M_0,n_0,k_P)$
2: **for** $i = 1,\ldots,h-1$ **do**
3:     $x_{i,0} \leftarrow x_{i-1,n_{i-1}}$
4:     $\rho_{i-1} \leftarrow \frac{1}{n_{i-1}}\sum_{j=1}^{n_{i-1}} 1_{x_{i-1,j}\neq x_{i-1,j-1}}$                     ▷ Average acceptance rate for chain $i$
5:     $\epsilon_i \leftarrow \epsilon_{i-1}\exp(\rho_{i-1}-0.57)$                                      ▷ Update step size
6:     $M_i \leftarrow \alpha_i M_i + (1-\alpha_i)\mathrm{cov}(\{x_{i-1,1}\ldots,x_{i-1,n_{i-1}}\})$        ▷ Update preconditioner matrix
7:     $\{x_{i,1}\ldots,x_{i,n_i}\} \leftarrow \texttt{MALA}(x_{i,0},\epsilon_i,M_i,n_i,k_P)$
8: **end for**

---

## D.3 Implementation of MALA

For implementation of MALA in Algorithm 4 we are required to specify a step size $\epsilon$ and a preconditioner matrix $M$. In general, suitable values for both of these parameters will be problem-dependent. Standard practice is to perform some form of manual or automated tuning to arrive at parameter values for which the average acceptance rate is close to 0.57, motivated by the asymptotic analysis of Roberts and Rosenthal (1998). Adaptive MCMC algorithms, which seek to optimise the parameters of MCMC algorithms such as MALA during the warm-up period, provide an appealing solution, and was the approach taken in this work.

The adaptive MALA algorithm which we used is contained in Algorithm 4, where we have let $\texttt{MALA}(x,\epsilon,M,n,k_P)$ denote the output from the preconditioned MALA with initial state $x$, step size $\epsilon$, preconditioner matrix $M$, and chain length $n$, described in Algorithm 1. In Algorithm 4, we use $\mathrm{cov}(\cdot)$ to denote the sample covariance matrix. The algorithm monitors the average acceptance rate and increases or decreases it according to whether it is below or above, respectively, the 0.57 target. For the preconditioner matrix, the sample covariance matrix of samples obtained from the penultimate tuning run of MALA is used. For all experiments that we report using MALA, we set $\epsilon_0 = 1$, $M_0 = I_d$, $h = 10$, and $\alpha_1 = \cdots = \alpha_9 = 0.3$. The warm-up epoch lengths were $n_0 = \cdots = n_8 = 1,000$ and the final epoch length was $n_9 = 10^5$. The samples $\{x_{h-1,1},\ldots,x_{h-1,n_{i-1}}\}$ from the final epoch are returned, and constituted output from MALA for our experimental assessment.

To sample from $P$ instead of $\Pi$, we used Algorithm 4 we formally set $k_P(x) = 1$ for all $x \in \mathbb{R}^d$, which recovers $\Pi = P$ as the target.

## D.4 Illustration on a GARCH Model

This appendix contains an additional illustrative experiment, concerning a GARCH model that is a particular instance of a model from the `PosteriorDB` database discussed in Section 4. The purpose of this illustration is to facilitate an empirical investigation in a slightly higher dimension ($d = 4$) and to explore the effect of changing the order $s$ of the KGM–Stein kernel defined in Appendix A.3.2.

First we describe the GARCH model that was used. These models are widely-used in econometrics to describe time series data $\{y_t\}_{t=1}^n$ in settings where the volatility process is assumed to be time-varying (but stationary). In particular, we consider the GARCH(1,1) model

$$
\begin{aligned}
y_t &= \phi_1 + a_t, \\
a_t &= \sigma_t \epsilon_t, \quad \epsilon_t \sim \mathcal{N}(0,1), \\
\sigma_t^2 &= \phi_2 + \phi_3 a_{t-1}^2 + \phi_4 \sigma_{t-1}^2,
\end{aligned}
$$

where $\phi_2 > 0$, $\phi_3 > 0$, $\phi_4 > 0$, and $\phi_3 + \phi_4 < 1$ are the model parameters, constrained to a subset of $\mathbb{R}^4$. For ease of sampling, a change of variables $\tau : (\phi_1,\phi_2,\phi_3,\phi_4) \mapsto \theta$ is performed in such a way that the parameter $\theta \in \mathbb{R}^4$ is unconstrained. Assuming an improper flat prior on $\theta$, the log-posterior density for $\theta$ is given up to an additive constant by

$$
\log p(\theta \mid y_1,\ldots,y_n) \overset{+C}{=} \sum_{t=1}^n \left[ -\frac{1}{2}\log(\sigma_t^2) - \frac{y_t^2}{2\sigma_t^2} \right] + \log|J_{\tau^{-1}}(\theta)|,
$$

where $|J_{\tau^{-1}}(\theta)|$ is the Jacobian determinant of $\tau^{-1}$.

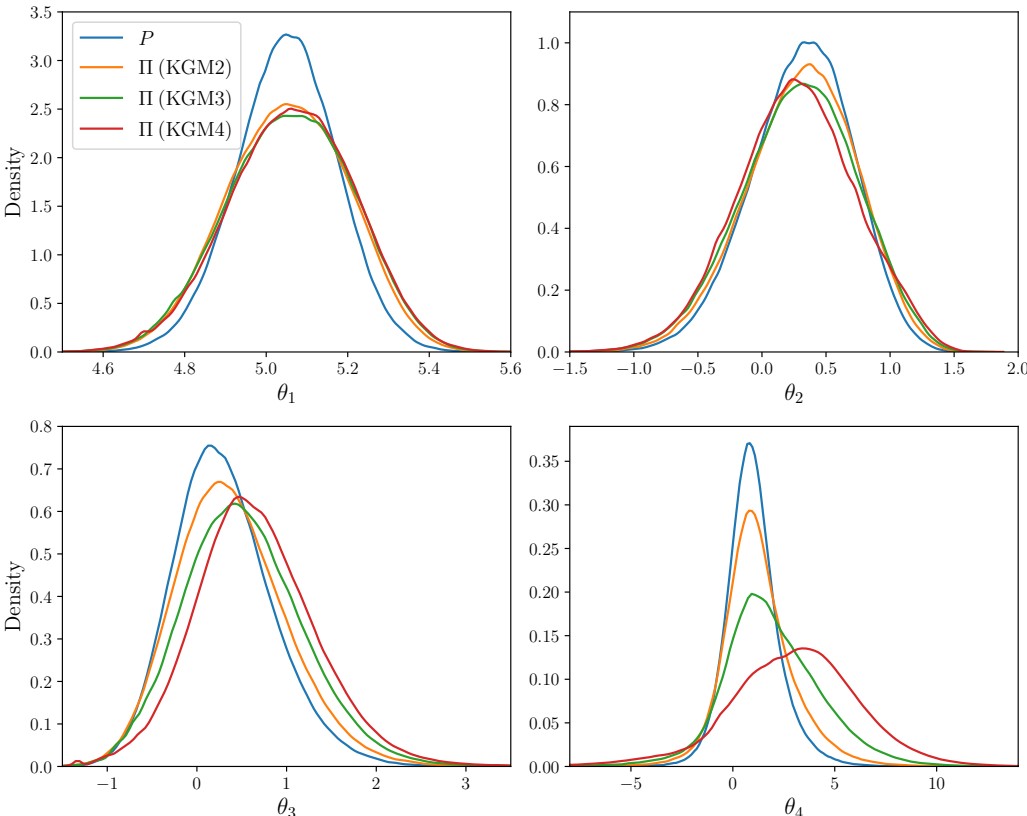

Figure S3: Illustrating the shape of $\Pi$ based on the KGM$s$–Stein kernel for a GARCH(1,1) model, controlling convergence of moments up to order $s \in \{2, 3, 4\}$. The marginal density functions of each distribution were approximated using one-million samples obtained using MCMC.

For this illustration, real data were provided within the model description of `PosteriorDB`, for which the estimated *maximum a posteriori* parameter is $\hat{\phi} = (5.04, 1.36, 0.53, 0.31)$. The marginal distributions of $\Pi$ corresponding to the KGM–Stein kernels of orders $s \in \{2, 3, 4\}$ are compared to the marginals of $P$ in Figure S3. It can be seen that higher orders $s$ correspond to greater over-dispersion of $\Pi$; this makes intuitive sense since larger $s$ corresponds to a more stringent KSD (controlling the convergence of moments up to order $s$) which places greater emphasis on how the tails of $P$ are approximated. Further, for the final skewed marginal of $P$, we note that the distribution $\Pi$ exaggerates the skew, placing more of its mass in the tail of the direction which is positively skewed. Further discussion of skewed targets is contained in Appendix D.8.

### D.5 Stein $\Pi$-Importance Sampling for `PosteriorDB`

To introduce objectivity into our assessment, we exploited the `PosteriorDB` benchmark (Magnusson et al., 2022). This ongoing project is an attempt toward standardised benchmarking, consisting of a collection of posteriors to be numerically approximated. The test problems in `PosteriorDB` are defined in the `Stan` probabilistic programming language, and so `BridgeStan` (Roualdes et al., 2023) was used to directly access posterior densities and their gradients as required. The ambition of `PosteriorDB` is to provide an extensive set of benchmark tasks; at the time we conducted our research, `PosteriorDB` was at Version 0.4.0 and contained 149 models, of which 47 came equipped with a gold-standard sample of size $n = 10^3$, generated from a long run of Hamiltonian Monte Carlo (the No-U-Turn sampler in `Stan`). Of these 47 models, a subset of 40 were found to be compatible with `BridgeStan`, which was at Version 1.0.2 at the time this research was performed. The version of `Stan` that we used was `Stanc3` Version 2.31.0 (Unix). Thus we used a total of 40 test problems for our empirical assessment.

For each test problem, a total of 10 replicate experiments were performed and standard errors were computed. A sampling method was defined as being *significantly better* for approximation of a given target, compared to all other methods considered, if had lower mean KSD *and* the standard error bar did not overlap with the standard error bar of any other method. Table 1 in the main text summarises the performance of SΠIS-MALA, fixing the number of samples to be $n = 3 \times 10^3$. In this appendix, full empirical results are provided.

For sampling from MALA, we used the adaptive algorithm described in Appendix D.3 with a final epoch of length $n_{\max} = 10^5$. Then, whenever a set of $n \ll n_{\max}$ consecutive samples from MALA are required for our experimental assessment, these were obtained by selecting at random a consecutive sequence of length $n$ from the total chain of length $10^5$. This ensures that the performance of unprocessed MALA that we report is not negatively affected by burn-in, in so far as is practical to control.

Full results are presented in Figure S4. These results broadly support the interpretation that SΠIS-MALA usually outperforms SIS-MALA, or otherwise both methods provide a similar level of performance, for the sufficiently large sample sizes $n$ considered. The sample size threshold at which SΠIS-MALA outperforms SIS-MALA appears to be dimension-dependent. A notable exception is panel 29 of Figure S4, a $d = 10$ dimensional task for which SΠIS-MALA provided a substantially worse approximation in KSD for the range of values of $n$ considered.

Figure S4: Benchmarking on `PosteriorDB`. Here we compared raw output from MALA (dotted lines) with the post-processed output provided by the default Stein importance sampling method of Liu and Lee (2017) (SIS-MALA; solid lines) and the proposed Stein Π-Importance Sampling method (SΠIS-MALA; dashed lines). The Langevin (purple) and KGM3–Stein kernels (blue) were used for SIS-MALA and SΠIS-MALA and the associated KSDs are reported as the number $n$ of iterations of MALA is varied. Ten replicates were computed and standard errors were plotted. The name of each model is shown in the title of the corresponding panel, and the dimension $d$ of the parameter vector is given in parentheses. [Langevin–Stein kernel: ········ MALA, ——— SIS-MALA, - - - - - SΠIS-MALA. KGM3–Stein kernel: ········ MALA, ——— SIS-MALA, - - - - - SΠIS-MALA.]

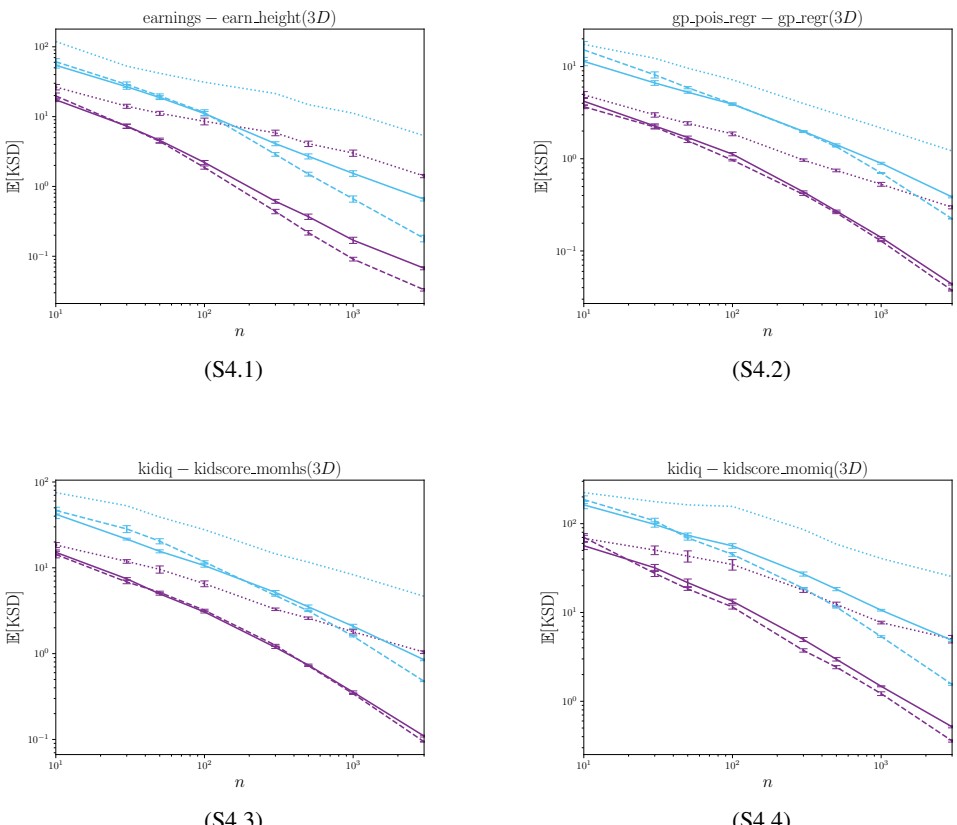

(S4.1)

(S4.2)

(S4.3)

(S4.4)

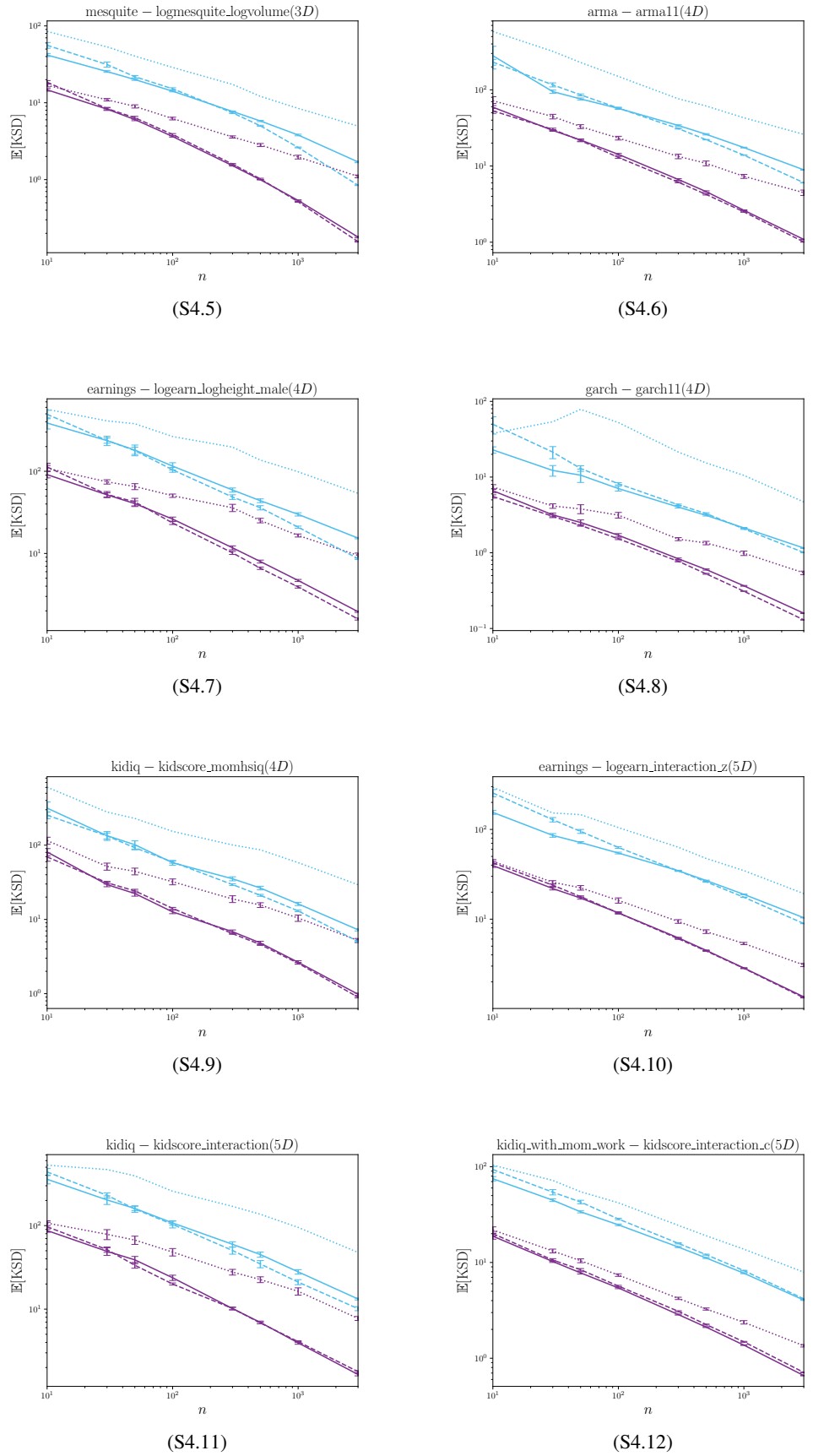

(S4.5)

(S4.6)

(S4.7)

(S4.8)

(S4.9)

(S4.10)

(S4.11)

(S4.12)

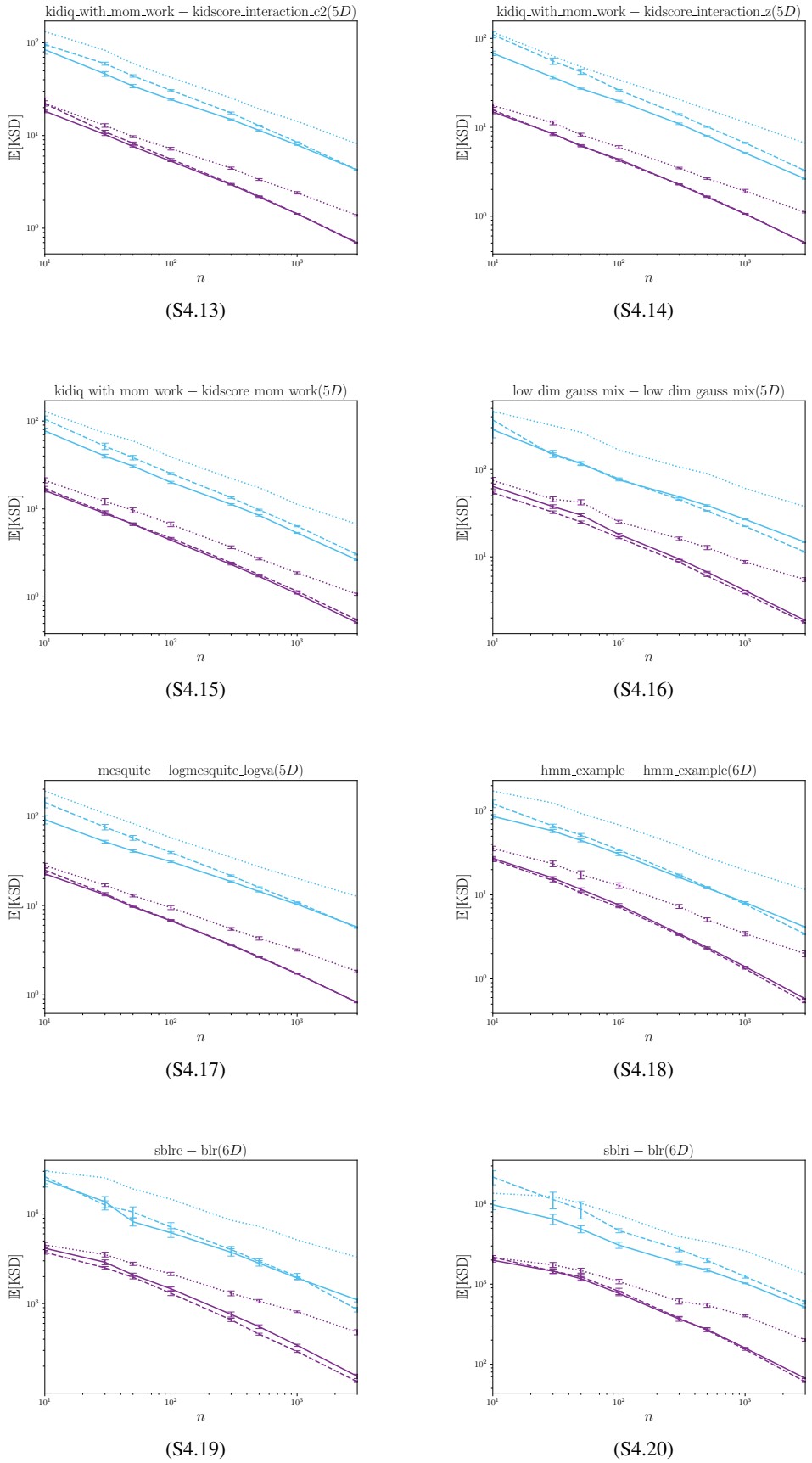

(S4.13)

(S4.14)

(S4.15)

(S4.16)

(S4.17)

(S4.18)

(S4.19)

(S4.20)

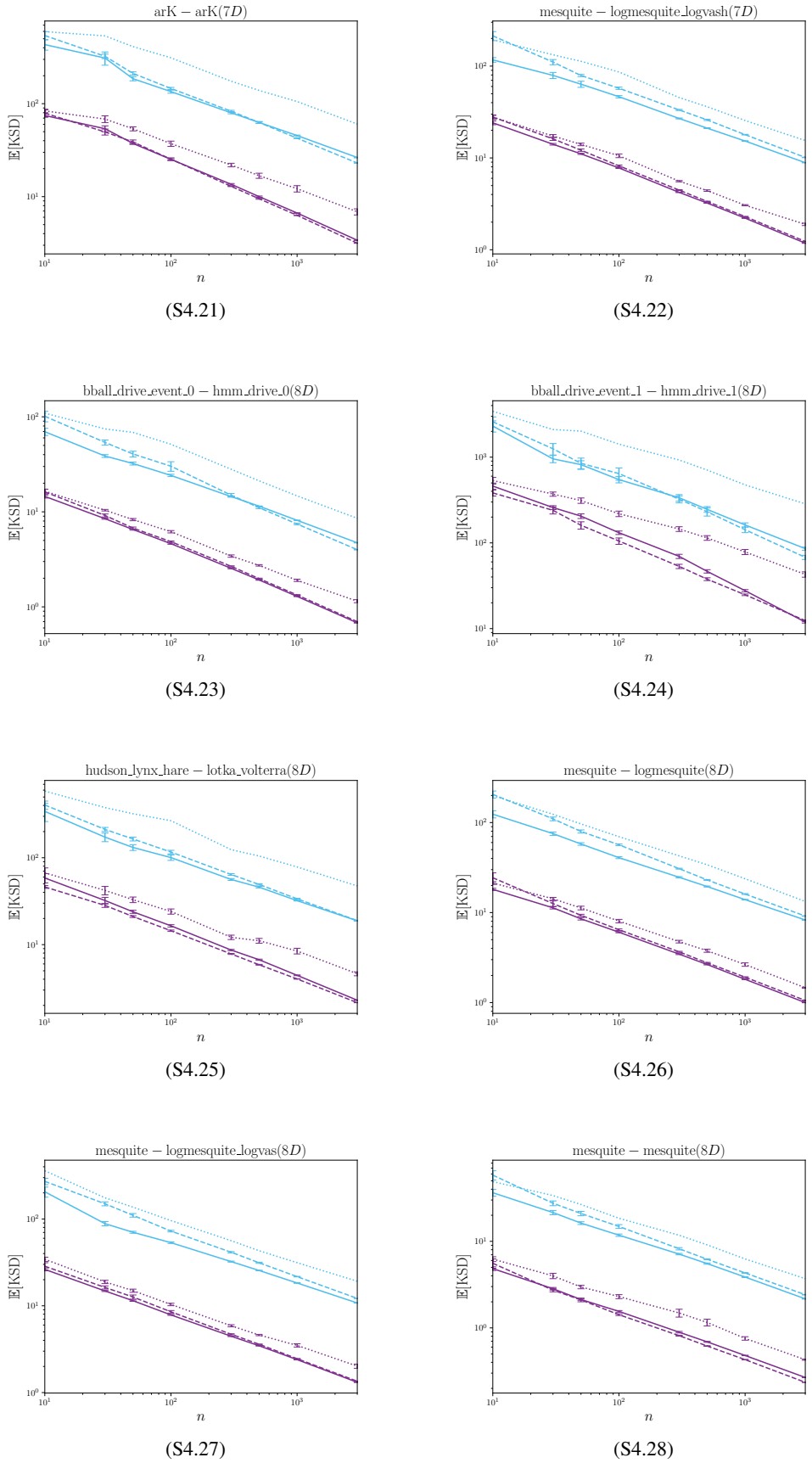

(S4.21)

(S4.22)

(S4.23)

(S4.24)

(S4.25)

(S4.26)

(S4.27)

(S4.28)

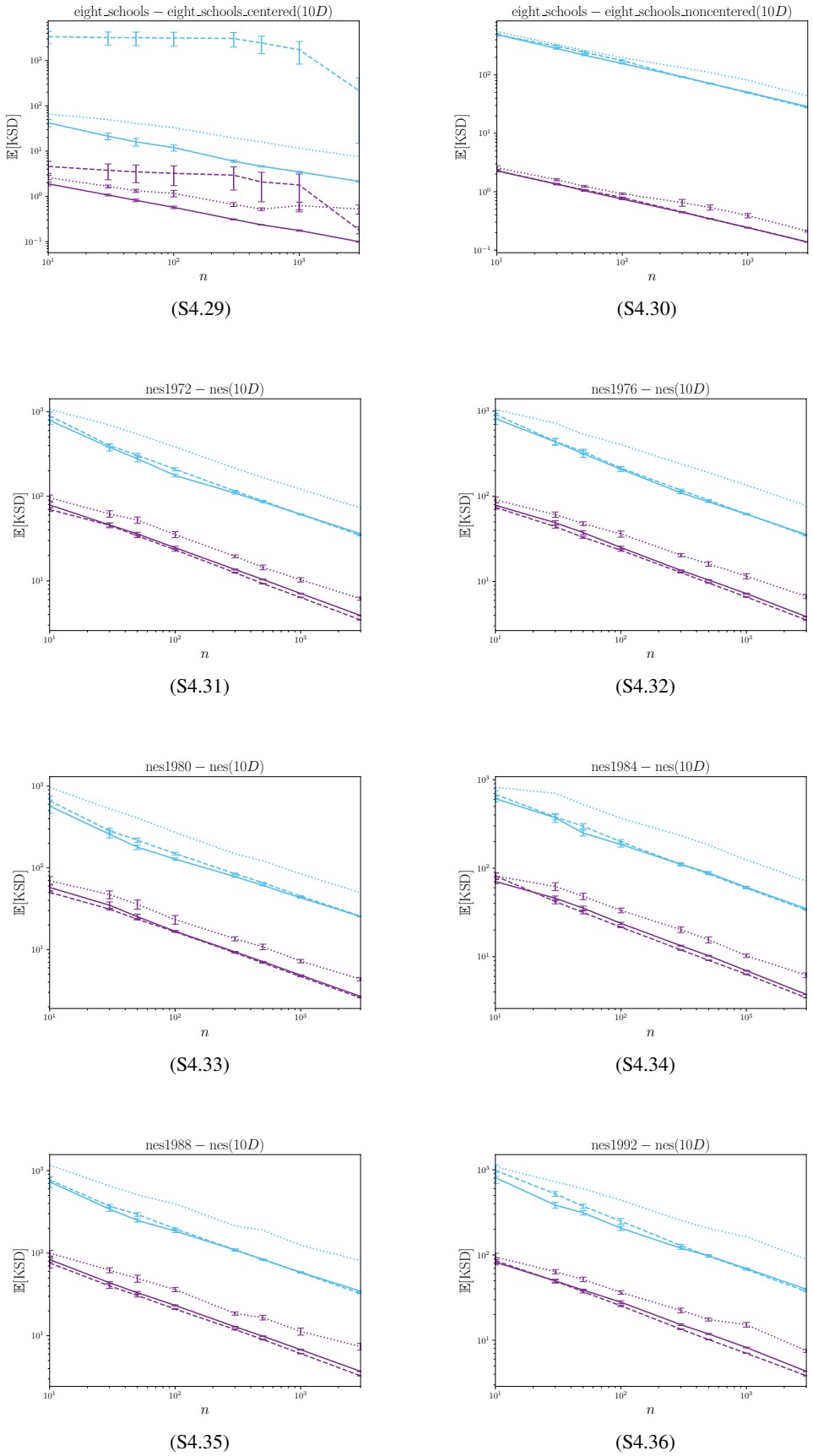

(S4.29)

(S4.30)

(S4.31)

(S4.32)

(S4.33)

(S4.34)

(S4.35)

(S4.36)

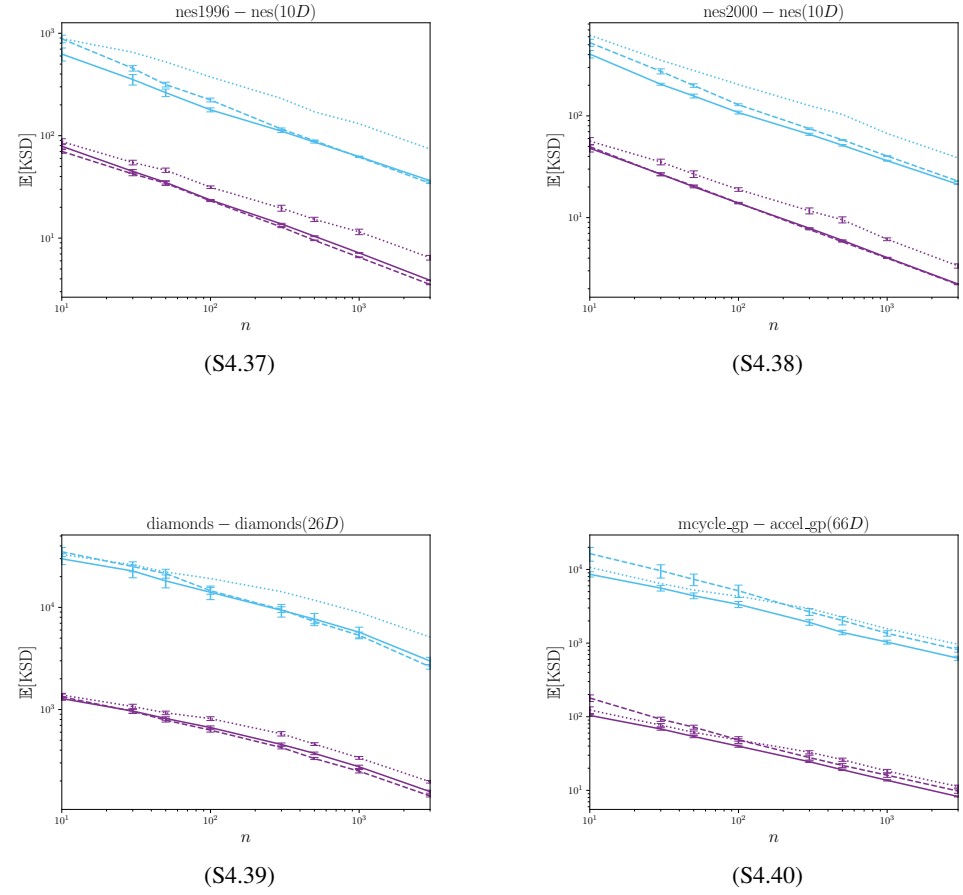

(S4.37)                                          (S4.38)

(S4.39)                                          (S4.40)

## D.6   Stein Π-Thinning for `PosteriorDB`

The results presented in the main text concerned $n = 3 \times 10^3$ samples from MALA, which is near the limit at which the optimal weights $w^\star$ can be computed in a few seconds on a laptop PC. For larger values of $n$, sparse approximation methods are likely to required. In the main text we presented *Stein* Π-*Thinning*, which employs a greedy optimisation perspective to obtain a sparse approximation to the optimal weights at cost $O(m^2 n)$, where $m$ are the number of greedy iterations performed. Explicit and verifiable conditions for the strong consistency of the resulting SΠT-MALA algorithm were established in Section 3.3. The purpose of this appendix is to empirically explore the convergence of SΠT-MALA using the `PosteriorDB` test bed.

In the experiments we report the number of MALA samples was fixed to $n = 10^3$ and the number of greedy iterations was varied from $m = 1$ to $m = 10^3$. The results, in Figure S5, indicate that for most models in `PosteriorDB` the minimum value of KSD is approximately reached when $m$ is anywhere from $\frac{n}{10}$ to $\frac{n}{2}$, representing a modest but practically significant reduction in computational cost compared to SΠIS-MALA. This agrees with the qualitative findings reported in the original Stein thinning paper of Riabiz et al. (2022).

Figure S5: Benchmarking on `PosteriorDB`. Here we investigate the convergence of the sparse approximation provided by the proposed Stein $\Pi$-Thinning method (S$\Pi$T-MALA). The Langevin (purple) and KGM3–Stein kernels (blue) were used for S$\Pi$T-MALA and the associated KSDs are reported as the number $m$ of iterations of Stein thinning is varied. Ten replicates were computed and standard errors were plotted. The name of each model is shown in the title of the corresponding panel, and the dimension $d$ of the parameter vector is given in parentheses. [Langevin–Stein kernel:———S$\Pi$T-MALA. KGM3–Stein kernel:———S$\Pi$T-MALA.]

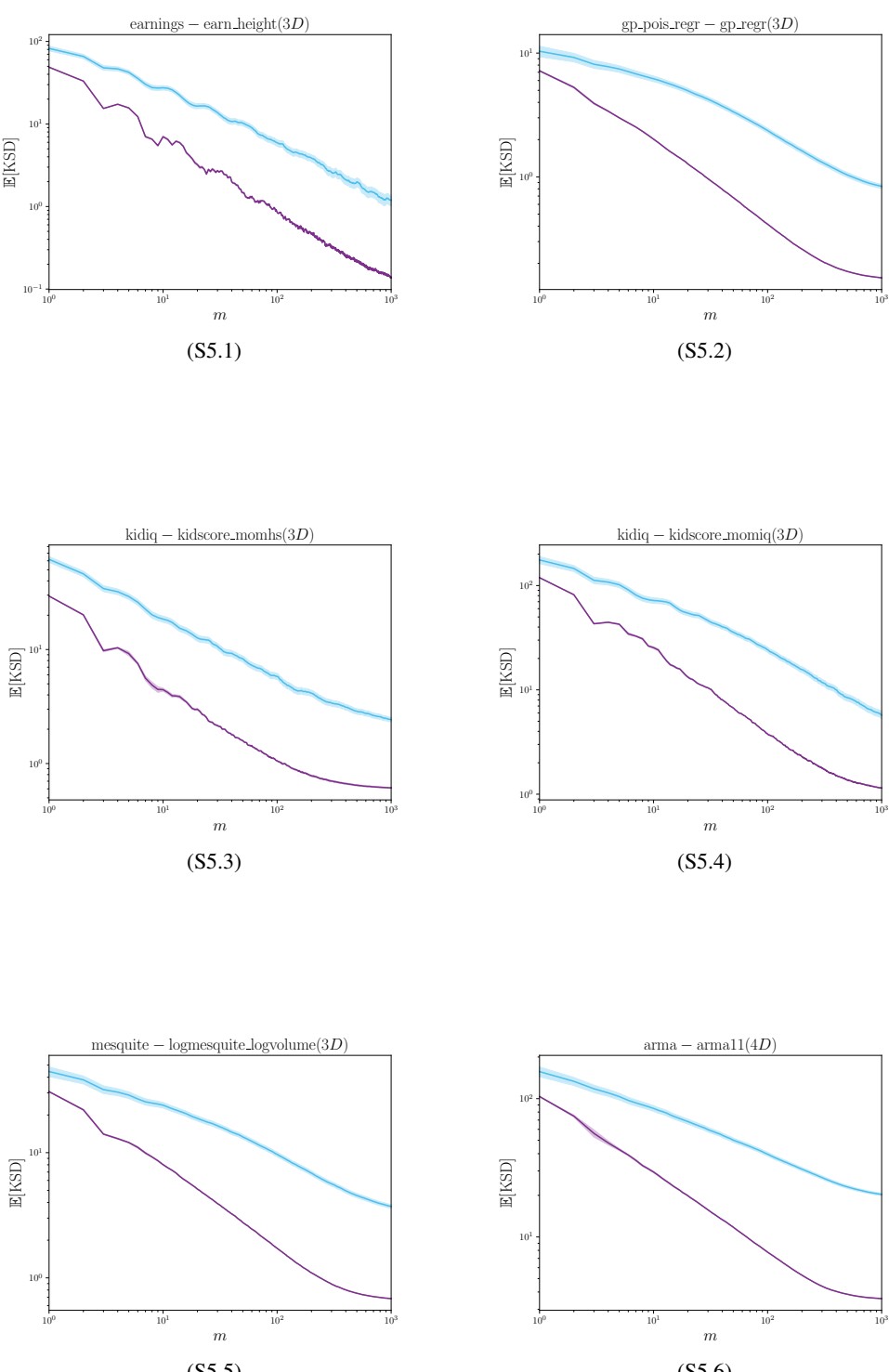

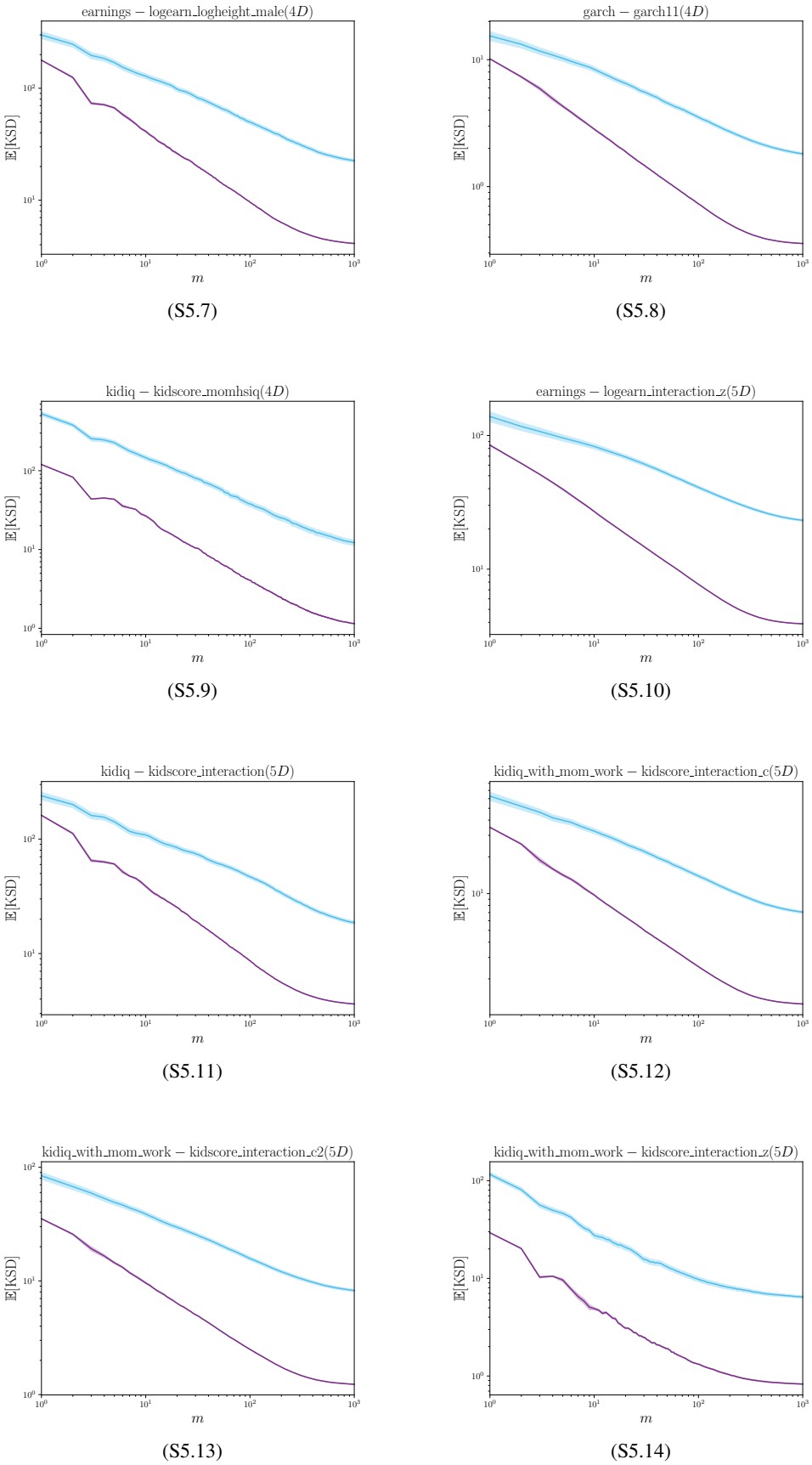

(S5.7)

(S5.8)

(S5.9)

(S5.10)

(S5.11)

(S5.12)

(S5.13)

(S5.14)

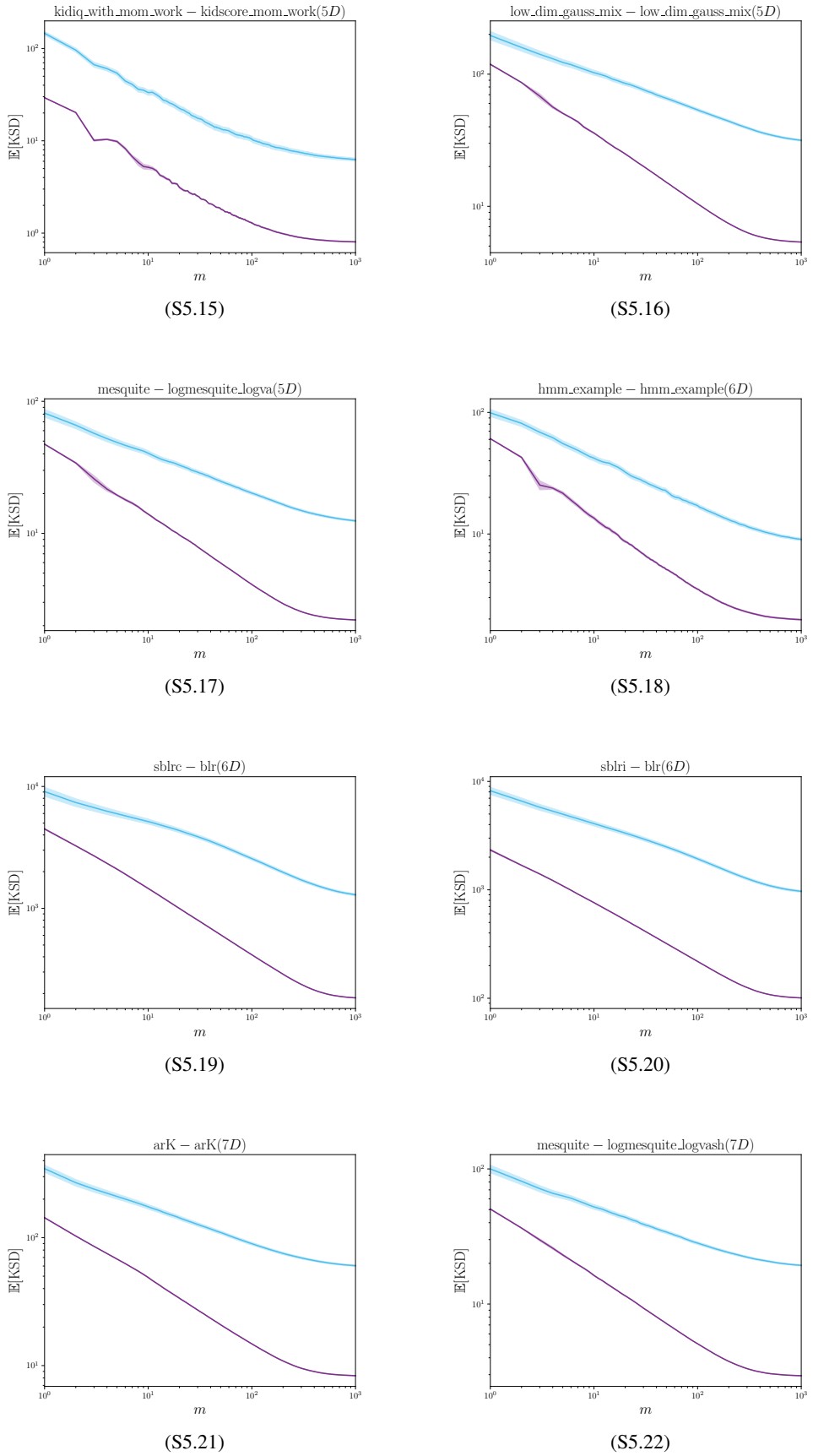

(S5.15)

(S5.16)

(S5.17)

(S5.18)

(S5.19)

(S5.20)

(S5.21)

(S5.22)

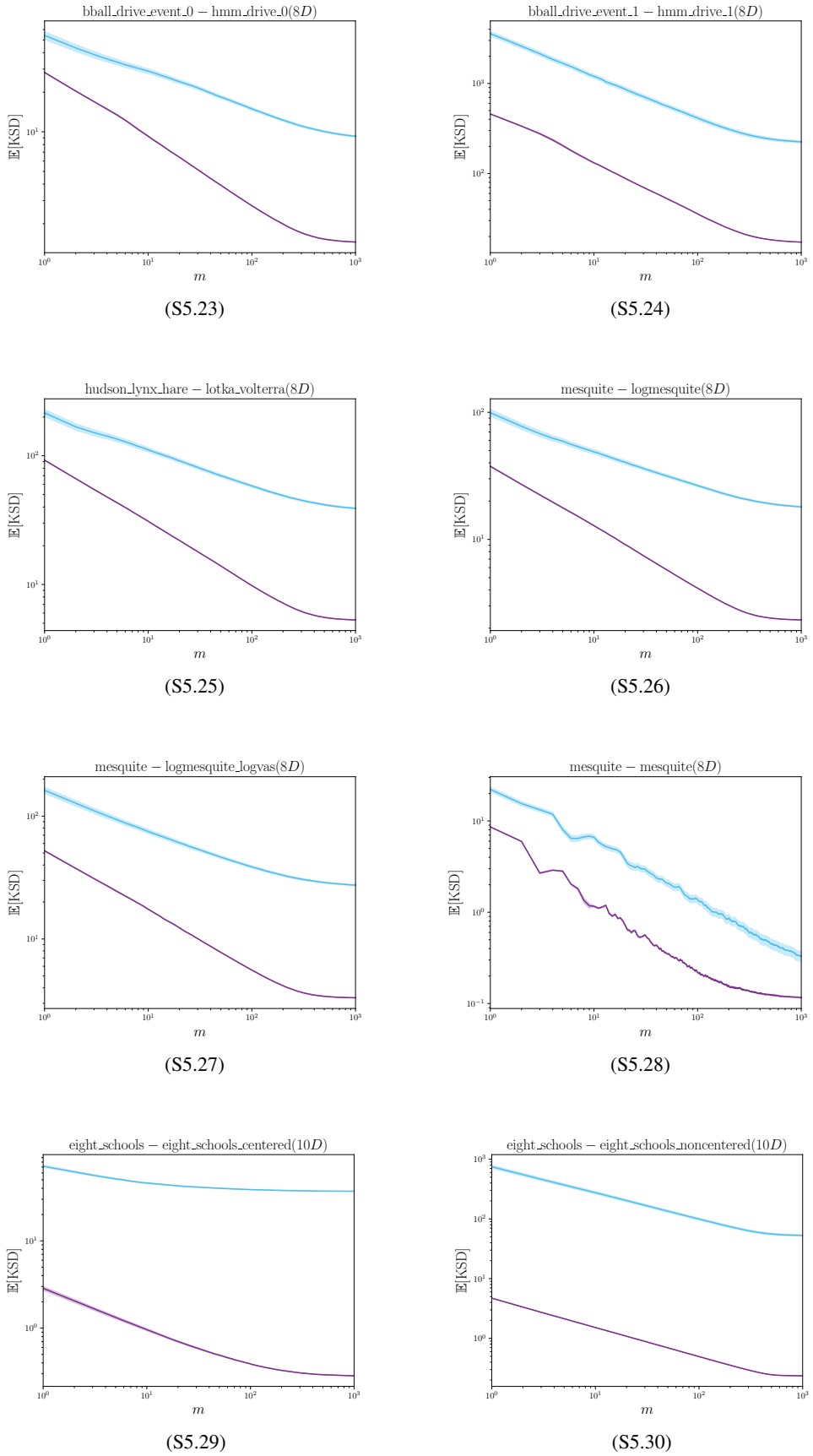

(S5.23)

(S5.24)

(S5.25)

(S5.26)

(S5.27)

(S5.28)

(S5.29)

(S5.30)

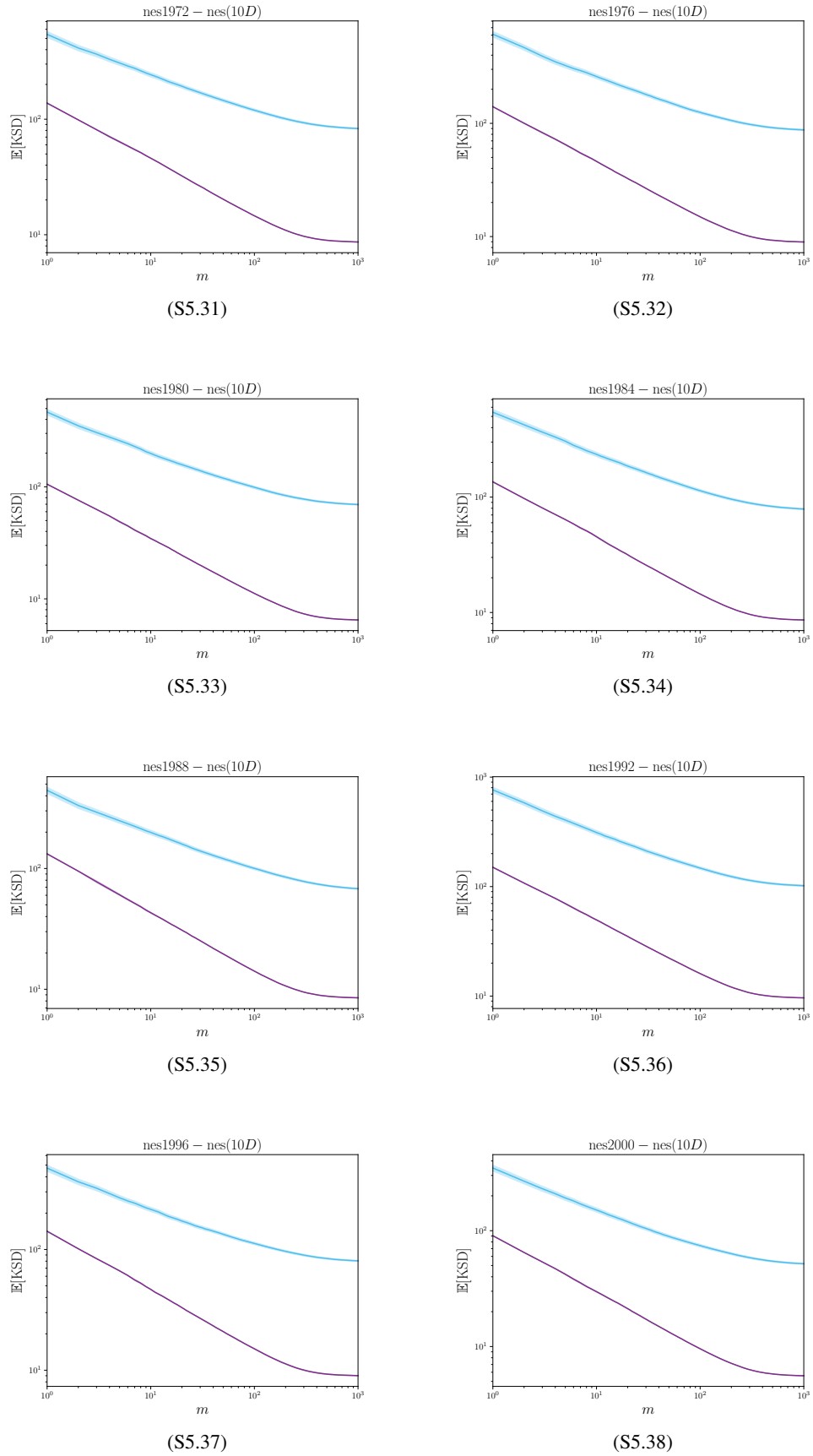

(S5.31)

(S5.32)

(S5.33)

(S5.34)

(S5.35)

(S5.36)

(S5.37)

(S5.38)

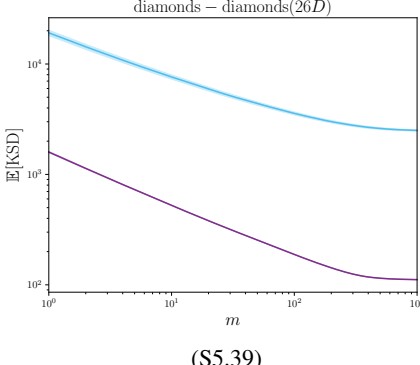

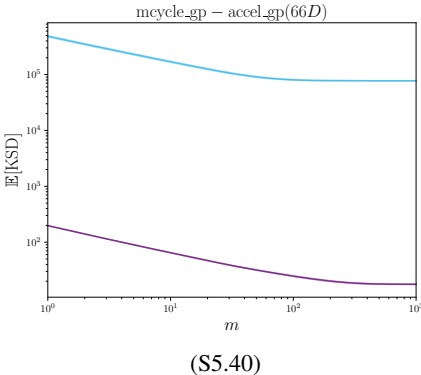

(S5.39)  (S5.40)

## D.7 Performance of Stein Discrepancies

The properties of Stein discrepancies was out of scope for this work. Nonetheless, there is much interest in better understanding the properties of KSDs, and in this appendix the performance of SΠIS-MALA in terms of 1-Wasserstein divergence is reported. This was made possible since `PosteriorDB` supplies a set of posterior samples obtained from a long run of Hamiltonian Monte Carlo (the No-U-Turn sampler in `Stan`) which we treat as a gold standard.

Full results are presented in Figure S6 and Table 2. Broadly speaking, in most cases the minimisation of KSD seems to be associated with minimisation of 1-Wasserstein distance, and in particular a significant improvement of SΠIS-MALA over SIS-MALA is reported for $\approx 63\%$ of tasks in the `PosteriorDB` benchmark. However there are some scenarios for which minimisation of KSD is loosely, if at all, related to minimisation of 1-Wasserstein divergence. In these cases, we attribute this performance to a combination of two factors: First, the *blindness to mixing proportions* phenomena, described in Wenliang and Kanagawa (2021); Koehler et al. (2022); Liu et al. (2023), which is a pathology of KSDs in general. Second, the Langevin–Stein kernel cannot be expected to control convergence in 1-Wasserstein, since convergence in 1-Wasserstein is equivalent to weak convergence plus convergence of the first moment. Focusing therefore on the KGM3–Stein kernel only, it is encouraging to note that SΠIS-MALA outperforms SIS-MALA on 83% of tasks in `PosteriorDB` in the 1-Wasserstein metric, as shown in Table 2. However, it is interesting to observe that MALA performed well in the 1-Wasserstein sense across the `PosteriorDB` test bed.

The development of improved Stein discrepancies is an active area of research, and we emphasise that the methodology developed in this work can be applied to *any* KSDs, including potentially KSDs with better or more direct control over standard notions of convergence (such as 1-Wasserstein) that in the future may be developed.

Figure S6: Performance of Stein discrepancies on `PosteriorDB`. Here we compared raw output from MALA (dotted lines) with the post-processed output provided by the default Stein importance sampling method of Liu and Lee (2017) (SIS-MALA; solid lines) and the proposed Stein Π-Importance Sampling method (SΠIS-MALA; dashed lines). The Langevin (purple) and KGM3–Stein kernels (blue) were used for SIS-MALA and SΠIS-MALA, and the 1-Wasserstein divergence is reported as the number $n$ of iterations of MALA is varied. Ten replicates were computed and standard errors were plotted. The name of each model is shown in the title of the corresponding panel, and the dimension $d$ of the parameter vector is given in parentheses. [Legend: ⋯⋯⋯Raw MALA. Langevin–Stein kernel: ——SIS-MALA, - - - - SΠIS-MALA. KGM3–Stein kernel: ——SIS-MALA, - - - - SΠIS-MALA.]

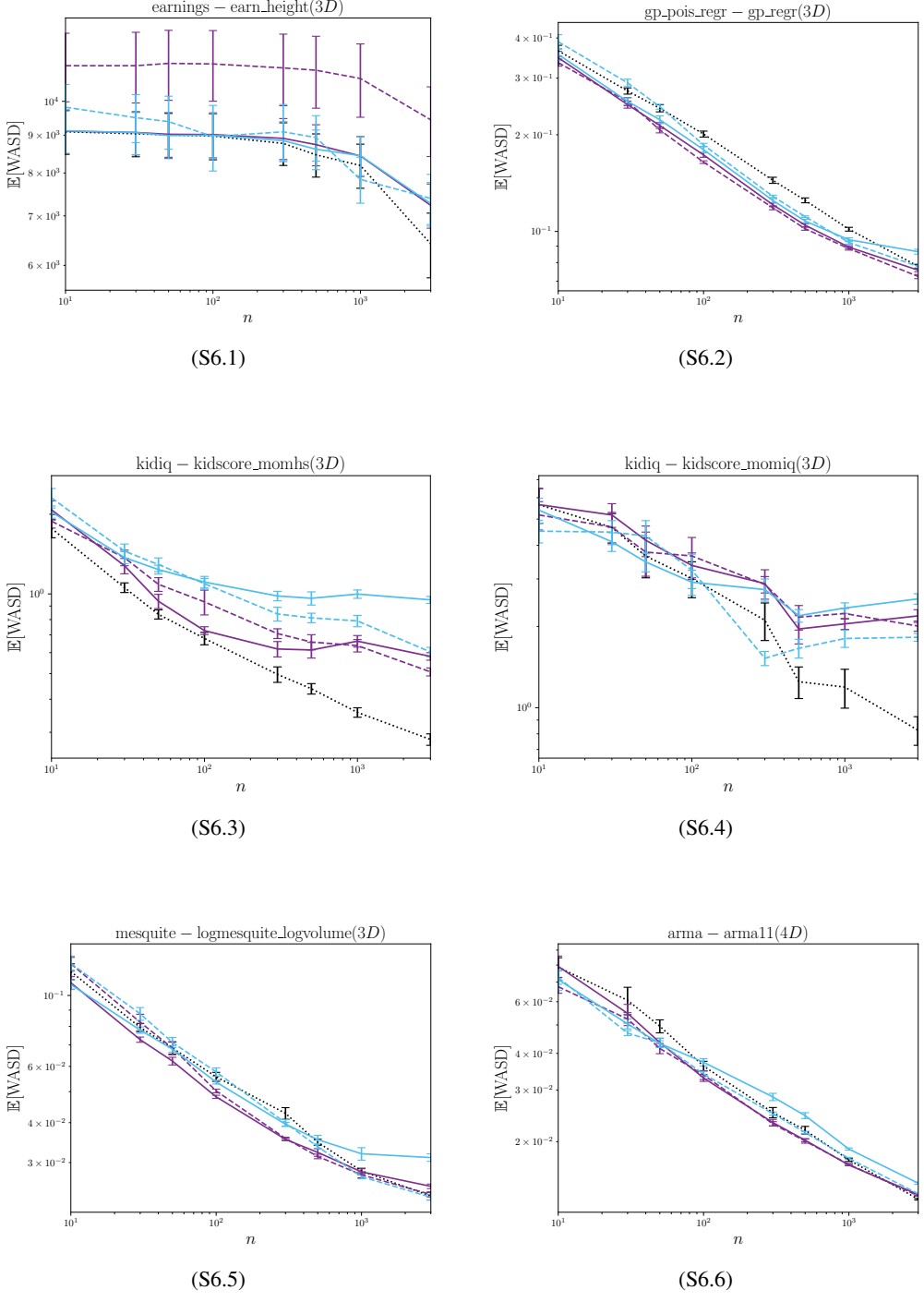

(S6.1)  (S6.2)

(S6.3)  (S6.4)

(S6.5)  (S6.6)

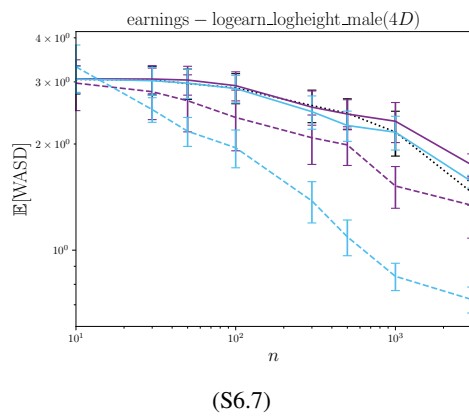

(S6.7)

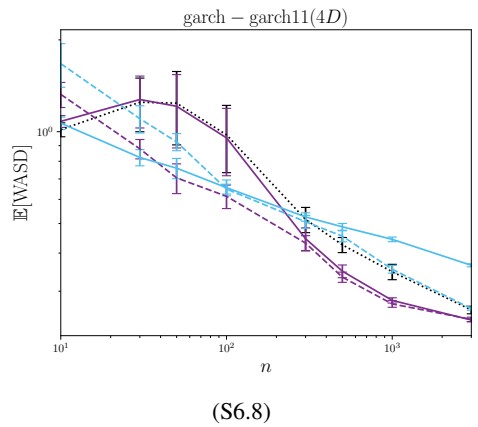

(S6.8)

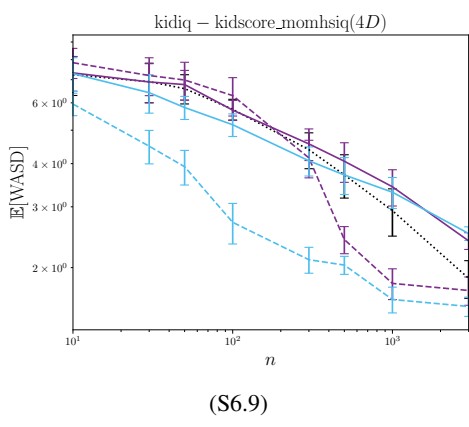

(S6.9)

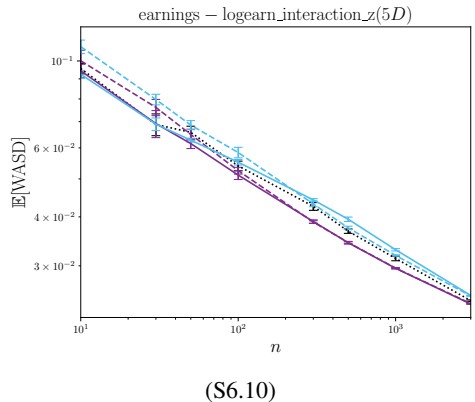

(S6.10)

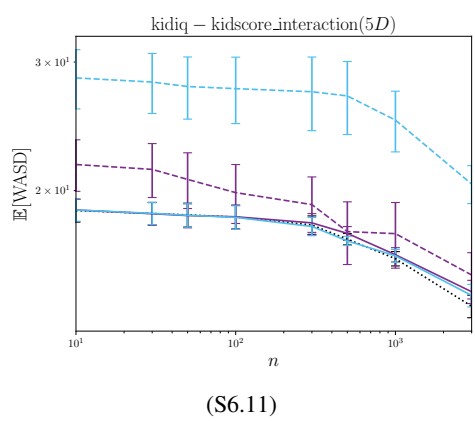

(S6.11)

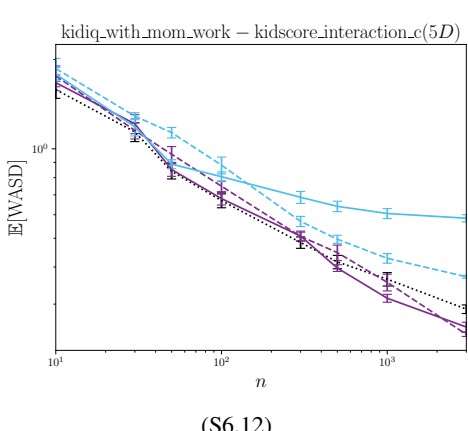

(S6.12)

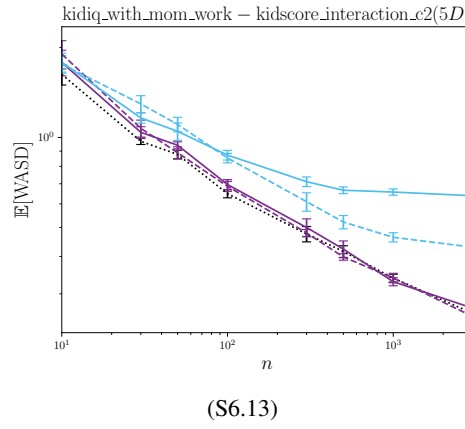

(S6.13)

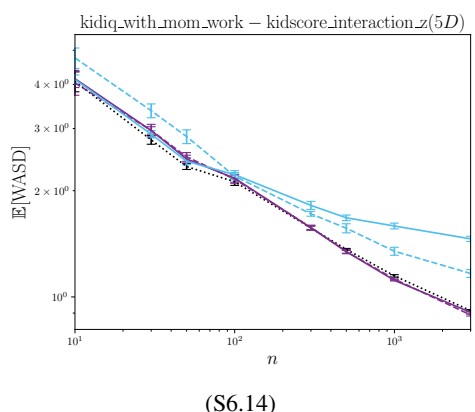

(S6.14)

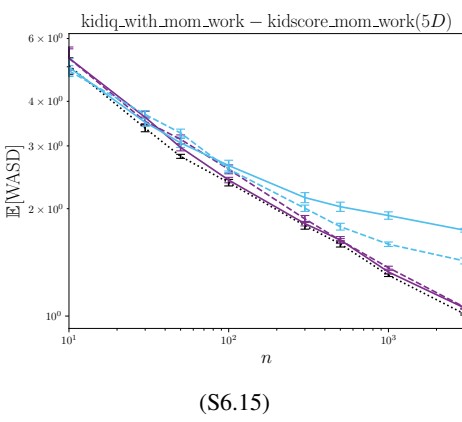

(S6.15)

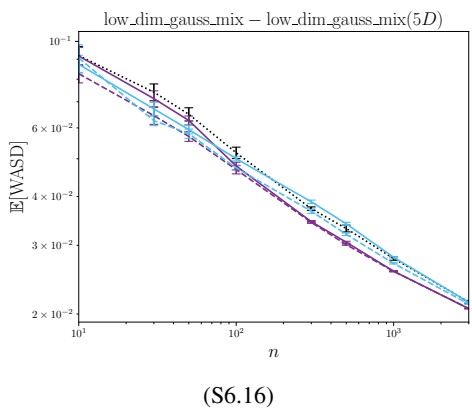

(S6.16)

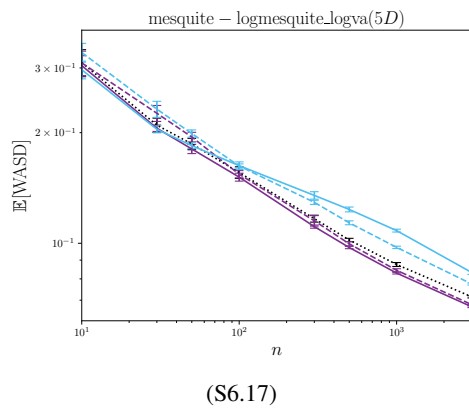

(S6.17)

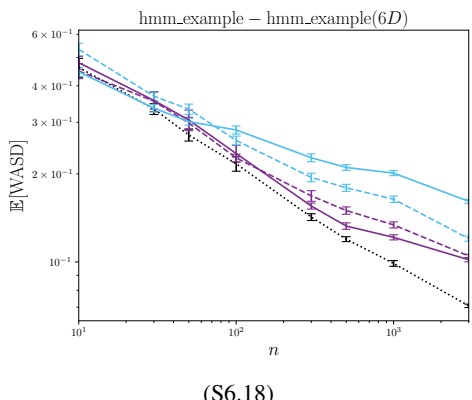

(S6.18)

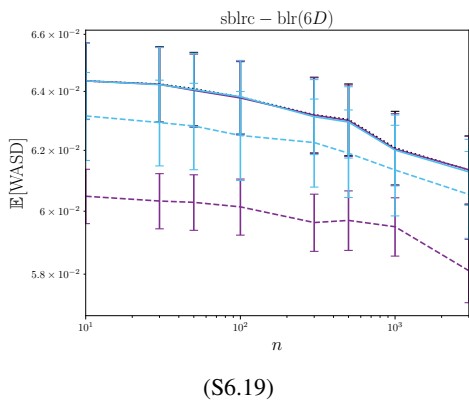

sblrc − blr(6*D*)

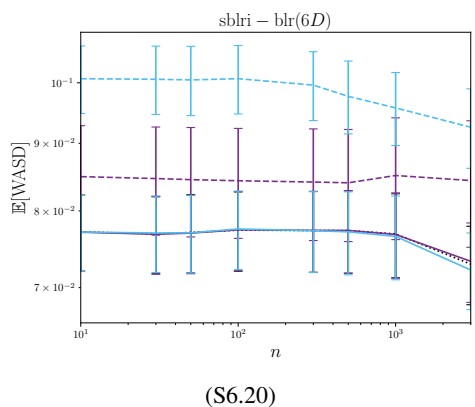

sblri − blr(6*D*)

(S6.19)

(S6.20)

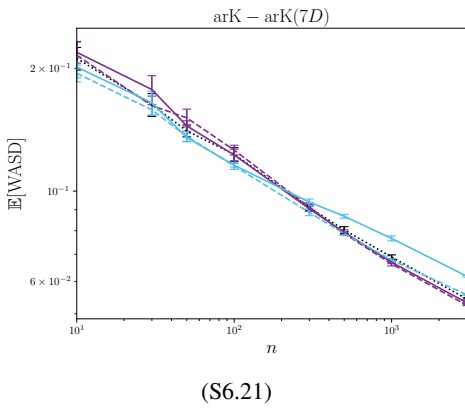

arK − arK(7*D*)

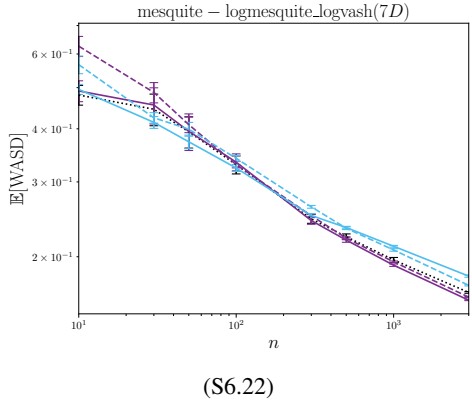

mesquite − logmesquite_logvash(7*D*)

(S6.21)

(S6.22)

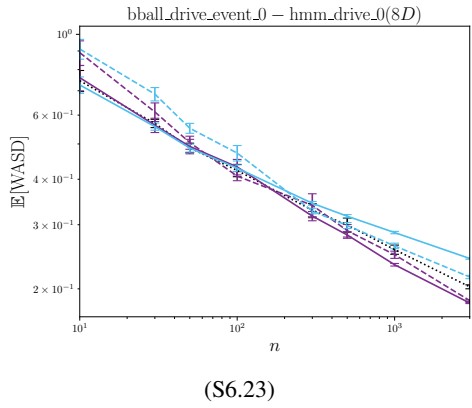

bball_drive_event_0 − hmm_drive_0(8*D*)

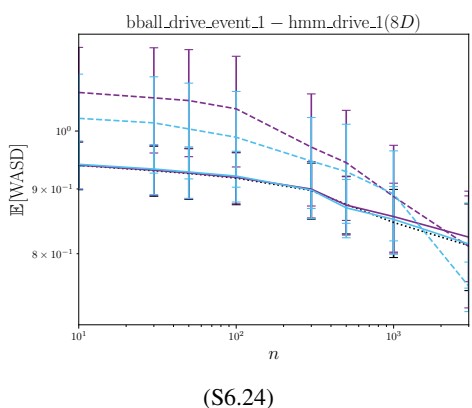

bball_drive_event_1 − hmm_drive_1(8*D*)

(S6.23)

(S6.24)

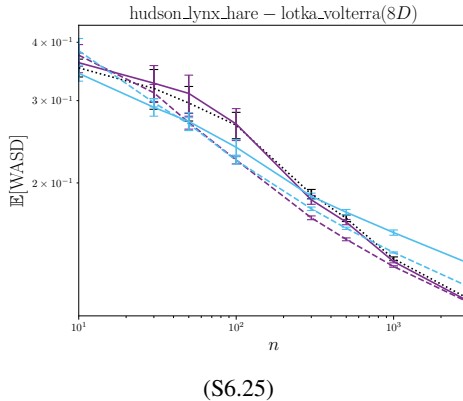

(S6.25)

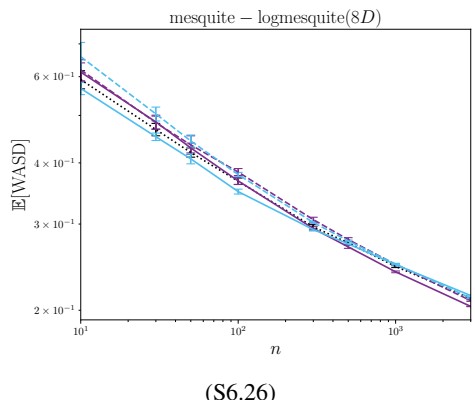

(S6.26)

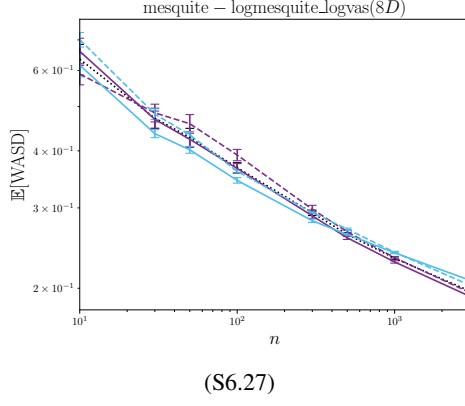

(S6.27)

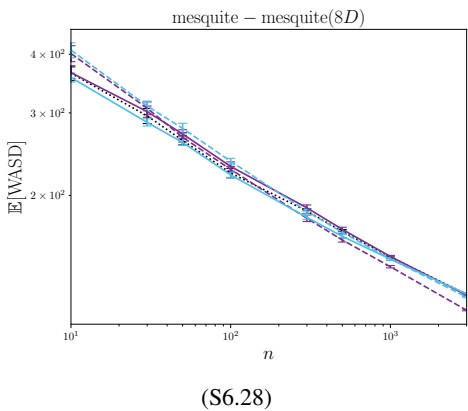

(S6.28)

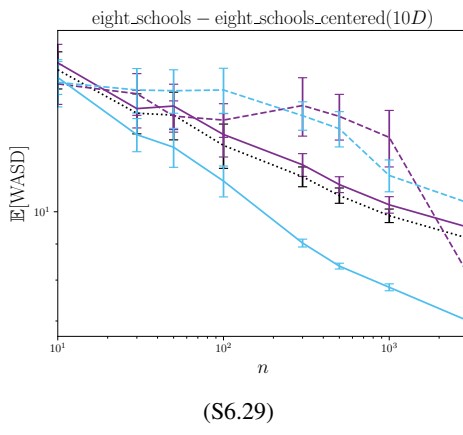

(S6.29)

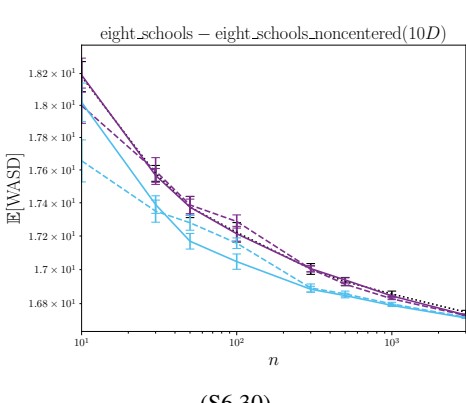

(S6.30)

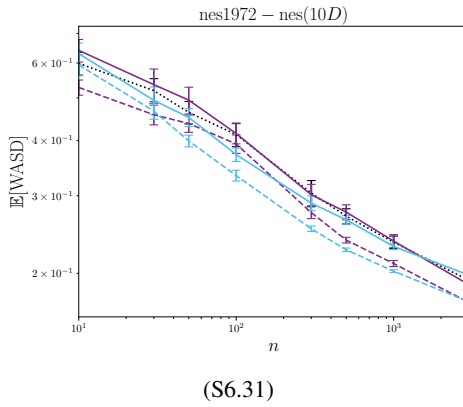

$\text{nes1972} - \text{nes}(10D)$

(S6.31)

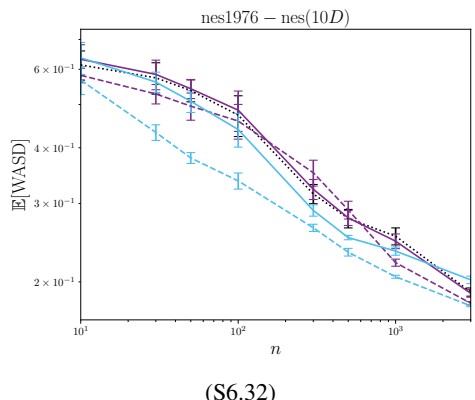

$\text{nes1976} - \text{nes}(10D)$

(S6.32)

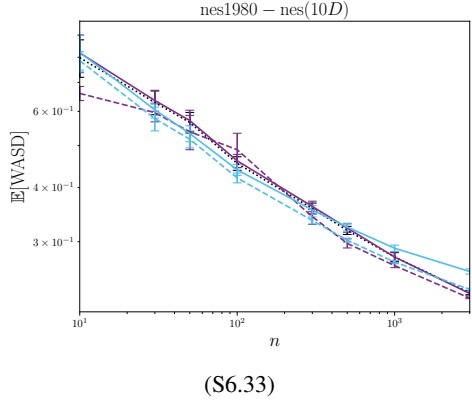

$\text{nes1980} - \text{nes}(10D)$

(S6.33)

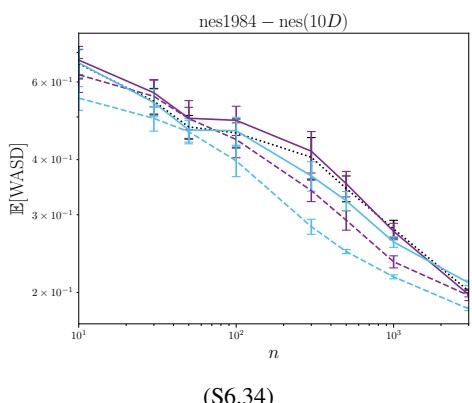

$\text{nes1984} - \text{nes}(10D)$

(S6.34)

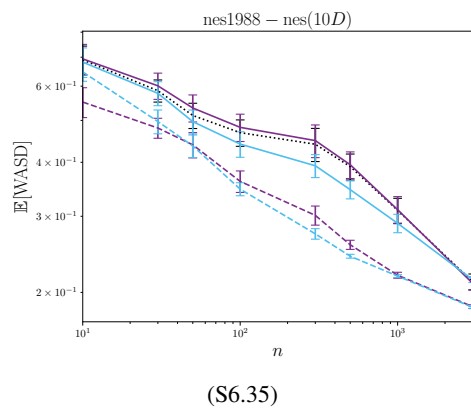

$\text{nes1988} - \text{nes}(10D)$

(S6.35)

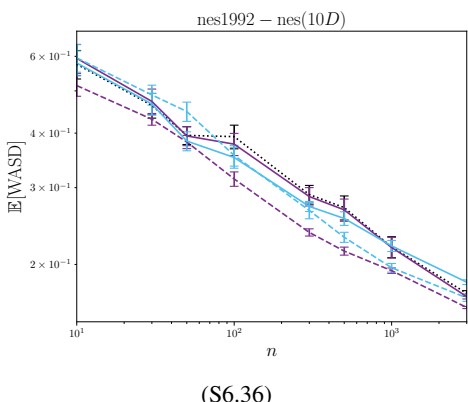

$\text{nes1992} - \text{nes}(10D)$

(S6.36)

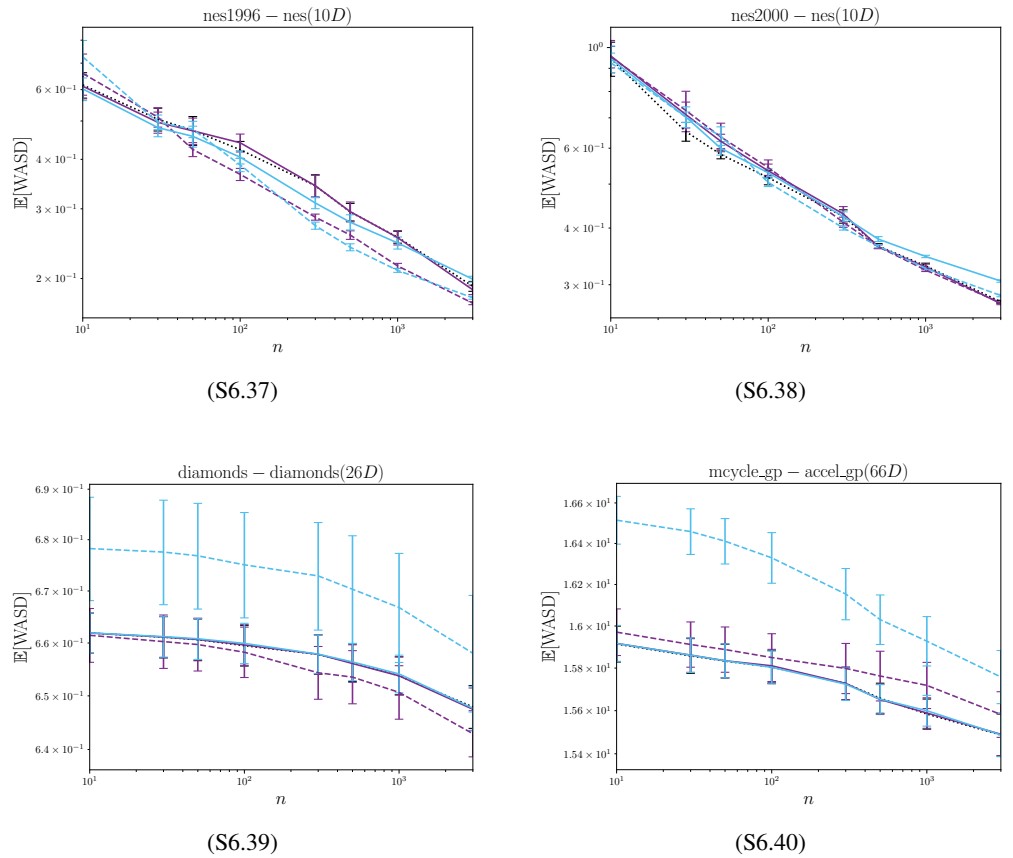

(S6.37)

(S6.38)

(S6.39)

(S6.40)

### D.8 Investigation for a Skewed Target

This final appendix contrasts the 1-Wasserstein optimal sampling distribution $\Pi_1$ (c.f. Section 2.1), with the choice of $\Pi$ that we recommended in (8). In particular, we focus on the KGM3–Stein kernel under a heavily skewed $P$, for which $\Pi_1$ and $\Pi$ can be markedly different.

For this investigation a bivariate skew-normal target was constructed, where the density is given by $p(x_1, x_2) = 4\phi(x_1)\Phi(6x_1)\phi(x_2)\Phi(-3x_2)$, with $\phi$ and $\Phi$ respectively denoting the density and distribution functions of a standard Gaussian. The density $p$ of $P$, together with the marginal densities of $\Pi_1$ and $\Pi$, are plotted in Figure S7. It can be seen that, while both $\Pi_1$ and $\Pi$ are over-dispersed with respect to $P$, our recommended $\Pi$ assigns proportionally more mass to the tail that is positively skewed.

The performance of Stein $\Pi$-Importance Sampling based on $\Pi_1$ and $\Pi$ is compared in Figure S8. Though both choices lead to an improvement relative to Stein importance sampling algorithm with $\Pi = P$, the use of $\Pi$ leads to a significant further reduction (on average) in KSD compared to $\Pi_1$. Based on our investigations, this finding seems general; the use of $\Pi_1$ does not realise the full potential of Stein $\Pi$-Imporance sampling when the target is skewed.

|  | | Langevin–Stein Kernel | | | KGM3–Stein Kernel | | |
|---|---|---|---|---|---|---|---|
| Task | $d$ | MALA | SIS - MALA | SΠIS - MALA | MALA | SIS - MALA | SΠIS - MALA |
| earnings-earn_height | 3 | **6420.0** | 7230.0 | 9440.0 | **6420.0** | 7280.0 | 7390.0 |
| gp_pois_regr-gp_regr | 3 | 0.0779 | 0.0758 | **0.0724** | 0.0779 | 0.0865 | **0.0778** |
| kidiq-kidscore_momhs | 3 | **0.282** | 0.581 | 0.508 | **0.282** | 0.950 | 0.604 |
| kidiq-kidscore_momiq | 3 | **0.826** | 2.19 | 2.01 | **0.826** | 2.53 | 1.82 |
| mesquite-logmesquite_logvolume | 3 | **0.0236** | 0.0253 | 0.0238 | 0.0236 | 0.0311 | **0.0234** |
| arma-arma11 | 4 | **0.0127** | 0.0132 | 0.0131 | **0.0127** | 0.0144 | 0.0133 |
| earnings-logearn_logheight_male | 4 | 1.46 | 1.75 | **1.34** | 1.46 | 1.56 | **0.725** |
| garch-garch11 | 4 | 0.260 | **0.241** | 0.243 | **0.260** | 0.365 | 0.263 |
| kidiq-kidscore_momhsiq | 4 | 1.86 | 2.39 | **1.72** | 1.86 | 2.51 | **1.54** |
| earnings-logearn_interaction_z | 5 | 0.0245 | **0.0240** | 0.0240 | 0.0245 | 0.0252 | 0.0251 |
| kidiq-kidscore_interaction | 5 | **13.9** | 14.5 | 15.3 | **13.9** | 14.4 | 20.4 |
| kidiq_with_mom_work-kidscore_interaction_c | 5 | 0.289 | 0.251 | **0.237** | **0.289** | 0.584 | 0.371 |
| kidiq_with_mom_work-kidscore_interaction_c2 | 5 | 0.258 | 0.269 | **0.252** | **0.258** | 0.638 | 0.430 |
| kidiq_with_mom_work-kidscore_interaction_z | 5 | 0.914 | 0.904 | **0.889** | **0.914** | 1.46 | 1.17 |
| kidiq_with_mom_work-kidscore_mom_work | 5 | **1.01** | 1.05 | 1.06 | **1.01** | 1.74 | 1.43 |
| low_dim_gauss_mix-low_dim_gauss_mix | 5 | 0.0215 | **0.0206** | 0.0207 | 0.0215 | 0.0214 | **0.0212** |
| mesquite-logmesquite_logva | 5 | 0.0715 | **0.0672** | 0.0681 | **0.0715** | 0.0833 | 0.0775 |
| hmm_example-hmm_example | 6 | **0.0708** | 0.102 | 0.105 | **0.0708** | 0.161 | 0.12 |
| sblrc-blr | 6 | 0.0613 | 0.0614 | **0.0581** | 0.0613 | 0.0613 | **0.0605** |
| sblri-blr | 6 | **0.0729** | 0.0733 | 0.0843 | 0.0729 | **0.0722** | 0.0926 |
| arK-arK | 7 | 0.0544 | 0.0533 | **0.0525** | **0.0544** | 0.0618 | 0.0557 |
| mesquite-logmesquite_logvash | 7 | 0.165 | **0.158** | 0.161 | **0.165** | 0.180 | 0.171 |
| bball_drive_event_0-hmm_drive_0 | 8 | 0.203 | **0.183** | 0.186 | **0.203** | 0.242 | 0.216 |
| bball_drive_event_1-hmm_drive_1 | 8 | 0.812 | 0.825 | **0.811** | 0.812 | 0.814 | **0.754** |
| hudson_lynx_hare-lotka_volterra | 8 | 0.113 | 0.112 | **0.111** | **0.113** | 0.135 | 0.120 |
| mesquite-logmesquite | 8 | 0.212 | **0.204** | 0.209 | 0.212 | 0.214 | **0.212** |
| mesquite-logmesquite_logvas | 8 | 0.197 | **0.192** | 0.196 | **0.197** | 0.208 | 0.203 |
| mesquite-mesquite | 8 | 123.0 | 122.0 | **114.0** | 123.0 | 123.0 | **121.0** |
| eight_schools-eight_schools_centered | 10 | 9.17 | 9.50 | **8.14** | 9.17 | **7.00** | 10.3 |
| eight_schools-eight_schools_noncentered | 10 | 16.8 | 16.7 | **16.7** | 16.8 | **16.7** | 16.7 |
| nes1972-nes | 10 | 0.193 | 0.189 | **0.172** | 0.193 | 0.198 | **0.172** |
| nes1976-nes | 10 | 0.190 | 0.189 | **0.179** | 0.190 | 0.202 | **0.177** |
| nes1980-nes | 10 | 0.229 | 0.228 | **0.222** | **0.229** | 0.256 | 0.233 |
| nes1984-nes | 10 | 0.202 | 0.198 | **0.197** | 0.202 | 0.210 | **0.184** |
| nes1988-nes | 10 | 0.212 | 0.210 | **0.185** | 0.212 | 0.214 | **0.185** |
| nes1992-nes | 10 | 0.172 | 0.169 | **0.159** | 0.172 | 0.182 | **0.168** |
| nes1996-nes | 10 | 0.191 | 0.187 | **0.173** | 0.191 | 0.199 | **0.179** |
| nes2000-nes | 10 | 0.275 | **0.273** | 0.274 | 0.275 | 0.306 | **0.284** |
| diamonds-diamonds | 26 | 0.648 | 0.647 | **0.643** | 0.648 | **0.648** | 0.658 |
| mcycle_gp-accel_gp | 66 | 15.5 | **15.5** | 15.6 | 15.5 | **15.5** | 15.8 |

Table 2: Benchmarking on `PosteriorDB`. Here we compared raw output from MALA with the post-processed output provided by the default Stein importance sampling method of Liu and Lee (2017) (SIS-MALA) and the proposed Stein Π-Importance Sampling method (SΠIS-MALA). Here $d = \dim(P)$ and the number of MALA samples was $n = 3 \times 10^3$. The Langevin and KGM3–Stein kernels were used for SIS-MALA and SΠIS-MALA and the associated 1-Wasserstein distances are reported. Ten replicates were computed and statistically significant improvement is highlighted in **bold**.

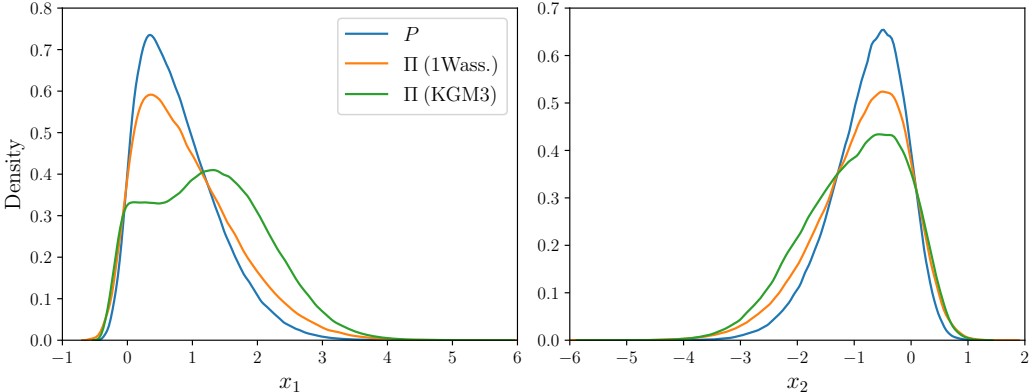

Figure S7: Comparing the proposed distribution Π (KGM3; based on the KGM3–Stein kernel) to $\Pi_1$ (1Wass.; the optimal choice for 1-Wasserstein quantisation from Section 2.1) for a bivariate skew-normal target ($d = 2$). The marginal density functions of each distribution were approximated using $10^6$ samples from MCMC.

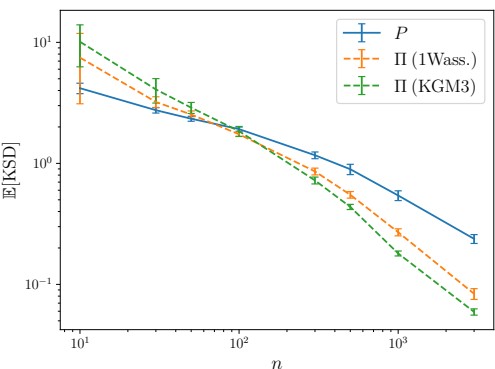

Figure S8: Comparing the performance of using the proposed distribution $\Pi$ (KGM3; based on the KGM3–Stein kernel) to $\Pi_1$ (1Wass.; the optimal choice for 1-Wasserstein quantisation from Section 2.1) for a bivariate skew-normal target ($d = 2$). The mean kernel Stein discrepancy (KSD) for Stein $\Pi$-Importance Sampling was estimated; in each case, the KSD based on the KGM3–Stein kernel was computed. Solid lines indicate the baseline case of sampling from $P$, while dashed lines indicate sampling from $\Pi$. (The experiment was repeated 10 times and standard error bars are plotted.)

