# OpenReview forum: "Stein $\Pi$-Importance Sampling"
_NeurIPS.cc/2023/Conference — NeurIPS 2023 spotlight_

### Official Review · Reviewer_pfRv · 2023-07-02

**Soundness:** 3 good
**Presentation:** 4 excellent
**Contribution:** 3 good
**Rating:** 7
**Confidence:** 3

**Summary:**

This paper presents a novel approach for constructing an MCMC target that is specifically designed for post-processing using Stein importance sampling and Stein thinning, where the goal is to assign optimal weights to a subset of sample particles in order to construct the best possible approximations of a distribution $P$.

The proposed method introduces a new target distribution, denoted as $\Pi$, which is obtained by tilting the original target density $p(x)$ with the square root of a Stein kernel $k_P(x)$. This construction is derived by solving a variational problem that minimises the trace of the variance of a limiting Gaussian distribution. The use of $\Pi$ as a target for the Metropolis-Adjusted Langevin Algorithm (MALA) instead of the original target distribution $P$ is justified through an almost sure consistency guarantee (Theorem 1) and numerical experiments conducted on a set of benchmark problems.

**Strengths:**

**Originality**: Whilst previous works have extensively studied post-processing using Stein's discrepancy, this paper introduces a novel perspective on improving Stein importance sampling through the design of a target distribution that is distinct from the original target $P$ but more suitable for post-processing techniques. This novel approach of leveraging the target design to improve post-processing methods is both interesting and original.

**Quality**: The construction of the proposed target distribution $\Pi$ is clearly explained (Section 3.1). The paper presents both a theoretical guarantee (Theorem 1) and extensive numerical evidence to support the choice of $\Pi$. The assumptions in Theorem 1 appear to be mild, and the authors provide comprehensive discussions comparing them with similar conditions in related literature. Overall, the significance of the proposed method is convincingly demonstrated.

**Clarity**: This paper exhibits a high level of clarity throughout, including an extensive review of related literature and methodologies (Section 2).

**Weaknesses:**

**Motivation**: While the authors have effectively justified the use of the proposed target $\Pi$ instead of the original target for constructing MALA samplers, the rationale for employing S$\Pi$IS over running MALA **without** post-processing is slightly weak. Specifically:

1. A main advantage of Stein importance sampling (SIS) lies in its ability to provide unbiased estimation when the MCMC sampler used to generate the sample is biased. Consequently, allocating computational resources to post-processing is crucial in such cases, as it is uncertain whether running the chain for a longer time would improve the quality due to the bias. However, for $\Pi$ sampling, one must set up an unbiased sampler (e.g., MALA) that targets $\Pi$, raising the question of whether conducting S$\Pi$IS is more beneficial than simply allocating the same computational budget to running the chain for an extended period. The reported experimental results do not seem to address this question, as they solely compare S$\Pi$IS with standard SIS. Including experiments comparing S$\Pi$IS with MALA without post-processing would be valuable in addressing this concern.
2. S$\Pi$IS requires the setup of an MCMC sampler targeting a distribution different from $P$. Thus, the generated sample must be used in conjunction with the post-processing step to perform inference on $P$. This contrasts with standard SIS, where samples can be drawn from a sampler targeting $P$, which is typically what practitioners would do regardless of their intention to use post-processing methods. This raises the question of whether setting up a sampler that targets such a specialised distribution and relies on post-processing is more practically attractive than directly targeting $P$. Including discussions addressing these concerns would be beneficial.

**Questions:**

L174: Could you elaborate on what is the relaxation and what one loses due to this relaxation, if any?

Figure 1.2: It seems that for small $n$, targeting $P$ is more beneficial than targeting $\Pi$. Is it purely coincidental, or does it reflect a specific characteristic or trade-off associated with the methodology? Could you offer some insights into this observation?

L229: Could you explain why the co-domain of $D_P$ is $\mathcal{X} \times \mathcal{X}$ instead of $\mathcal{P}(\mathcal{X})$?

Table 1: Are the results averages over multiple repetitions or obtained from a single repetition? t

**Limitations:**

Some limitations of Stein IS and the use of different Stein kernels are addressed in Section 4 and Section 3.2.

---

> ### Author Rebuttal · Authors · 2023-08-04
>
> Thank you for your thoughtful report.
>
> > A main advantage of Stein importance sampling (SIS) lies in its ability to provide unbiased estimation when the MCMC sampler used to generate the sample is biased.  However, for $\Pi$ sampling, one must set up an unbiased sampler (e.g., MALA) that targets $\Pi$
>
> This is a good point; for example, the unadjusted Langevin algorithm (ULA) can generate approximate samples at lower computational cost compared to MALA, and Stein Importance Sampling can be used to retrospectively correct for the bias in ULA.  If it were clear how to design a ULA algorithm to target $\Pi$ then this could be directly applied in Stein $\Pi$ Importance Sampling, however in general the explicit characterisation of the invariant distribution for ULA is intractable, making it unclear how to proceed.  This would be an interesting direction for further research and we will add a discussion on this point to the conclusion section of the revised manuscript.
>
> > Including experiments comparing S$\Pi$IS with MALA without post-processing would be valuable in addressing this concern.
>
> Please allow us to point out that the columns labelled "MALA" in Table 1 correspond to classical $P$-invariant MALA, which we believe is what you have asked for?  It is true that, then $P$ and its gradients are cheap to evaluate, the computational cost of MALA is lower than that of S$\Pi$IS-MALA, and one could run more iterations of MALA for an equivalent computational cost.  But for more complex $P$ the computational cost of all algorithms will be gated by the number of times $P$ and its gradients need to be evaluated, making the direct comparison in Table 1 meaningful.  Further, if we aim for a compressed representation of $P$, then some form of post-processing of MALA would be required, which would then entail an additional computational cost.  We aim to discuss these important points in more detail in the revised manuscript.
>
> > This raises the question of whether setting up a sampler that targets such a specialised distribution and relies on post-processing is more practically attractive than directly targeting $P$. Including discussions addressing these concerns would be beneficial.
>
> As we also mentioned to Reviewer X2c9, at present it is unclear whether these algorithms will stand the test of time compared to MCMC, but we believe they are certainly worth investigating.  We will expand the conclusion section of the manuscript to address this broader point.
>
> > L174: Could you elaborate on what is the relaxation and what one loses due to this relaxation, if any?
>
> Actually, nothing was lost due to relaxing the constraints (S1-2), since we verified in lines 177-178 that the solution to the relaxed problem also happens to satisfy the constraints (S1-2).
>
> > Figure 1.2: It seems that for small $n$, targeting $P$ is more beneficial than targeting $\Pi$. Is it purely coincidental, or does it reflect a specific characteristic or trade-off associated with the methodology? Could you offer some insights into this observation?
>
> In this particular example $P$ is uni-modal while $\Pi$ is multi-modal, and we conjecture that there is a "warm up" period where samples are needed in each of the "modes" of $\Pi$ before S$\Pi$IS starts to perform well.  In contrast, SIS requires only samples from $P$, which is uni-modal, leading to a shorter "warm up" period.
>
> > L229: Could you explain why the co-domain of $D_P$ is $\mathcal{X} \times \mathcal{X}$ instead of $\mathcal{P}(\mathcal{X})$?
>
> Thank you for catching this typo!  Indeed, $D_P : \mathcal{P}(\mathcal{X}) \rightarrow [0,\infty]$.
>
> > Table 1: Are the results averages over multiple repetitions or obtained from a single repetition?
>
> They are averages over ten replicates; in the caption we wrote that "ten replicates were computed", but we will make explicit that averages over the ten replicates are being reported.

---

> > ### Comment · Reviewer_pfRv · 2023-08-11
> >
> > Thank you for your detailed response, which answered all of my questions.

---

### Official Review · Reviewer_m4Rk · 2023-07-04

**Soundness:** 3 good
**Presentation:** 2 fair
**Contribution:** 3 good
**Rating:** 6
**Confidence:** 4

**Summary:**

The paper studies the design of the MCMC algorithm which is well suited for post-processing to obtain consistent approximation $P_n^\star$ of the target measure $P$ using Stein kernel discrepancies ($D_P( \cdot)$). The authors suggest the following novel procedure: (1) choose a measure $\Pi$ which differs from the target measure $P$ by the factor $\sqrt{k_p}$, where $k_p$ is the Stein kernel  (solving variational problem) (2); Sample from $\Pi$ using (pre-conditioned) MALA; (3) solve linearly-constrained quadratic problem or construct sparse approximation to obtain $P_n^\star$.
Theorem 1 provides assumptions to ensure convergence of Stein kernel discrepancies $D(P_n^\star)$ to zero.  Step (1) is based on SNIS procedure and the choice of $\Pi$ which gives the smallest variance of the Stein kernel discrepancies between SNIS estimate P_n and target measure $P$.
The results are illustrated by numerical experiments.

**Strengths:**

-Novel variational algorithm for explicit construction of measure \Pi.
-Theoretical analysis of the algorithm in the case of using MALA as a MCMC sampler


**Weaknesses:**

- Only asymptotic convergence in D_P metric is proved in Theorem 1.
- Would be good to sketch ideas of construction of Stein's kernel in the main text.
- It could be difficult to calculate Stein's kernel in more practical problems
- Works in moderate dimensions



**Questions:**

It is well known that MALA sampler is not good for mixtures of distributions. Is it possible to replace MALA by some other MCMC sampler as HMC or adaptive MCMC?


**Limitations:**

-

---

> ### Author Rebuttal · Authors · 2023-08-04
>
> Thank you for carefully considering our manuscript.
>
> > It is well known that MALA sampler is not good for mixtures of distributions. Is it possible to replace MALA by some other MCMC sampler as HMC or adaptive MCMC?
>
> This can of course be done in practice, but ensuring consistency of the resulting algorithm could be considerably more difficult.  It is certainly a good suggestion, and one that we will take forward in further work.

---

> > ### Comment · Reviewer_m4Rk · 2023-08-16
> >
> > Thanks a lot! I will keep my score.

---

### Official Review · Reviewer_yEHQ · 2023-07-04

**Soundness:** 2 fair
**Presentation:** 2 fair
**Contribution:** 2 fair
**Rating:** 6
**Confidence:** 2

**Summary:**

This paper proposes a proposal distribution $\Pi$ to generate finite points
such that a weighted version approximates the target distribution $P$ under
Stein discrepancy.

**Strengths:**

The main strength lies in a new proposal for the sampling distribution $\Pi$ that
is more efficient for follow-up approximation to the original distribution $P$ in the
sense of Stein discrepancy.

Asymptotic consistency is established, and extensive simulations on Bayesian computation
is also conducted to illustrate the benefit of the proposal.

**Weaknesses:**

The authors might consider expanding the section on the actual contribution (the proposal of $\Pi$),
and be more brief on the background. For instance, the section on Wasserstein distance doesn't seem necessary.

The second limitation is in the theoretical guarantees. Can one provide a non-asymptotic convergence guarantee
for the proposed method?

**Questions:**

1. Can the authors comment more on the effect of dimensionality on the improvement of $\Pi$ over $P$?

2. Can the same proposal work for other kernel functions ( non Stein)?

---

> ### Author Rebuttal · Authors · 2023-08-04
>
> Thank you for your positive comments on our manuscript.
>
> > the section on Wasserstein distance doesn't seem necessary
>
> Thank you, we will think carefully about how to improve all aspects of our presentation, bearing in mind that Reviewers fuoL and pfRv appreciated this part of the manuscript specifically ("the motivational section on the optimal quantization of Wasserstein distance is well-put", "a high level of clarity throughout, including an extensive review of related literature and methodologies").
>
> > Can one provide a non-asymptotic convergence guarantee for the proposed method?
>
> We believe this is possible in principle by exploiting Theorem 2 of Riabiz et al (2022).  However, this strategy would only provide a bound on $\mathbb{E}[\text{KSD}^2]$.  In contrast, our manuscript contains an almost sure (albeit asymptotic) convergence result.

---

> > ### Comment · Reviewer_yEHQ · 2023-08-15
> >
> > Thanks for the response. However the questions under Question section have not been addressed.

---

> > > ### Author Response · Authors · 2023-08-16
> > >
> > > Thank you, we apologise for our oversight:
> > >
> > > > 1. Can the authors comment more on the effect of dimensionality on the improvement of $\Pi$ over $P$?
> > >
> > > Appendix D.1 is dedicated to this point:  In brief, a naive kernel choice can make the diagonal $k_P(x,x)$ of the Stein kernel effectively a constant, in which case $\Pi$ becomes effectively identical to $P$ (see the difference between the Langevin and KGM kernels in Figure S1). This is not necessarily a problematic result, as it was already known that kernel choice is important for high-dimensional applications of KSD (e.g. see the discussion and guidance on kernel choice in Schrab et al, 2022).
> > >
> > > Some alternatives to KSD, such as Sliced KSD (Gong et al, ICLR 2021), have been proposed specifically for the high-dimensional context.  Though these alternatives do not currently enjoy the same convergence control guarantees as KSD, it could be interesting to seek an optimal choice of $\Pi$ for these discrepancies as well.  We will highlight this as a possible future research direction in the conclusion of the manuscript.
> > >
> > > > 2. Can the same proposal work for other kernel functions (non Stein)?
> > >
> > > The argument that we made for selection of $\Pi$ is not specific to a Stein kernel.  You may be wondering whether such an approach could be useful for Bayesian quadrature, for example?  The trouble here is that Bayesian quadrature is usually performed with a translation-invariant kernel, and for any translation-invariant kernel our $\Pi$ becomes equal to $P$.   In recent work , some authors have advocated for the use of non-stationary kernels as a default in Bayesian quadrature (Fisher et al, 2020).  Evaluating the potential benefit of sampling from $\Pi$ in the latter context could be an interesting avenue for further work, and we will also highlight this in the revised manuscript.
> > >
> > > We hope that these adequately address your questions, and thank you again for your report.
> > >
> > > References:
> > >
> > > Fisher MA, Oates CJ, Powell C, Teckentrup A., 2020. A Locally Adaptive Bayesian Cubature Method. International Conference on Artificial Intelligence and Statistics (AISTATS 2020).
> > >
> > > Gong, W., Li, Y. and Hernández-Lobato, J.M., 2021. Sliced Kernelized Stein Discrepancy. In International Conference on Learning Representations (ICLR 2021).
> > >
> > > Schrab, A., Guedj, B. and Gretton, A., 2022. KSD aggregated goodness-of-fit test. Advances in Neural Information Processing Systems, 35, pp.32624-32638.

---

### Official Review · Reviewer_X2c9 · 2023-07-05

**Soundness:** 4 excellent
**Presentation:** 4 excellent
**Contribution:** 4 excellent
**Rating:** 7
**Confidence:** 2

**Summary:**

The paper analyses which target distribution to use for MCMC in the situation where a stein-descrepency will be used to post-process it's output samples.
They propose to use a different target distribution for the MCMC to the distribution being approximated, and show this improves performance on a variety of posterior inference problems.

**Strengths:**

- The paper identifies and clear question: how to choose the invariant distribution \pi for MCMC if using a stein-descrepency to post-process the samples.
- They provide a method for selecting $\pi$ via a variational problem framing, where $\pi$ is selected to minimize the variance in the post-processed approximation. This gives a closed form expression for $\pi$ (up to a normalizing constant) such that it can be easily used within MCMC. Their analysis is agnostic to the choice of stein kernel making their results broadly applicable.
- Figure 1 nicely illustrates the property of over-dispersion that their choice of $\pi$ has, and shows that on a simple 1D problem that this results in lower error bars compared to using $P$ for the MCMC.
- In their experiments their proposed method is shown to consistently improve (better results in 70% of  PosteriorDB tasks) upon the baseline of setting $\pi$ to the target distribution being approximated.
- Their presentation is generally very clear, with informative figures provided.

**Weaknesses:**

I could not see any weaknesses in this paper, however the it's subfield is not within my area of expertise and I did not check the proofs.

**Questions:**

How does using the stein-discrepency **during** sample generation (e.g. stein variational gradient descent) compare to using it for post-processing?
I note that this question is not relevant for the contributions of the paper, but an answer would be useful for my understanding of the paper's general usefulness.

**Limitations:**

The authors address the limitations of their method in the discussion, namely that it requires second order derivatives of the model

---

> ### Author Rebuttal · Authors · 2023-08-04
>
> Thank you for your kind comments on our manuscript.
>
> > How does using the stein-discrepancy during sample generation (e.g. stein variational gradient descent) compare to using it for post-processing?
>
> There have been some attempts to directly address this issue, in particular Stein Points (Chen et al, ICML 2018), Stein Point MCMC (Chen et al, ICML 2019), and Kernel Stein Discrepancy Descent (Korba et al, ICML 2021).  Whilst these algorithms do make use of Stein discrepancy for guiding sampling, it has to be acknowledged that these algorithms are not widely used.  SVGD is more widely used, but it is a gradient flow on the KL divergence rather than on the KSD.  At present it is unclear whether these algorithms will stand the test of time compared to MCMC, but we believe they are certainly worth investigating.

---

> > ### Comment · Reviewer_X2c9 · 2023-08-11
> > **Thanks**
> >
> > Thank you for pointing out this literature. I have no further questions and am happy to recommend acceptance of the paper.

---

### Official Review · Reviewer_fuoL · 2023-07-16

**Soundness:** 3 good
**Presentation:** 4 excellent
**Contribution:** 3 good
**Rating:** 6
**Confidence:** 3

**Summary:**

This work proposes a design of a probability density that is more over-dispersed than the target density, so that, somewhat surprisingly, the resulting MCMC samples, after optimally reweighted, can achieve lower KSD than MCMC samples from the true target density. Consistency of the two proposed algorithms, SPiIS-MALA and SPiT-MALA, is proved. These two algorithms are benchmarked on PosteriorDB dataset to demonstrate their superior performance over raw MALA and SIS-MALA (MALA plus optimal reweighting).


**Strengths:**

- The paper tackles an interesting yet to my knowledge underexplored problem in sampling, which is how to design a density so that the resulting samples provide a good discrete support for the ensuing optimally reweighting step that finds a weight to minimize KSD.
- The angle of the attack is, although not new (e.g. Graf and Luschgy 2007), quite surprising (i.e. the density needs to be over-dispersed). The motivational section on the optimal quantization of Wasserstein distance is well-put.
- The consistency of the two proposed methods is proved which is nice.
- Many kernels are studied (Langevin-Stein/KGM/Riemann-Langevin-Stein) and are used in producing the empirical results.
- The writing of the paper is excellent and lots of intuition and toy examples are given to illustrate the points.


**Weaknesses:**

- The consistency proof of Theorem 1 seems like a rather straightforward application of Theorem 2 (Durmus and Moulines 2022) and Theorem 3(Riabiz et al. 2022). In particular, it seems to me the same proof should go through for quite generic $\Pi$, not necessarily the one that takes the form in (8). Hence, it is not clear whether the design (8) is theoretically justified, other than the heuristic argument given in Sec. 3.1.
- There is a considerable gap between the heuristic argument in Sec. 3.1 and the proposed algorithm, which is the weights used in Sec. 3.1 is not optimal but in the algorithms they are. The authors claimed the choice $dP/d\Pi$ is "near-optimal" (without justification), but then noted that using $dP/d\Pi$ will perform substantially worse than the optimal weight, which seems contradictory to the first claim.
- The experimental results comparing SIS-MALA and SPiIS-MALA are somewhat mixed. For an end user, there is no provided criterion on whether they should use the proposed method or the baseline SIS-MALA. Moreover, I cannot find standard derivation of the reported numbers.
  *  There seems to be missing experiments that benchmark the performance of SPiT-MALA compared to baseline (e.g. $\Pi=P4). The only experiment I found for SPiT-MALA is in D.6 where the consistency is verified.
- In many places it is hinted that $P$ will be close to $\Pi$ as the dimension $d$ increases. This implies that the proposed method is only applicable to small dimensions and thus the application value is limited.  Further analysis on the relation to dimension could be helpful. Moreover, there is only one experiment for $d=66$ (last row in Table 1) that corroborate the point that the extent of improvement decreases when the dimension increases; more experiments could be used to strengthen this point.


**Questions:**

- I would like to see a rigorous statement and proof of the heuristic argument from Sec 3.1. The sketched-out argument makes sense (other than one detail --- see below) so I'm wondering why it is not made into a complete proof.
  * In the paragraph below (S2), it seems to assume that $E_{x \sim \Pi}[\frac{dP}{d\Pi}(x)(k(\cdot, x) - \mu_P)] = 0$. Why is this true? I think $E_{x \sim \Pi}[\frac{dP}{d\Pi}(x)k(\cdot, x) - \mu_P] = 0$ but not when $\mu_P$ is multiplied by the importance weight. Of course, if $\mu_P = 0$ then it does not matter.
- A heuristic argument is given in D.1 for the Langevin-Stein kernel and $P$ is the standard $d$-dimensional Gaussian. Aside from this very simple $P$, is it true that $P \approx \Pi$ in general? Can we say anything theoretical about it?
- How is the simplex-constrained minimization in Algorithm 2 implemented? What is the time complexity? This also seems like the computational bottleneck of Alg. 2 and the reason $n$ is ony a few thousands in all experiments.
- For the experiment done in D.8, if the plot is for Wasserstein-1 distance, will $\Pi(1Wass)$ result in better numbers than $\Pi(KGM3)$?


**Limitations:**

The authors have adequately addressed the limitations.

---

> ### Author Rebuttal · Authors · 2023-08-04
>
> Thank you for your detailed report.
>
> >  it is not clear whether the design (8) is theoretically justified, other than the heuristic argument given in Sec. 3.1
>
> Our proposed $\Pi$ is not the only choice for which consistency can be established; consistent approximation is possible also for $\Pi = P$, for example.  Rather, we heuristically motivate a specific choice of $\Pi$ that is expected to out-perform alternatives, and then we verify that consistency occurs for this specific choice of $\Pi$.
>
> > The authors claimed the choice $dP/d\Pi$ is "near-optimal" (without justification), but then noted that using $dP/d\Pi$ will perform substantially worse than the optimal weight
>
> You are also correct that there is a performance gap between the weights that we analyse in Sec 3.1 and the Stein importance sampling weights; we will replace the phrase "near optimal" with a more nuanced explanation, which acknowledges the performance gap but explains that nevertheless the weights that we analyse in Sec 3.1 are expected to perform much better than simpler choices, such as uniform weights.
>
> > For an end user, there is no provided criterion on whether they should use the proposed method or the baseline SIS-MALA.
>
> We see this as a fundamentally difficult question, akin to asking how to pick a MCMC method.  In the MCMC setting, it is typical to try one algorithm and, if it is not performing well, to try a different algorithm instead.  On the other hand, if we are talking in theoretical terms, then low-to-moderate dimensional posteriors that are not highly multi-modal are likely to be well-suited to S$\Pi$IS-MALA, but otherwise plain MALA (rather than SIS-MALA) is likely to perform best.  We emphasise that this is due to the fundamental pathologies of KSD itself, rather than our algorithms to minimise it.  A discussion will be added to the manuscript.
>
> > I cannot find standard derivation of the reported numbers.
>
> The full results, including standard error bars, are contained in supplemental Appendix D.5 - space considerations prevented us from including these in Table 1.  It can be verified that the error bars are all relatively small.
>
> > There seems to be missing experiments that benchmark the performance of SPiT-MALA compared to baseline (e.g. $\Pi=P$).
>
> Thank you for the opportunity to discuss this point - we are confident that S$\Pi$T-MALA can outperform the baseline Stein Thinning algorithm with $\Pi = P$, but this will occur only at large sample sizes $n$.  The reason for this is that Stein Thinning is a greedy algorithm which favours the inclusion of high probability samples in its initial phase - once the modes have been well described it will only then move on to sampling from the tail.  We wanted to demonstrate this phenomenon, but it appears that in many cases we need $n > 3,000$ to see the behaviour just described.  Due to the super-linear cost of S$\Pi$T-MALA, we have so far not been able to deploy sufficient computational resources to thoroughly examine this effect.  This, together with the development of more efficient approximations to S$\Pi$T-MALA, are actives area of ongoing work.  As such, we focused most the manuscript on S$\Pi$IS-MALA, only noting in passing that we also obtain a consistency proof for S$\Pi$T-MALA.
>
> > There is only one experiment for $d=66$ that corroborate the point that the extent of improvement decreases when the dimension increases; more experiments could be used to strengthen this point.
>
> This is a good suggestion; at the time of writing this was the highest-dimensional model in PosteriorDB that complied, but we can explore the inclusion of other high-dimensional examples in the revised manuscript.
>
> > I would like to see a rigorous statement and proof of the heuristic argument from Sec 3.1
>
> With respect, we believe everything here is rigorously stated and proven, albeit not within a theorem environment in latex.
>
> > In the paragraph below (S2) it seems to assume that $\mathbb{E}_{x \sim \Pi} [ \frac{\mathrm{d}P}{\mathrm{d}\Pi}(x) ( k(\cdot,x) - \mu_P(\cdot) ) ] = 0$.  Why is this true?
>
> The equality holds due to the following argument:
>
> $\mathbb{E}_{x \sim \Pi} [ \frac{\mathrm{d}P}{\mathrm{d}\Pi}(x) ( k(\cdot,x) - \mu_P(\cdot) ) ]$
>
> $=\mathbb{E}_{x \sim P} [ k(\cdot,x) - \mu_P(\cdot) ]$
>
> $=\mathbb{E}_{x \sim P} [ k(\cdot,x) ] - \mu_P(\cdot) = \mu_P(\cdot) - \mu_P(\cdot) = 0$
>
> > Is it true that $P \approx \Pi$ in general?
>
> Since $\frac{\mathrm{d}\Pi}{\mathrm{d}P}(x) = k_P(x,x)^{1/2}$, the difference between $P$ and $\Pi$ is driven by the Stein kernel $k_P$.  While $k_P(x,x)$ is usually an unbounded function as $\|x\| \rightarrow \infty$, the tail behaviour is controlled by the choice of Stein operator and base kernel.  There are of course moment-type constraints on $\Pi$ for consistency of SPiIS-MALA than mean it cannot differ arbitrarily from $P$ for our theory to hold.
>
> > How is the simplex-constrained minimization in Algorithm 2 implemented? What is the time complexity?
>
> This is a linearly-constrained quadratic programme which we solved in Python 3.10.4 using the qpsolvers package version 3.4.0 as the frontend in conjunction with the ProxSuite package version 0.3.6 serving as the backend.  The full details are contained in the accompanying code, and we will add an explicit mention of the packages that can be used to run Algorithm 2 into the main text.  The time complexity is difficult to quantify, but we believe it is upper-bounded by $O(n^3)$.  These details will be included in the revised manuscript.
>
> > For the experiment done in D.8, if the plot is for Wasserstein-1 distance, will $\Pi$ (1-Wass) result in better numbers than $\Pi$ (KGM3)?
>
> We can certainly investigate -- but Wasserstein-1 and KSD (KGM3) are quite different performance metrics, the former not capturing convergence of second and third moments, unlike the latter.  Our aim in this work was limited to designing an algorithm that is able to minimise a user-specified KSD.

---

> > ### Comment · Reviewer_fuoL · 2023-08-12
> >
> > Dear authors,
> >
> > Thank you for the detailed response that has clarified all of my questions.
> >
> > I would like to keep my current score due to a few areas that can still be improved:
> > 1. S$\Pi$T-MALA lacks empirical verification (or at least a discussion of the challenge of empirical verification as you mentioned in the rebuttal);
> > 2. More empirical or theoretical justification to the hypothesis that the benefit of S$\Pi$IT-MALA vanishes as the dimension increases;
> > 3. Important details should be added in the main text (or add in appendix and refer in the main text), such as the standard deviation in Table 1, the implementation of Algorithm 2 and its complexity (and whether the QP solver is exact or approximate; if latter what is the error);
> > 4. Turn the "heuristic" argument in 3.1 into a rigorous statement and perhaps add more math background on the Hilber-space CLT result used, how to compute the trace of $\mathcal{C}$, etc.

---

### Decision · Program_Chairs · 2023-09-21

**Decision:**

Accept (spotlight)

**Comment:**

This paper was well received by all the reviewers. The paper makes an elegant contribution the active field of using stein methods for MCMC processing. They consider the task of optimizing the target distribution of an MCMC where its known that stein-descrepency for target measure P will be used to post-process it's output samples. They show that surprisingly the optimal choice to target is not P, and provide a characterization of this choice. The paper is well written with numerous illustrations, solid theory, and a compelling experiment section, and thus a clear accept.